# Injury prevents Ras mutant cell expansion in mosaic skin

Sara Gallini[1], Karl Annusver[2,6], Nur-Taz Rahman[3,6], David G. Gonzalez[1], Sangwon Yun[1], Catherine Matte-Martone[1], Tianchi Xin[1], Elizabeth Lathrop[1], Kathleen C. Suozzi[4], Maria Kasper[2✉] & Valentina Greco[1,5✉]

Healthy skin is a mosaic of wild-type and mutant clones[1,2]. Although injury can cooperate with mutated Ras family proteins to promote tumorigenesis[3–12], the consequences in genetically mosaic skin are unknown. Here we show that after injury, wild-type cells suppress aberrant growth induced by oncogenic Ras. $Hras^{G12V/+}$ and $Kras^{G12D/+}$ cells outcompete wild-type cells in uninjured, mosaic tissue but their expansion is prevented after injury owing to an increase in the fraction of proliferating wild-type cells. Mechanistically, we show that, unlike $Hras^{G12V/+}$ cells, wild-type cells respond to autocrine and paracrine secretion of EGFR ligands, and this differential activation of the EGFR pathway explains the competitive switch during injury repair. Inhibition of EGFR signalling via drug or genetic approaches diminishes the proportion of dividing wild-type cells after injury, leading to the expansion of $Hras^{G12V/+}$ cells. Increased proliferation of wild-type cells via constitutive loss of the cell cycle inhibitor p21 counteracts the expansion of $Hras^{G12V/+}$ cells even in the absence of injury. Thus, injury has a role in switching the competitive balance between oncogenic and wild-type cells in genetically mosaic skin.

Throughout our lifetimes, we acquire mutations in our skin, owing to its constant exposure to environmental insults. As a result, phenotypically normal skin contains a mosaic of epithelial stem cells with somatic mutations, including in genes that are associated with cancer development, such as the GTPase Ras family[1,2]. Constitutive activation of Ras oncogenes has been identified as the initial genetic event in 3–30% of human cutaneous squamous cell carcinomas[13–15] (cSCCs) and in experimentally induced cSCCs in mice[16,17]. In mouse models with mosaic epithelial expression of the constitutive active form of Hras ($Hras^{G12V/+}$), mutant cells outcompete wild-type cells and expand in the uninjured skin epidermis[18–20]. Although activated-Hras mutant cells are tolerated within otherwise wild-type and uninjured skin epithelium[18–20], injury has been shown to cooperate with oncogenic mutations to trigger tumorigenesis in various mouse models[3–12]. We hypothesized that the expansion of $Hras^{G12V/+}$ cells in the epidermis could represent a vulnerability upon injury; $Hras^{G12V/+}$ cells could futher expand and lead to tumours. For instance, the hyperproliferative environment generated during injury repair may further stimulate the proliferative behaviour of mutant cells and break the tolerance of the tissue. Here we investigated how injury affects the oncogenic potential of $Hras^{G12V/+}$ within a genetically mosaic and phenotypically relevant context.

## Injury suppresses growth in *Hras* mosaics

The stratified skin epidermis is uniquely accessible for direct observation, which enables the visualization of the emergence of aberrant growth at single-cell resolution. The basal layer contains epidermal stem cells, which can self-renew to generate more basal cells or differentiate and delaminate upwards to replace outer, barrier-forming cells[21,22] (Extended Data Fig. 1a). We hypothesized that injury repair would cooperate with the constitutive activation of the *Hras* oncogene ($Hras^{G12V}$) to promote tumorigenesis in phenotypically normal, genetically mosaic skin. To test this hypothesis, we generated mice in which we could induce and follow populations of $Hras^{G12V/+}$ mutant cells within wild-type epithelium (*Krt14*-CreER; flox and replace (FR)-$Hras^{G12V/+}$; Lox-STOP-Lox (LSL)-tdTomato; *Krt14*-H2B−GFP; Methods). In these mice, tamoxifen treatment activates Cre in keratin 14-expressing basal cells and, in turn, induces the co-expression of $Hras^{G12V/+}$ from its endogenous promoter and a cytoplasmic fluorescent tdTomato reporter that provides an approximation of mutant cells. Moreover, these mice also express histone H2B-GFP in basal cells, which persists throughout differentiation[23], enabling the visualization of all basal stem cells and their progeny (Fig. 1a). We treated mice with tamoxifen at 3 weeks of age and, three days later, introduced a full-thickness injury down to the cartilage (4 mm diameter punch biopsy) in one ear. We used two doses of tamoxifen to drive $Hras^{G12V/+}$ expression in either approximately 99% of basal cells ($Hras^{G12V/+}$ max) to recapitulate previous studies of homogeneous models[3–5,24,25], or in approximately 65% of basal cells ($Hras^{G12V/+}$ mosaic) to mimic genetically mosaic skin (Fig. 1b). As a control, we also engineered *Krt14*-CreER; LSL-tdTomato; *Krt14*-H2B−GFP mice and treated them similarly to drive tdTomato expression in approximately 65% of wild-type basal cells (wild-type mosaic) (Fig. 1b). By longitudinally imaging the same

[1]Department of Genetics, Yale School of Medicine, New Haven, CT, USA. [2]Department of Cell and Molecular Biology, Karolinska Institutet, Stockholm, Sweden. [3]Bioinformatics Support Program, Cushing/Whitney Medical Library, Yale School of Medicine, New Haven, CT, USA. [4]Dermatologic Surgery, Yale School of Medicine, New Haven, CT, USA. [5]Departments of Cell Biology and Dermatology, Yale Stem Cell Center, Yale Cancer Center, Yale School of Medicine, New Haven, CT, USA. [6]These authors contributed equally: Karl Annusver, Nur-Taz Rahman. ✉e-mail: maria.kasper@ki.se; valentina.greco@yale.edu

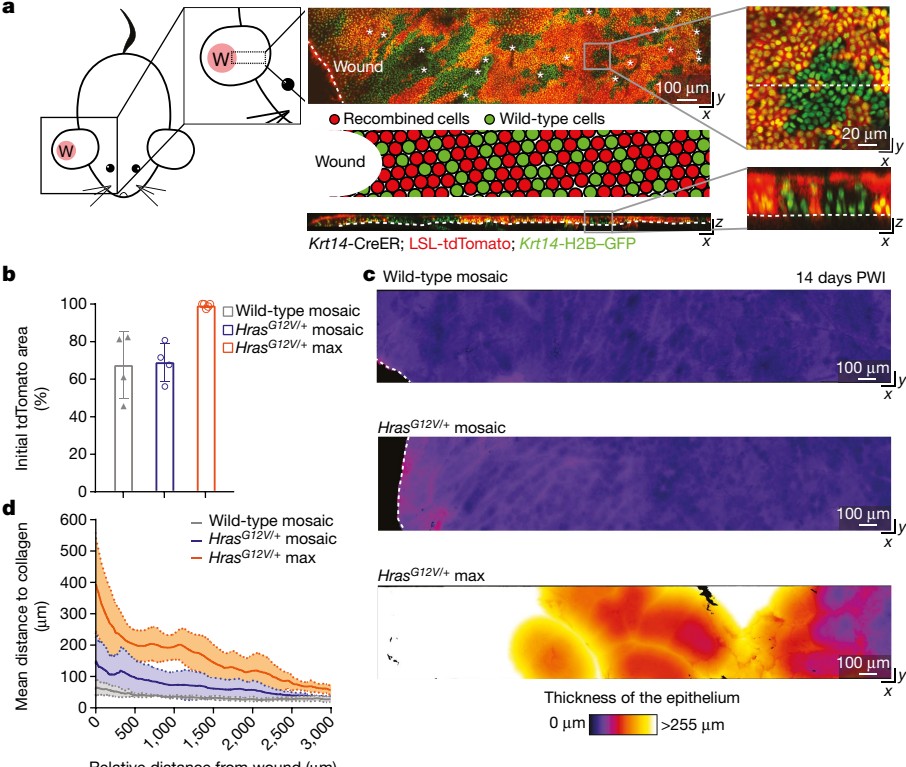

**Fig. 1 | Injury-induced aberrant growth of $Hras^{G12V/+}$ cells is suppressed in mosaic skin. a**, Left, cartoon depicting the 4-mm full-thickness wound (W) on a mouse ear and the imaged area. Centre, top-down (x–y; top) and transverse (x–z; bottom) views of a two-photon image of the skin epithelium at 14 days PWI from a *Krt14*-CreER; LSL-tdTomato; *Krt14*-H2B–GFP mouse (asterisks, hair canals; dashed lines, basement membrane in x–z view and wound edge in x–y view). Centre middle, cartoon schematic of wild-type (green) and recombined cells (red) after tamoxifen injection around the injury. Right, magnification of top-down (x–y) and transverse (x–z) views of the skin epithelium show epithelial cell nuclei (*Krt14*-H2B–GFP) in green and recombined cells expressing tdTomato in red (dashed lines mark the transversal section

in the x–y view and the basement membrane in the x–z view). **b**, The initial tdTomato⁺ area 3 days PWI (n = 4 wild-type mosaic and $Hras^{G12V/+}$ mosaic mice and n = 5 $Hras^{G12V/+}$ max mice). At least three independent areas of approximately 300 μm² were analysed for each mouse (Methods). Data are mean ± s.d. **c**, Heat maps of the top-down (x–y) view of representative two-photon images adjacent to the injury at 14 days PWI (dashed lines highlight the wound edge). Colour represents the thickness of the epithelium and identifies the presence of aberrant growth. **d**, The thickness of the epithelium at 14 days PWI around the wound. Solid lines represent means and dashed lines show s.d. n = 4 wild-type mosaic and $Hras^{G12V/+}$ mosaic mice and n = 5 $Hras^{G12V/+}$ max mice.

regions of the skin epidermis and always comparing $Hras^{G12V/+}$ mosaic models to wild-type mosaic models, we were able to control for any potential leakiness of the CreER system and study the consequences of Hras^{G12V/+} expression on the tissue overall and cell behaviours (see Methods).

We monitored the injured epithelium over time by combining deep tissue imaging with quantitative analyses using IMARIS and MatLab software, which enabled us to evaluate tissue thickness with intensity heat maps (Fig. 1c,d, Extended Data Fig. 1f, Supplementary Videos 1–4 and Methods). Notably, at 14 days post-wound induction (PWI), our $Hras^{G12V/+}$ mosaic models, which are on an outbred genetic background, did not develop the aberrant growth and thick epithelium observed in $Hras^{G12V/+}$ max models (Fig. 1c,d, Extended Data Fig. 1f, Supplementary Videos 1–4 and Methods). Histopathological analysis further showed that the skin epithelium around the repaired injury in the $Hras^{G12V/+}$ mosaic model was normal, despite the high burden of the $Hras^{G12V/+}$ mutation (Extended Data Fig. 1b,c,e,g,i). By contrast, abnormal growth formed rapidly within two weeks after injury induction in $Hras^{G12V/+}$ max, as expected (Extended Data Fig. 1d,e,h,i).

Collectively, these data show that $Hras^{G12V/+}$ cells break homeostatic tissue architecture during injury repair only when nearly all cells in the basal stem cell layer have this genotype.

## Competition dynamics switch upon injury

Having found that injury repair does not trigger aberrant growth in $Hras^{G12V/+}$ mosaic tissue, we next investigated how $Hras^{G12V/+}$ and wild-type cells within mosaic epithelia respond to injury. Previous studies had shown that embryonically induced $Hras^{G12V/+}$ basal stem cells integrate and expand in the skin epidermis, eventually outcompeting wild-type cells[18–20]. We had hypothesized that this proliferative advantage of $Hras^{G12V/+}$ cells would be amplified during injury repair, which has a higher proliferative demand than uninjured skin. To test our hypothesis, we revisited the same wild-type and $Hras^{G12V/+}$ cells in the skin epidermis of live mice for one month, with or without injury repair (Fig. 2a and Extended Data Fig. 2a). Epithelial cells begin to contribute to re-epithelialization approximately three days after injury[26,27]. We therefore started our analysis three days PWI (six days after tamoxifen-induced mosaicism) (Fig. 2a). We drew boundaries between GFP⁺tdTomato⁺ and GFP⁺tdTomato⁻ regions and represented the tdTomato⁺ areas as a percentage of the total area (Fig. 2b–e).

We found that the Hras^{G12V/+}-tdTomato⁺ population expanded substantially in uninjured $Hras^{G12V/+}$ mosaic epithelium, to a greater extent than the tdTomato⁺ population in uninjured wild-type mosaic epithelium. In uninjured $Hras^{G12V/+}$ mosaic mice, Hras^{G12V/+} cells outcompeted

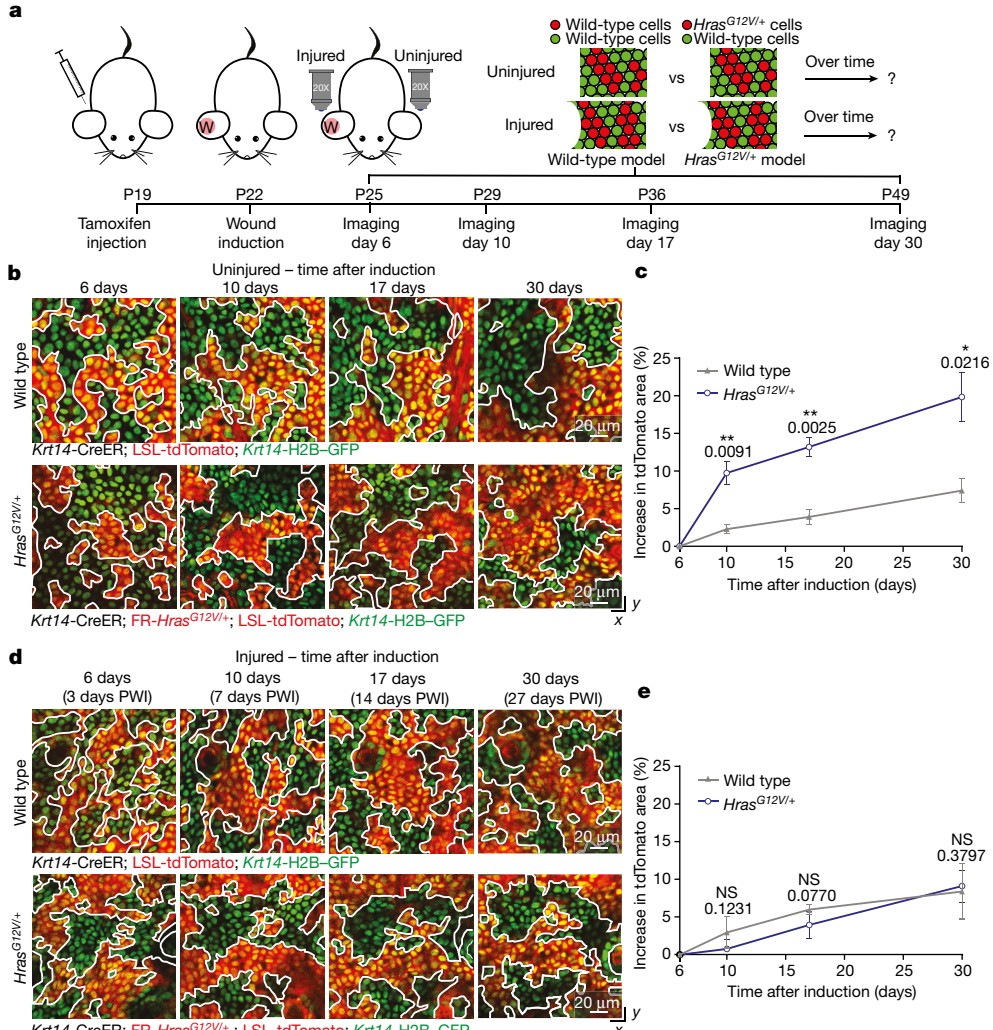

**Fig. 2 | Injury repair alters the competitive balance between wild-type and *Hras^G12V/+* cells in mosaic skin. a**, Schematic of the experimental design using the injury repair model. **b**, Representative two-photon revisit images of the basal stem cell layer of the epidermis. White lines highlight the boundaries between tdTomato^+ and tdTomato^− populations. Epithelial nuclei are in green (*Krt14*-H2B–GFP) and recombined cells are in red (LSL-tdTomato). **c**, The increase in tdTomato^+ area over time in the uninjured condition. *n* = 3 mice.

**d**, Representative two-photon revisit images of the basal stem cell layer of the epidermis during injury repair. **e**, The increase in tdTomato^+ area over time in the injured condition. *n* = 4 mice. Unpaired, two-tailed *t*-test comparing wild-type with *Hras^G12V/+* mice at different time points in uninjured and injured conditions. *P* values are shown. At least three independent areas of approximately 300 μm^2 were analysed for each mouse (Methods). Data are mean ± s.d. *P < 0.05, **P < 0.01, ***P < 0.001, ****P < 0.0001; NS, not significant.

wild-type cells and increased their occupancy of the basal stem cell layer by approximately 20% after one month, consistent with previous work using constitutive Cre expression[18] (Fig. 2b,c and Extended Data Fig. 2b). By contrast, one month after injury, *Hras^G12V/+* cells did not expand over wild-type cells and the proportion of tdTomato^+ cells was similar between wild-type and *Hras^G12V/+* mosaic models (Figs. 1b and 2d,e). Notably, these results were also observed in mosaicism with a lower Hras^G12V/+ mutational burden, which more closely mimic physiological conditions (Extended Data Fig. 2c–f). This finding was unexpected and indicates that *Hras^G12V/+* cells did not outcompete wild-type cells and expand after injury repair in *Hras^G12V/+* mosaic mice.

Collectively, these data demonstrate that the injury repair process does not amplify but rather abrogates the competitive advantage that *Hras^G12V/+* cells have over wild-type cells in the absence of injury.

## Cellular mechanism upon injury

The suppressed expansion of *Hras^G12V/+* cells after injury prompted us to investigate how this process affects different cellular behaviours of

mutant and wild-type cells. Specifically, we first examined the number of dividing cells, given that Ras family proteins are key regulators of epithelial cell proliferation in the skin epithelium. Indeed, epithelial stem cells in vitro and in vivo do not proliferate upon ablation of all Ras isoforms[28]. To score the number of dividing cells, we immunostained cells for the mitotic marker phosphorylated histone H3 (p-histone H3). We observed an increase in the amount of epithelial cell divisions accompanying efficient wound repair at 3 days PWI in the wild-type mosaic model (Fig. 3a,b). In sharp contrast, although we observed an increase in mitotic events in wild-type cells in the *Hras^G12V/+* mosaic model, the proliferation of *Hras^G12V/+* cells was unaltered during repair (Fig. 3b). These findings were corroborated by measuring mitotic figures (Extended Data Fig. 3a–c). Therefore, wild-type cells have an unexpected and selective competitive advantage over *Hras^G12V/+* cells in the acute phase of injury repair.

To determine whether the increased number of wild-type cell divisions that we observed at 3 days PWI was sustained over time, we scored mitotic events over 4 weeks. In the wild-type mosaic model, the initial increase in the number of mitotic cells observed in tdTomato^+ and

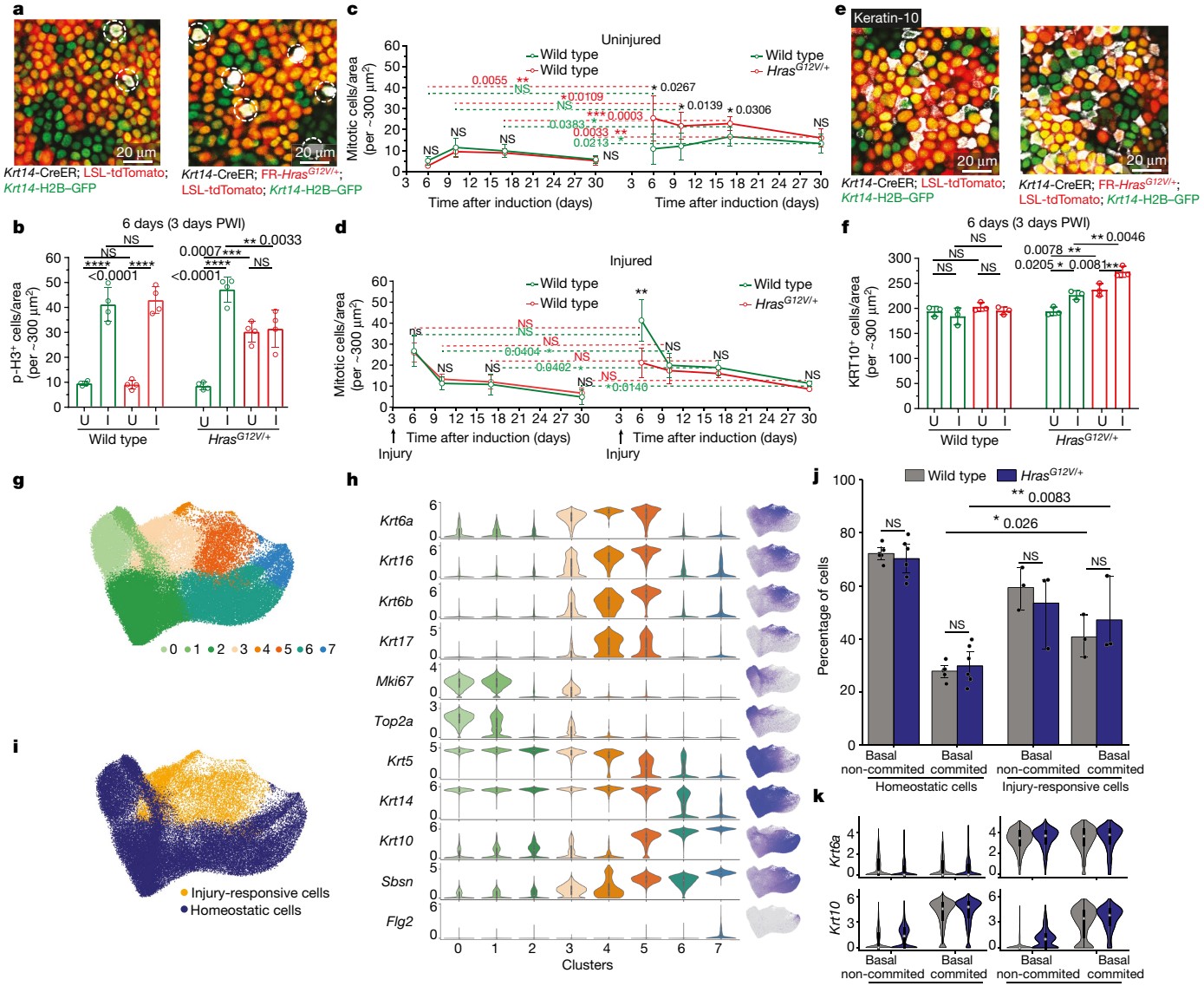

**Fig. 3 | Injury selectively induces the proliferation of wild-type cells in**
**$Hras^{G12V/+}$ mosaic skin. a**, Representative two-photon images of the epidermal
preparation, immunostained for p-histone H3 (highlighted by white dashed
circle). **b**, Quantification of p-histone H3-positive (p-H3⁺) cells in tdTomato⁺
and tdTomato⁻ populations in injured (I) and uninjured (U) skin. $n = 4$ mice.
**c,d**, Quantification of mitotic cells in uninjured (**c**) and uninjured (**d**) ears.
$n = 4$ mice. **e**, Representative two-photon images of the epidermal preparation,
immunostained for keratin-10 (white). **f**, Quantification of keratin-10-positive
(KRT10⁺) cells in tdTomato⁺ and tdTomato⁻ populations in injured and
uninjured skin. $n = 3$ mice. **b–d,f**, Paired or unpaired two-tailed $t$-test for
comparison between tdTomato⁺ and tdTomato⁻ populations in the same group
or in different groups of mice. At least three independent areas of approximately
300 μm² were analysed for each mouse (Methods). KRT10, keratin-10.
**g**, Uniform manifold approximation and projection (UMAP) showing epidermal
keratinocyte clusters from uninjured and injured conditions. **h**, Violin plots
(left) showing gene expression in clusters from **g**, together with the gene
expression superimposed on the UMAP (right). **i**, Cell classification into
homeostatic and injury-responsive populations based on clustering and gene
expression, overlaid on the UMAP. **g–i**, $n = 12$ mice. **j**, Quantification of wild-type
and $Hras^{G12V/+}$ cells as a proportion of tdTomato⁺ cells in basal non-committed
and basal committed groups for homeostatic and injury-responsive cells.
Data are averaged results for each biological replicate. One-sided $t$-test with
Holm–Sidak correction. **k**, Violin plots showing $Krt6a$ and $Krt10$ expression in
cells grouped as in **j**. In **h,k**, internal box plots denote the 25th, 50th and 75th
centiles, with whiskers depicting minima and maxima, excluding outliers that
are beyond 1.5× the interquartile range. **j,k**, Homeostatic cells from $n = 12$ mice
and injury-responsive cells from $n = 6$ mice. $P$ values are shown. Data are
mean ± s.d.

tdTomato⁻ wild-type cells returned to baseline by 7 days PWI (10 days
after tamoxifen-induced mosaicism) and appeared similar to unin-
jured mice, as expected (Fig. 3c,d). The initial increase in divisions of
wild-type cells observed in the injured $Hras^{G12V/+}$ mosaic model also
decreased over time, and eventually returned to a similar level as in
neighbouring $Hras^{G12V/+}$ cells, but remained higher than in wild-type
cells following injury in the wild-type mosaic model (Fig. 3c,d). By
contrast, the proliferative capacity of $Hras^{G12V/+}$ cells in the $Hras^{G12V/+}$
mosaic model was not substantially affected by injury repair at any of
the time points analysed (Fig. 3c,d). Thus, the balanced mitotic events
of wild-type and $Hras^{G12V/+}$ cells sustained at later time points after injury
would effectively continue to prevent the expansion of $Hras^{G12V/+}$ cells
in the $Hras^{G12V/+}$ mosaic model.

To test whether injury repair leads to genotype-specific changes in
other cellular behaviours, we monitored apoptosis, which is an estab-
lished cell competition mechanism[29], and inhibited by Ras signalling[30].
We examined cell death by scoring for either nuclear fragmenta-
tion events or expression of an apoptotic marker, active caspase-3.

The overall frequency of apoptosis was low, and we did not observe significant differences in cell death events of wild-type or $Hras^{G12V/+}$ cells in mice with or without injury at 6 days after tamoxifen-induced mosaicism and at later time points (Extended Data Fig. 3d–g). Similarly, we found no evidence of cell senescence (Extended Data Fig. 3h). Differentiation is another mechanism of cell loss that could influence competition between wild-type and $Hras^{G12V/+}$ cells in the skin epidermis. We therefore comprehensively evaluated the number of differentiation events and the expression of early differentiation markers using both protein (immunostaining) and transcriptional (single-cell RNA sequencing (scRNA-seq)) analyses in uninjured and injured (3 days PWI) settings of wild-type mosaic and $Hras^{G12V/+}$ mosaic models (Fig. 3e–k and Extended Data Figs. 3i and 4a–h). The scRNA-seq analysis revealed an increased fraction of epidermal basal cells committing towards differentiation upon injury, which we inferred from the ratio of basal cell states (stem cells versus differentiation-committed[22]) and corresponding changes in differentiation and stemness transcripts (Fig. 3j,k and Methods). The proliferation signature that emerged in Gene Ontology terms for 'biological process' in uninjured settings did not appear in injured contexts when comparing $Hras^{G12V/+}$ mosaic to wild-type mosaic models, further corroborating the live-imaging observations (Extended Data Fig. 4i,j). To determine whether the increased differentiation depended on the genotype (wild-type versus $Hras^{G12V/+}$ in $Hras^{G12V/+}$ mosaic mice), we quantified the expression of the early differentiation marker keratin-10 in wild-type and $Hras^{G12V/+}$ cells by immunostaining. We observed a similar increase of differentiating cells for both genotypes upon injury compared to the uninjured condition, in contrast to the selective increase in the fraction of dividing wild-type cells (Fig. 3b,f). At 7 days PWI, we showed that the number of differentiating wild-type and $Hras^{G12V/+}$ cells in the $Hras^{G12V/+}$ mosaic model was similar, which in addition to their equal mitotic capacities, supports the maintenance of the mutant and wild-type clone sizes over time after injury (Fig. 3d and Extended Data Fig. 3i).

Collectively, these results show that injury increases the number of divisions of wild-type cells only, and therefore suppresses $Hras^{G12V/+}$ cell expansion in the $Hras^{G12V/+}$ mosaic model in the acute phase of injury repair.

## *Kras* mosaics mimic *Hras* mosaics

Next, we investigated whether the injury repair process in mosaic skin could effectively suppress $Kras^{G12D}$, a more aggressive Ras family oncogene. Mice with homogeneous $Kras^{G12D/+}$ activation in the skin epidermis rapidly develop oncogenic growth in areas of constant abrasion[31,32] (data not shown). *Kras* is one of the most frequently mutated oncogenes in human cancer, and is broadly activated across epithelial cancers, including cSCCs[33]. We generated mice in which we could induce and follow $Kras^{G12D/+}$ cells within wild-type epithelium (*Krt14*-CreER; LSL-$Kras^{G12D/+}$; LSL-tdTomato; *Krt14*-H2B–GFP; Methods). Similar to $Hras^{G12V/+}$ tdTomato⁺ cells in the $Hras^{G12V/+}$ mosaic model, $Kras^{G12D/+}$ tdTomato⁺ cells expanded in the uninjured $Kras^{G12D/+}$ mosaic mouse skin but did not expand after injury (Fig. 4a,b and Extended Data Fig. 5a). We scored mitotic events before and after injury induction, and again observed a selective increase in the amount of dividing wild-type cells but not of mutant cells in the $Kras^{G12D/+}$ mosaic skin, similar to our observations in the $Hras^{G12V/+}$ mosaic model (Fig. 4c).

To monitor phenotypes at the tissue level, we applied two-photon microscopy with quantitative analyses of epidermal thickness represented by intensity heat maps. Despite the high burden of the $Kras^{G12D/+}$ mutation (approximately 65% of recombined cells), the skin epithelium of $Kras^{G12D/+}$ mosaic mice remained similar to wild-type mosaic models after injury (Figs. 1c and 4d,e and Extended Data Fig. 5a,b,d). By contrast, the $Kras^{G12D/+}$ max model, in which nearly all basal cells had the $Kras^{G12D/+}$ genotype, displayed rapid abnormal growth within the first two weeks after injury (Fig. 4d,e and Extended Data Fig. 5a,c,d).

Overall, our work strongly suggests that the selective increase in wild-type cell divisions during injury repair of mosaic skin limits the expansion of mutant cells expressing different oncogenic variants of the Ras gene family.

## Molecular mechanism upon injury

We found that injury repair in mosaic skin triggers a specific competitive advantage of wild-type cells but not $Kras^{G12D/+}$ and $Hras^{G12V/+}$ cells. However, it remained unclear whether the increased fraction of dividing wild-type cells per se prevented the expansion of Ras mutant cells. To identify the mechanisms responsible for this competitive switch, we extended our scRNA-seq analysis to epithelial cells, fibroblasts and immune cells in our wild-type and $Hras^{G12V/+}$ models (Extended Data Figs. 6 and 7), to analyse the expression of growth factors, among other soluble mediators, that influence epithelial cell proliferation. Across all injured models, we found a particularly high enrichment of EGFR ligands in both fibroblasts and epithelial cells when compared to other growth factors (Fig. 5a and Extended Data Figs. 6 and 7). Notably, EGFR is one of the best characterized upstream activators of the Ras pathway and previous studies have shown that EGFR signalling promotes epithelial cell proliferation during injury repair in wild-type skin[34–36]. In agreement with this, our western blot analyses and quantification from wild-type mice showed increased EGFR activation in injured versus uninjured conditions (Fig. 5b; for gel source data, see Supplementary Fig. 1). Additionally, $Hras^{G12V/+}$ cells displayed a downregulation of the receptor in uninjured and injured skin epithelium, as previously shown in vitro[37] (Fig. 5b; for gel source data, see Supplementary Fig. 1). We therefore designed a strategy to inhibit EGFR to selectively reduce the number of dividing wild-type cells after injury. We used the EGFR inhibitor Gefitinib, which we verified repressed the activity of EGFR and its downstream target ERK1/2 during injury repair in wild-type mosaic mice (Extended Data Fig. 8a,b and Methods; for gel source data, see Supplementary Fig. 1). As expected, Gefitinib treatment selectively inhibited the amount of wild-type cell divisions but not those of $Hras^{G12V/+}$ cells, in $Hras^{G12V/+}$ mosaic models after injury (Fig. 5c). To assess how Gefitinib treatment affects cell competition during injury repair, we tracked the percentage of surface coverage of $Hras^{G12V/+}$ cells (tdTomato⁺) at 3, 7 and 14 days PWI (6, 10 and 17 days, respectively, after tamoxifen-induced mosaicism). We found that EGFR inhibition re-establishes the competitive advantage of $Hras^{G12V/+}$ cells over their wild-type neighbours during injury repair in mosaic mice (Fig. 5d and Extended Data Fig. 8e). Because this drug treatment broadly affects multiple cell types in addition to the epithelium, we generated a genetic model that targets the expression of the dominant negative (DN) form of EGFR (lacking the cytoplasmic tyrosine kinase domain) to the basal cells of the skin epidermis (*Krt14*-CreER; *Krt14*-rtTA; FR-$Hras^{G12V/+}$; LSL-tdTomato; TRE-*Egfr*-DN; Krt14-H2B–GFP; Extended Data Fig. 8c,d and Methods). This selective EGFR inhibition in the basal stem cell layer reduced the amount of dividing wild-type cells and re-established the competitive advantage of $Hras^{G12V/+}$ cells over their wild-type neighbours during injury repair of mosaic mice (Extended Data Fig. 8c,d).

Collectively, our data demonstrate that the EGFR–Ras signalling pathway is required to increase wild-type cell divisions after injury and to prevent $Hras^{G12V/+}$ cell expansion in mosaic models.

## Cellular mechanism without injury

To examine whether the increase in mitotic events in wild-type cells is sufficient to counteract the expansion of Ras mutant cells, we directly manipulated cell proliferation in an uninjured setting using a p21-null genetic model (constitutive p21^null mice[38]; the p21 gene is also known as *Cdkn1a*). This approach, in the absence of injury, enabled us to more directly assess the role of wild-type cell proliferation in preventing Ras mutant cell expansion, as it lacks the influence of other factors

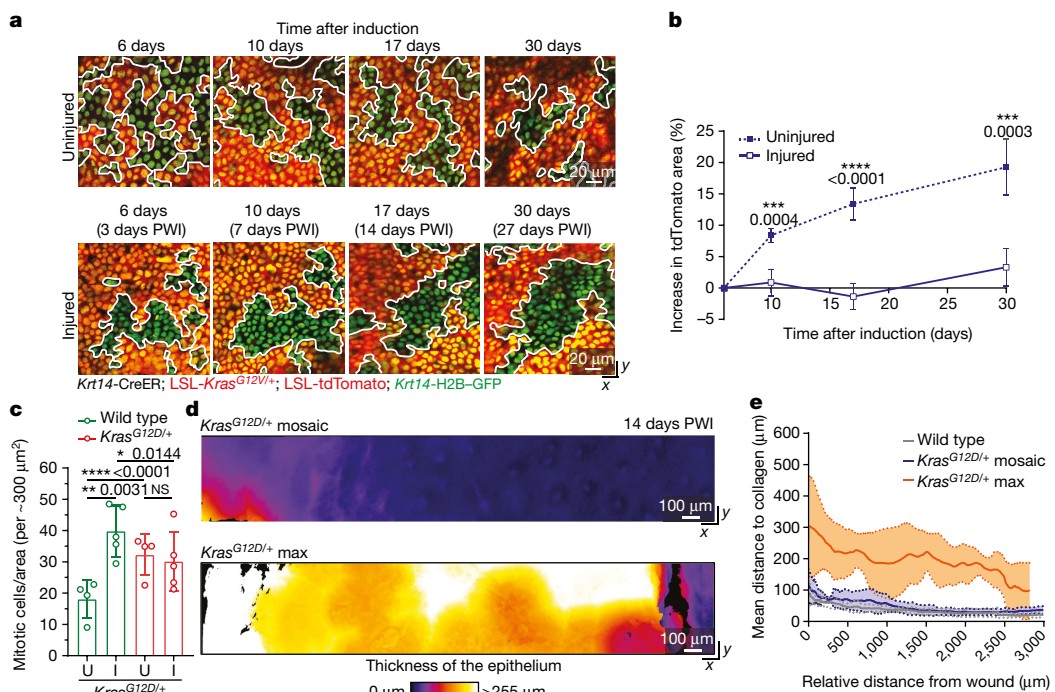

**Fig. 4 | *Kras*<sup>G12D/+</sup> cells lose their competitive advantage during injury repair of mosaic skin. a**, Representative two-photon revisit images of the epidermal basal stem cell layer. White lines mark borders between mutant and wild-type cells. **b**, The increase in tdTomato⁺ area in uninjured and injured conditions following induction with tamoxifen (uninjured *Kras*<sup>G12D/+</sup> mosaic, *n* = 4 mice; injured *Kras*<sup>G12D/+</sup> mosaic, *n* = 5 mice). **c**, Quantification of mitotic cells in tdTomato⁺ and tdTomato⁻ areas. Uninjured *Kras*<sup>G12D/+</sup> mosaic, *n* = 4 mice; injured *Kras*<sup>G12D/+</sup> mosaic, *n* = 5 mice. **d**, Heat maps of the top-down (*x*–*y*) view of representative two-photon images around the injury at 14 days PWI. Colour represents the thickness of the epithelium. **e**, Average epithelial thickness at 14 days PWI around the wound in wild-type (*n* = 3 mice), *Kras*<sup>G12D/+</sup> mosaic (*n* = 4 mice) and *Kras*<sup>G12D/+</sup> max (*n* = 3 mice). Solid lines represent means and dashed lines show s.d. **b,c**, Paired two-tailed *t*-test comparing tdTomato⁺ and tdTomato⁻ populations in the same group of mice. Unpaired two-tailed *t*-test comparing tdTomato⁺ and tdTomato⁻ populations in different groups of mice and *Kras*<sup>G12D/+</sup> mutant mice in uninjured and injured conditions at different time points. *P* values are shown. At least three independent areas of approximately 300 μm² were analysed for each mouse (Methods). Data are mean ± s.d.

that participate in injury repair (such as recruitment of immune cells and activated fibroblasts). The G1/S phase cyclin-dependent kinase inhibitor p21 is expressed in G1 phase cells to maintain skin epithelium homeostasis[39,40], and its genetic ablation leads to increase proliferation in wild-type mouse skin epithelial cells in vitro and in vivo[39–42]. Indeed, p-histone H3 immunostaining of a p21-null model revealed a substantially increased number of dividing basal cells, but not dermal cells, in uninjured mice (Extended Data Fig. 8f–h). We hypothesized that *Hras*<sup>G12V/+</sup> cells downregulate p21 expression to promote mitotic events. p21 expression was significantly reduced at both mRNA and protein levels in *Hras*<sup>G12V/+</sup> mosaic models compared to wild-type mosaic models (Extended Data Fig. 9a–d). Therefore, we reasoned that constitutive p21 deletion in the *Hras*<sup>G12V/+</sup> mosaic model may selectively manipulate wild-type cells without affecting *Hras*<sup>G12V/+</sup> cells, which express low levels of p21. To this end, we combined the p21-null model with the tamoxifen-inducible *Hras*<sup>G12V/+</sup> model (*Krt14*-CreER; FR-*Hras*<sup>G12V/+</sup>; LSL-tdTomato; constitutive p21-null; *Krt14*-H2B–GFP; Methods). p21 loss increased the proliferation of wild-type cells but not that of *Hras*<sup>G12V/+</sup> cells in p21-null *Hras*<sup>G12V/+</sup> mosaic mice, mimicking the injured context (Fig. 5e,f). This p21-null-triggered increased proliferation was sufficient to suppress the competitive advantage of *Hras*<sup>G12V/+</sup> cells over time in uninjured mice (Fig. 5g,h and Extended Data Fig. 8i), recapitulating the effects of injury repair in *Hras*<sup>G12V/+</sup> mosaic mice. The fraction of differentiating wild-type cells was also accelerated by the loss of p21, which explains why wild-type cells did not outcompete *Hras*<sup>G12V/+</sup> cells in the p21-null *Hras*<sup>G12V/+</sup> mosaic model (Extended Data Fig. 8j).

To explore the molecular mechanism that specifically increases the division of wild-type cells but not *Hras*<sup>G12V/+</sup> cells in p21-null and injury models, we probed the activation status of key regulators

of cell proliferation downstream of Ras[43], ERK1/2 and AKT protein kinases (Extended Data Fig. 10a,b; for gel source data, see Supplementary Fig. 1). In uninjured, wild-type skin, there was a lower level of activated ERK1/2 (p-ERK1/2) when compared with *Hras*<sup>G12V/+</sup> mosaic and *Hras*<sup>G12V/+</sup> max models, as expected (Extended Data Fig. 10a). We observed increased levels of p-ERK1/2 in wild-type skin that was undergoing injury repair or lacked p21 (Extended Data Fig. 10a). However, the increase in p-ERK1/2 was not significantly different in *Hras*<sup>G12V/+</sup> mosaic and *Hras*<sup>G12V/+</sup> max models in uninjured and injured conditions (Extended Data Fig. 10a). Moreover, p-ERK1/2 levels were similar after injury in all three models (wild-type, *Hras*<sup>G12V/+</sup> mosaic and *Hras*<sup>G12V/+</sup> max) (Extended Data Fig. 10a). Of note, we performed p-ERK1/2 immunostaining to explore the ERK1/2 activity in individual cells. This experiment confirmed the data from the western blot analyses, with wild-type clones selectively increasing the number of cells expressing p-ERK1/2 upon wounding (Extended Data Fig. 10c,d). Finally, the activation of AKT, another downstream target of the Ras pathway, was not significantly affected by injury or p21 loss (Extended Data Fig. 10b).

Overall, these data suggest that injury repair or loss of p21 specifically increase the activity of ERK1/2, a downstream effector of Ras, in wild-type cells to increase the fraction of dividing cells, enabling them to effectively suppress the competitive advantage of oncogenic Ras mutant cells in mosaic mice.

## Discussion

Healthy tissues, including skin, harbour a number of somatic mutations, some of which are in known tumour driver genes[1,2,44,45]. Models have shown that tumours can arise from the accumulation of multiple

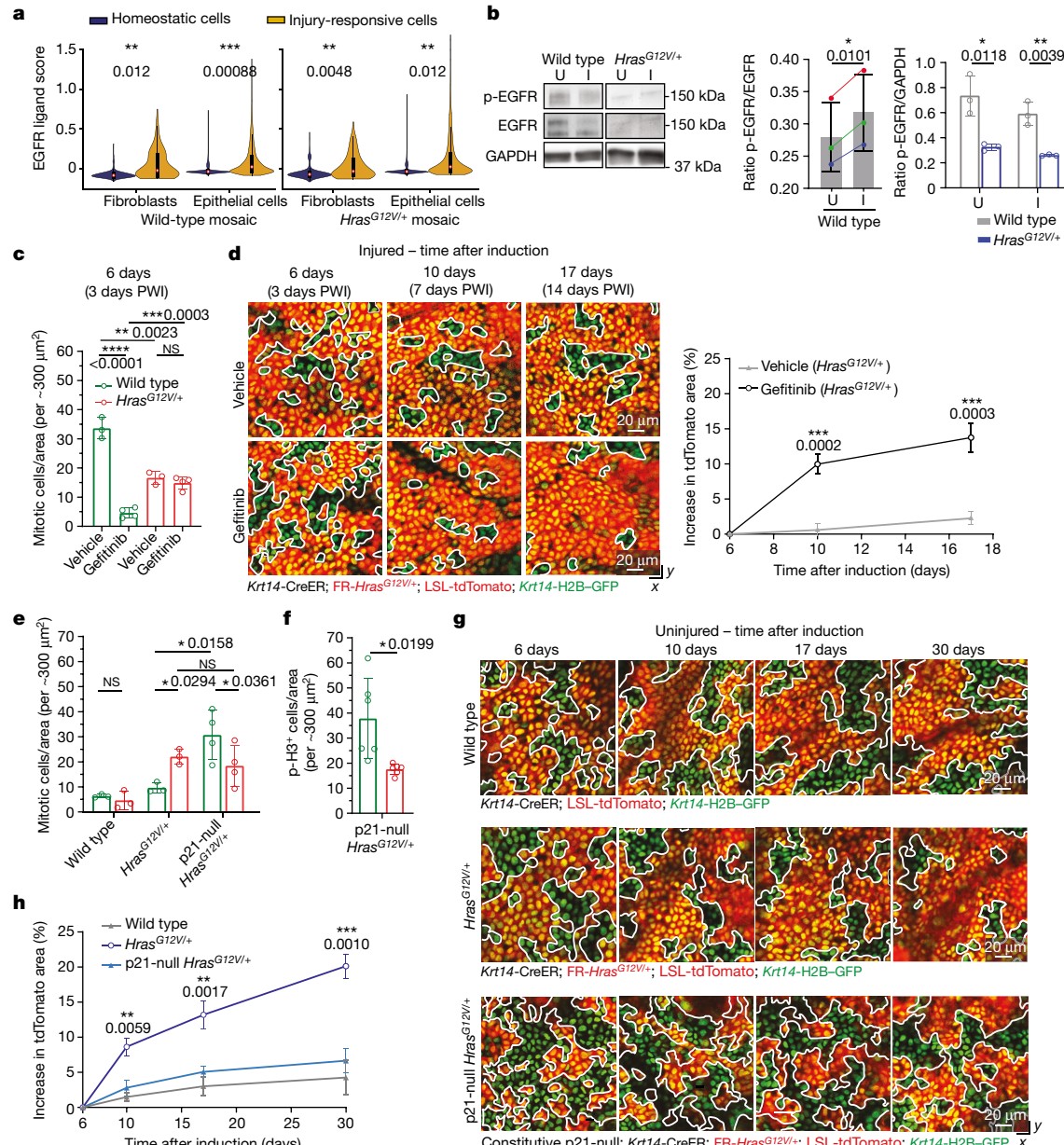

**Fig. 5 | Increased proliferation of wild-type cells is sufficient to counteract expansion of *Hras*$^{G12V/+}$ cells in mosaic skin. a**, Violin plots showing cell scoring based on expression of EGFR ligands (Extended Data Fig. 6i), separated by cell type and genotype. Two-tailed *t*-test comparing the averages of biological replicates according to conditions (fibroblasts: $n = 3$ wild-type mosaic and $n = 6$ *Hras*$^{G12V/+}$ mosaic; epithelial cells $n = 6$ mice per genotype), *P* values are shown. Internal box plots denote the 25th, 50th and 75th centiles, with whiskers depicting minima and maxima, excluding outliers that are beyond 1.5× the interquartile range. **b**, Western blot analysis (left) and quantification (middle) of phosphorylated EGFR (p-EGFR) and total EGFR in injured and uninjured conditions ($n = 3$ mice). Paired, two-tailed *t*-test. Pairs of coloured dots represent ratios for individual mice. Right, western blot analysis of total EGFR normalized to GAPDH ($n = 3$ mice). Blots were processed at the same time. Unpaired, two-tailed *t*-test. **c**, Quantification of mitotic cells in tdTomato$^+$ and tdTomato$^-$ areas. **d**, Left, representative two-photon revisit images following injury. Right, the increase in tdTomato$^+$ area after tamoxifen induction. In **c**,**d**, vehicle, $n = 3$ mice; Gefitinib, $n = 4$ mice. **e**, Quantification of mitotic cells in tdTomato$^+$ and tdTomato$^-$ areas in wild-type mosaic ($n = 3$ mice), *Hras*$^{G12V/+}$ mosaic ($n = 3$ mice) and constitutive p21-null *Hras*$^{G12V/+}$ mosaic ($n = 4$ mice). **f**, Quantification of p-histone H3-positive cells in tdTomato$^+$ and tdTomato$^-$ areas ($n = 6$ mice). **g**, Representative two-photon revisit images following injury in wild-type mosaic, *Hras*$^{G12V/+}$ mosaic and constitutive p21$^{null}$-*Hras*$^{G12V/+}$ mosaic mice. **h**, The increase in tdTomato$^+$ area. Unpaired, ordinary one-way ANOVA comparing wild-type mosaic, *Hras*$^{G12V/+}$ mosaic and constitutive p21$^{null}$-*Hras*$^{G12V/+}$ mosaic at different time points in the uninjured condition. In **g**,**h**, $n = 3$ wild-type mosaic mice, $n = 3$ *Hras*$^{G12V/+}$ mosaic mice and $n = 4$ constitutive p21$^{null}$-*Hras*$^{G12V/+}$ mosaic mice. **c**–**f**, Unpaired or paired two-tailed *t*-test for comparisons between different groups or within the same group of mice. *P* values are shown. At least three independent areas of approximately 300 µm$^2$ were analysed for each mouse (Methods). Data are mean ± s.d.

mutations or from a lower mutational burden cooperating with additional exogenous insults, such as injury[3–11,24,25]. Here we found that Ras mutant cells break tissue architecture during injury repair only when they represent nearly all cells in the basal stem cell layer of the skin epidermis. By contrast, when Ras mutant cells coexist with wild-type neighbours, injury selectively activates the endogenous proliferation programme in wild-type cells only, which counteracts the expansion of the Ras mutant cells (Extended Data Fig. 10e). Specifically, after

an initial spike following injury induction, the number of dividing wild-type cells equalizes to the level of Ras mutant cells, higher than the level of wild-type cell division in homogeneous wild-type models. Neighbouring wild-type cells exert a powerful defensive mechanism, even in the presence of a higher mutational burden from the most aggressive isoform of the Ras family, Kras.

We found that although injury repair is coordinated by various growth factors and receptors, the EGFR–Ras signalling pathway emerges as key for increasing the number of dividing wild-type cells to suppress the expansion of Ras mutant cells. In the absence of injury, the constitutive loss of p21 also leads to a selective increase in the fraction of dividing wild-type cells in uninjured mosaic skin, recapitulating the responses to injury and highlighting the sufficiency of an increased number of wild-type cell divisions as a protective mechanism against the expansion of Ras mutant cells. Thus, genetic and environmental mechanisms promote a competitive advantage for wild-type cells, whereas Ras mutant cells are insensitive. Our data suggest that the *Hras^{G12V}* mutation renders cells insensitive to pro-proliferative stimuli mediated by the EGFR–Ras pathway during injury repair, in part because *Hras^{G12V}* cells already exhibit high levels of ERK1/2 activation, which promotes cell proliferation. On the basis of this observation, we propose that the fold change in Ras pathway activation before and after injury contributes to the selective capacity of wild-type cells to respond to pro-proliferative stimuli during injury repair that is already maximized in *Hras^{G12V/+}* cells (Extended Data Fig. 10e).

Our findings have broad implications, given that as well as in uninjured skin, Ras mutant cells have competitive advantages over wild-type cells in other uninjured tissues such as intestinal crypts and blood[46,47]. Recent studies in a single-layer epithelium in vitro and in vivo have shown that *Ras^{G12V}* mutant cells are apically extruded when surrounded by wild-type epithelial cells[48–50]. In the stratified epithelium of skin epidermis, our studies and others have shown that Ras mutant cells expand to outcompete wild-type neighbours and integrate in healthy tissue, suggesting a different mode of cell competition compared with the systems above[18–20]. We reconcile the above evidence by noting that different tissues preserve and maintain their specific architecture through distinct cell behaviours. Endogenous behaviours for maintaining monolayer epithelia are proliferation and extrusion, whereas multilayer epithelia rely on proliferation and delamination–differentiation. Thus, in a stratified epithelium, differentiation is analogous to extrusion in monolayer epithelia. Consistent with this reasoning, our evidence suggests that apoptosis, an ectopic behaviour in the adult skin epithelium, is not involved in the competition between wild-type and *Hras^{G12V/+}* cells.

Traditional therapeutic approaches used for cancer treatment involve suppressing the proliferation of both mutant and wild-type cells. Although these approaches restrain tumour expansion, they also impair the opportunity for the tissue to deploy natural defences, such as the selective promotion of wild-type cell proliferation. The next step towards an effective therapeutic treatment would be to determine how to promote the proliferative advantage bestowed on wild-type cells in the injury environment or in the pro-proliferative p21-null state. Our data suggest that in precancerous states and to prevent tumour relapse, EGFR activation—such as through EGF treatment—might provide a competitive advantage to wild-type cells in the presence of neighbours expressing the constitutively active form of a Ras oncogene. Future studies could investigate how the competitive advantage of wild-type cells may be leveraged in the setting of field cancerization with more complex mutant clones, which would represent a powerful treatment approach for field therapy in precancerous conditions such as actinic keratosis, the precursor of cSCC.

Collectively, this work provides a way forward for future research and clinical applications to focus on the mechanisms that empower wild-type cells in the competition with mutated neighbouring cells.

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

## Methods

### Mice

We used an outbred mouse strain (CD1) that is less susceptible to generating papilloma or squamous cell carcinomas than more inbred mouse strains used in other studies[11,12]. Mice were generated by interbreeding mice carrying the following alleles: *Krt14*-CreER[51] and FR-Hras[G12V/+] (ref. 52), constitutive p21 (Cdkn1a) loss of function[38] (JAX stock no. 016565), LSL-tdTomato[53] (JAX stock no. 007909), *Krt14*-H2B–GFP[23], *Krt14*-rtTA[54] (JAX stock no. 008099), TRE-EGFR-DN[55] (JAX stock no. 010575) and LSL-*Kras*[G12D/+] (ref. 56). Mice expressing *Krt14*-H2B–GFP were used to track epithelial cells with the two-photon microscope. The tdTomato reporter line was used to visualize CreER-driven recombination upon tamoxifen injection. CreER/*LoxP* lines are well documented to exhibit a certain degree of tamoxifen-independent (leaky) Cre activity over time. To account for the leakiness of the system in our experiments and to correctly interpret the data, we always compared wild-type mosaic models to Ras-mosaic models as well as tracked the same cells in the same animals over time. Mice from experimental and control groups were randomly selected for either sex. No blinding was done. All procedures involving animal subjects were performed under the approval of the Institutional Animal Care and Use Committee (IACUC) of the Yale School of Medicine. The mice were sacrificed if tumours reached 1 cm$^3$ (not allowed by IACUC) or if they presented signs of distress or weight loss. The tumour size limit was not exceeded in any of the experiments.

### Tamoxifen induction and drug treatment of mice

To induce CreER-driven recombination, mice were administered a single dose of 100 μg (mosaic induction) or 2 mg (maximal induction) tamoxifen (Sigma T5648-5G in corn oil) at postnatal day 19 by intraperitoneal injection (this time is designated day 0 for experiments). To induce rtTA-driven induction of EGFR-DN, mice were administered 2% of doxycycline (Sigma D9891) and 2% sucrose (Sigma S9378) in drinking water. All time courses began 6 days after tamoxifen injection. Gefitinib (ZD1839-Selleckchem) was resuspended in water with 0.5% (w/v) methylcellulose and 0.2% (v/v) Tween-80 (vehicle) and was administered orally (200 mg kg$^{-1}$ body weight) starting 2 days before wound induction until 14 days PWI (mice were not treated at day 7 and day 8 PWI).

### Injury induction

At postnatal day 21, mice were anaesthetized by intraperitoneal injection of a ketamine and xylazine mix (100 mg kg$^{-1}$ and 10 mg kg$^{-1}$, respectively in phosphate-buffered saline). Once the anaesthetized mouse did not physically respond to a noxious stimulus, a punch biopsy was performed using a 4-mm-diameter punch biopsy tool (Integra Miltex Standard Biopsy Punches). The punch biopsy tool was used to make a circular full-thickness injury on the dorsal side of a mouse ear or back skin. The injury did not penetrate the entire ear but remained above the cartilage. The skin epithelium in the mouse ear was chosen in this study for its accessibility to two-photon imaging and revisits over time. For recovery from the wound procedure Meloxicam (Metacam Loxicom) was administered via subcutaneous injection (0.3 mg kg$^{-1}$).

### Lentiviral production and in utero injection

Large-scale production and concentration of lentivirus expressing CreER (LV-CreER) was performed as previously described[57,58]. Detailed descriptions of in utero-guided lentiviral transduction can be found elsewhere[57,59]. To induce LV-CreER-mediated recombination, a maximal dose of tamoxifen (2 mg) was intraperitoneal injected at postnatal day 19.

### In vivo imaging

Mice were anaesthetized by intraperitoneal injection of ketamine and xylazine cocktail mix (100 mg kg$^{-1}$ and 10 mg kg$^{-1}$, respectively in phosphate-buffered saline) and then anaesthesia was maintained throughout the course of the experiment with the delivery of vaporized isoflurane by a nose cone as previously described[60]. Image stacks were acquired with a LaVision TriM Scope II (LaVision Biotec) laser scanning microscope equipped with a tunable Two-photon Chameleon Vision II (Coherent) Ti:Sapphire laser. To acquire serial optical sections, a laser beam 940 nm was focused through a 20× water immersion lens (NA 1.0; Zeiss) or a 25× water immersion lens (NA 1.1; Nikon) and scanned with a pixel size of $0.49 \times 0.49$ μm$^2$ or $0.43 \times 0.43$ μm$^2$ at 800 Hz. *Z*-stacks were acquired in 2–3 μm steps to image a total depth of 90–900 μm of the tissue. ImSpector v7.5.2 (LaVision Biotec) was used for 3D image acquisition. To visualize large areas, 5–12 tiles of optical fields were imaged using a motorized stage to automatically acquire sequential fields of view as previously described[60]. Mice were imaged at different time points after tamoxifen treatment and injury induction as indicated. To revisit the same area of the skin epidermis, organizational clusters of hair follicles and vasculature were used as landmarks.

### Whole-mount, OCT section and epidermal preparation immunostaining and imaging

To prepare whole mounts of mouse ear, skin was separated from connective tissue using forceps and incubated in 4% paraformaldehyde (PFA) at 37 °C for 4 h. To prepare the epidermal preparation, the skin separated from the connective tissue was incubated with Dispase (5 mg ml$^{-1}$, Roche) for 10 min at 37 °C and then the intact sheet of epidermis was gently peeled away from the dermis. The epidermal preparations were fixed in 4% PFA for 45 min at room temperature. Immunostaining was performed on whole mounts and epidermal preparations blocked with 5% normal goat serum/1% BSA/2% Triton X-100/PBS at room temperature. For tissue-section analysis, mouse ears were fixed in 4% PFA for 1 h at room temperature and then embedded in optimal cutting temperature (OCT; Tissue Tek). Frozen OCT blocks were sectioned at 10 μm. The skin preparations were incubated with primary antibodies (active caspase-3 (AF835-R&D Systems) 1:300, p-histone H3 (06-570-Millipore) 1:300, keratin-6A (905701-BioLegend) 1:500, p21 (also known as Cdkn1a) (ab188224-Abcam) 1:50; p-p44/42 MAPK (Erk1/2) (4370-Cell Signaling) 1:300, Alexa Fluor 647 Phalloidin (A22287-Thermofisher) 1:200 and keratin-10 (03-GP-K10-ARP) 1:200 diluted in blocking buffer (~66 h at 37 °C for whole mounts and 4 °C overnight for epidermal preparations and OCT sections). Secondary antibodies (A-21071-Invitrogen goat anti-rabbit IgG (H + L) secondary antibody, Alexa Fluor 633, A-21105-Invitrogen goat anti-guinea pig IgG (H + L) highly cross-adsorbed secondary antibody, Alexa Fluor 633, A-21206, donkey anti-rabbit IgG (H + L) highly cross-adsorbed, Alexa Fluor 488 and A-10042 donkey anti-rabbit IgG (H+L) highly cross-adsorbed secondary antibody, Alexa Fluor 568) were diluted 1:300 in blocking buffer and applied to skin sections (~66 h at 37 °C for whole mounts or 1 h at room temperature for epidermal preparations and OCT sections). DAPI was added to label nuclei. Image stacks of whole-mount, epidermal preparation and OCT section immunostaining were acquired with the two-photon microscope described above with an additional tunable two-photon Chameleon Discovery (Coherent) Ti:Sapphire laser. To acquire serial optical sections, laser wavelengths of 800 nm, 880 nm (Vision II) and 1080 nm (Discovery) were focused through a 20× water immersion lens (NA 1.0; Zeiss) or a 25× water immersion lens (NA 1.1; Nikon) and scanned with a pixel size of $0.49 \times 0.49$ μm$^2$ or $0.43 \times 0.43$ μm$^2$ at 800 Hz. Image stacks of epidermal preparation immunolabelled for p-ERK1/2 and phalloidin were acquired with confocal microscope Zeiss LSM 980, objective Zeiss 20× (0.8 NA Dry) and scanned with a pixel size of $0.20 \times 0.20$ μm$^2$ with Software ZEN (blue edition). *Z*-stacks were acquired in 1-μm (whole mounts) or 2-μm (epidermal preparations and OCT sections) steps to image total depth of the samples.

### Senescence-associated β-galactosidase activity

Senescence was measured in epidermal preparation of uninjured and injured (day 3 PWI) mouse ear skin and in mouse pancreas and kidney,

as positive controls, with a Beta Galactosidase Assay Kit from Abcam (ab287846). This kit uses the fluorogenic fluorescein digalactoside galactosidase substrate, which, upon cleavage by β-galactosidase, generates a fluorescent product that can be measured with an ELISA plate reader. In brief, ~10 mg of the tissues described above was lysed with protein lysis buffer (included in the kit) and then the kit guidelines were followed to prepare samples, positive controls provided by the kit and the standard curve. The fluorescence in each sample was quantified with a GloMax Plate reader (Promega) at 475 nm excitation and 500–550 nm emission at two different time points (30 min apart). β-Gal levels in each sample were calculated using a β-galactosidase standard curve.

## Two-photon image analysis

Raw two-photon image stacks were analysed in ImageJ (NIH Image, 1.53c) or IMARIS (v. 9.9.1, Oxford Instruments). ImageJ was used to draw the boundaries between tdTomato$^+$ and tdTomato$^-$ areas in the basal stem cell layers of the skin epidermis and to measure the percentage of coverage of these areas. The average of the percentage of coverage of tdTomato$^+$ cells of three areas (294 μm$^2$ each) taken at different distances from the wound and randomly in the uninjured condition were calculated for each mouse at every time point. Then, the percentage increase of tdTomato$^+$ area over time was represented in the graphs. To be reproducible in the measurements, the areas quantified in the injured ears were not localized close to the wound edge because of the increased thickness of the epithelium in that region that prevents an accurate isolation of the basal stem cell layer due to reduced image resolution. tdTomato$^+$ and tdTomato$^-$ regions for each uninjured or injured ear were revisited and quantified over time.

We measured the number of events (mitotic figures, nuclear fragmentation events, or cells positive for p-histone H3, keratin-10 and active caspase-3 immunostaining) per unit of surface (1 μm$^2$) of tdTomato$^+$ or tdTomato$^-$ areas and then multiplied that value by the total surface (294 μm$^2$) to compare wild-type and mutant cell populations. ~900 μm$^2$ were analysed for each uninjured or injured mouse ear. We measured the number of cells positive for p-ERK1/2 per unit of surface (1 μm$^2$) of tdTomato$^+$ or tdTomato$^-$ areas and then multiplied that value by the total surface (150 μm$^2$) to compare wild-type and mutant cell populations. An area of approximately 600 μm$^2$ was analysed for each uninjured or injured mouse ear.

To quantify the thickness of the skin epithelium, we used IMARIS (v. 9.9.1). By utilizing the second harmonic collagen signal in the dermis and, when absent near the wound, the basal epithelial cell layer, we created a surface to approximate the basement membrane of the epithelium. When unable to visualize the basal–dermal interface near the wound due to excessive epithelia thickness, such as in the case of $Hras^{G12V/+}$ max or $Kras^{G12D/+}$ max, we created a surface along the bottom of the stack to ensure our thickness measurements were stringent in that and reflected only as much tissue as we could effectively image. From this surface we could extract the distance to the top of the cornified layer around the injured area of the epithelium and visually depict the tissue thickness with an intensity heat map. Using MatLab (v. R2018a) we could extract individual pixel intensity values from this heat map that directly correlate to tissue thickness and plot them based on relative distance from the wound edge.

To quantify the nuclear signal of p21 in the basal stem cell layer of the skin epidermis, we used IMARIS (v. 9.9.1). In IMARIS, surfaces were created using the $Krt14$-H2B–GFP signal to isolate a mask of the p21 signal within the basal epidermal stem cell nuclei. A maximum intensity projection of this mask was used to quantify the mean fluorescence intensity of the p21 signal within each individual nucleus. The mean fluorescence intensity of the p21 signal of each cell was normalized for background by subtracting the average p21 fluorescent intensity of mitotic cells within the field of view, as these cells would be negative for p21.

## scRNA-seq sample preparation and data analysis

After the sacrifice of wild-type, $Hras^{G12V/+}$ mosaic and $Hras^{G12V/+}$ max models at 6 days after tamoxifen injection (3 days PWI), the uninjured and injured ears of each mouse were cut in small pieces with a punch biopsy of 8 mm in diameter (the wound was kept at the centre the 8 mm biopsy to mostly isolate cells involved in injury repair). The ear epidermis was dissociated from the dermis and incubated in 0.25% Trypsin at 37 °C for 30 min. The epidermal preparation was placed in a 70 μm cell strainer, smashed with a piston and rinsed three times with PBS + 0.04% BSA. The flow-through was subsequently filtered through a 40 μm cell strainer, spun down and resuspended in 300–400 μl of PBS + 0.04% BSA. The viability of the cell suspension was determined using trypan blue. To prepare the single-cell library, the cellular suspensions were counted and diluted to a final concentration of 1,200 cells per μl in PBS/0.04% BSA and then loaded on a Chromium Controller to generate single-cell gel bead emulsions, targeting 3′. Single-cell 3′ RNA-seq libraries were generated according to the manufacturer's instructions (Chromium Single Cell 3′ Reagent v3 Chemistry Kit, 10X Genomics). Libraries were sequenced to an average depth of ~20,000 reads per cell on an Illumina Novaseq 6000 system.

Single-cell data from each sample—that is, all wild-type and $Hras^{G12V/+}$ mosaic and $Hras^{G12V/+}$ max, uninjured and injured conditions (24 independent samples, Extended Data Figs. 4a, 6a and 7a) were first processed with SoupX[61] (https://github.com/constantAmateur/SoupX) to remove barcodes that most probably represent ambient RNA as opposed to whole cells, using the algorithm's automated method. The resulting matrix was then processed with the Seurat package[62] (v.3, https://satijalab.org/seurat/index.html), to retain genes or features that are detected in at least 3 cells and include cells for which at least 200 genes or features are detected. Additionally, cells expressing greater than 12.5% of mitochondrial transcripts were filtered out as possible dead or dying cells. According to Seurat's normal workflow, the data was log-normalized and scaled. Linear dimensionality reduction was carried out using principal component analysis and the first 15 principal components were chosen for the downstream analysis steps. Clustering was carried out using Louvain algorithm, for resolution of 0.1. Non-linear dimensionality reduction was carried out by running UMAP. Next, the DoubletFinder[63] package (https://github.com/chris-mcginnis-ucsf/DoubletFinder) was used to get rid of barcodes that may represent possible doublets. The resulting cell matrix was normalized, scaled, and re-clustered, using the same steps mentioned above with the Seurat package. Three replicates for each group were then integrated using Seurat's canonical correlation analysis (CCA). The data were re-clustered as described previously[22], and the clusters were annotated using the top 5–10 highly expressed genes in each cluster. To further remove doublets that were not identified by the DoubletFinder algorithm and other contaminating cell populations, infundibulum cells, immune cells, and red blood cells (RBCs) were removed. In brief, using a chosen set of features, each cell in the Seurat object was assigned a score using the AddModuleScore function. The features used for each type of cells are listed below. Infundibulum-specific features: "Sostdc1", "Aqp3", "Ptn", "Fst", "Aldh3a1", "Postn", "Krt17","Alcam", "Apoe", "Sox9", "Vdr", "Nfib", "App", "Gsn", "Hmcn1", "Cspg4", "Efnb2", "Nedd4", "Adh7", "Defb6", "Mgst1", "Krt79". Immune cell-specific features: "H2-Aa", "H2-Ab1", "H2-Eb1", "Cd74", "Ptprc". RBC-specific features: "Hba-a1", "Hbb-bs", "Hba-a2", "Hbb-bt", "Bpgm", "Hebp2". For infundibulum signature, cells with scores higher than 0.4 were removed, while for immune and RBCs, cells with scores higher than 0.5 were removed. The following cell numbers (per sample and biological replicate (R)) passed QC and constitute the final dataset: HMU: 11,437 (R1), 9,864 (R2), 9,650 (R3); HMW: 9,590 (R1), 10,281 (R2), 8,922 (R3); WTMU: 11,856 (R1), 9,916 (R2), 8,416 (R3); WTMW: 9,181 (R1), 8,747 (R2), 8,320 (R3); HFU: 13,278 (R1), 10,711 (R2), 6,997 (R3); HFW: 9,613 (R1), 8,999 (R2), 7,825 (R3); WTFU: 13,379 (R1), 11,701 (R2), 6,433 (R3); WTFW: 9,649 (R1), 9,374 (R2), 6,350 (R3).

For interfollicular epidermis (IFE) keratinocytes, all mosaic samples were integrated and annotated using Scanpy (1.6-1.9)[64]. In brief, raw counts for the selected IFE cells were log-normalized and cell cycle stages were scored (sc.pp.score_genes_cell_cycle) based on a gene list from[65]. Biological replicate batches were corrected with bbknn (1.4.1)[66]. Next, the selected cells were scored for stress, immune and infundibulum related gene expression signatures (see the notebooks on GitHub: https://github.com/kasperlab/Gallini_et_al_2023_Nature), classified with a Gaussian mixture model (scikit-learn, 0.24.2[67]) and positive cells were filtered out. Similarly, classification was performed to annotate cells based on tdTomato and GFP expression. The remaining healthy IFE keratinocytes were then mapped (sc.tl.ingest) onto the characterized IFE differentiation trajectory and annotated accordingly based on the basal-suprabasal status and commitment that have been previously defined[22]. Finally, appropriate IFE groups were integrated with CCA as described above, using the Seurat package. Differential gene expression analysis was carried out between corresponding datasets and cell types.

For fibroblast and immune cell characterization, all mosaic and max datasets were analysed in Scanpy, with similar preprocessing and batch effect removal as for keratinocytes. Mixed cell populations were removed based on shared gene expression signatures. Wound-related cells were annotated based on Leiden clustering, sample type and wound-related gene expression signatures with additional confirmation by differential abundance testing using miloR (1.2.0)[68]. In brief, a Milo graph was built using the integrated dataset with the following parameters: $k = 20$, $d = 30$ and differential abundance was tested for the injury condition. Differential gene expression analysis was performed with scanpy.tl.rank_genes_groups function using Wilcoxon rank-sum test and Holm–Sidak correction for multiple comparisons. Gene set enrichment analysis was performed using the enrichr method in GSEAPY package (v0.12) with Gene Ontology biological process 2021 gene sets[69–73].

## Histology
Uninjured and injured sections of ear skin were fixed in 10% neutral formalin for 24 h and stored in 70% ethanol until paraffin embedding. Haematoxylin and eosin (H&E)-stained skin sections were used for histopathology analysis. Images were taken using an Olympus BX61 microscope equipped with a SPOT flex 15.2 64-Mp shifting pixel camera, 4×, 10× and 20× objectives, and SPOT v 5.2 software.

## Western blot analysis
Uninjured and injured ear skin were lysed with ice-cold RIPA buffer (Pierce) supplemented with cOmplete Protease Inhibitor Cocktail and PhosSTOP (Sigma) and centrifuged at maximum speed for 30 min to collect lysates. Protein concentration was measured with the BCA protein assay (Pierce). An aliquot of 20–30 µg of total protein per sample was loaded into 7.5 or 10% Mini-PROTEAN TGX Precast Protein Gels (Bio-Rad) and separated by SDS–PAGE. Proteins were transferred to PVDF membranes (BioRad). The following rabbit primary antibodies were used at the given concentrations; p-p44/42 MAPK (ERK1/2) (Thr202/Tyr204) (1:500, Cell Signaling 9101), p44/42 MAPK (ERK1/2) (1:500, Cell Signaling 4695), p-EGFR (Tyr1068) (1:100, Cell Signaling 2234), EGFR (1:100 Cell Signaling 4267; Extended Data Fig. 8b), EGFR (1:100, Cell Signaling 2232; Fig. 5b), p-AKT (Ser473) (1:200, Cell Signaling 4060), AKT (1:200, Cell Signaling 9262) and GAPDH (14C10) (1:500, Cell Signaling 2118). An anti-rabbit IgG HRP (1:500, Cell Signaling 7074) secondary antibody was used. Western blot analyses were performed on whole ear skin at 6 days after tamoxifen injection (3 days PWI).

## Statistics and reproducibility
Statistical analyses were performed using an unpaired, two-tailed Student's $t$-test for comparison between different groups of mice. Paired, two-tailed Student's $t$-tests were used for comparison between tdTomato$^+$ and tdTomato$^-$ populations in the same group of mice. Unpaired, ordinary one-way ANOVA was used for comparison between mice with three different genotypes. Statistical analyses were performed using Prism (v. 9) as indicated in the figure legends. Gene expression differences between different conditions from scRNA-seq data were performed with Student's $t$-test and Holm–Sidak correction for multiple comparisons. $P$ values of less than 0.05 were considered statistically significant (*$P < 0.05$, **$P < 0.005$, ***$P < 0.0005$, ****$P < 0.0001$). $n$ is defined for each experiment, and always indicates the number of mice used for each condition examined. Box plots within violin plots denote the 25th, 50th and 75th quartiles, with whiskers depicting the minima and maxima of the data, excluding outliers that are beyond 1.5× the interquartile range.

## Reporting summary
Further information on research design is available in the Nature Portfolio Reporting Summary linked to this article.

## Data availability
All data from this study are available from the authors on request. The raw data files of the scRNA-seq analyses reported in the manuscript have been uploaded to the Gene Expression Omnibus under accession GSE195892. Previously published scRNA-seq data that were used for reference are available under accession codes GSE152044, GSE129218 and GSE67602. Source data are provided with this paper.

## Code availability
Annotated and analysed sequencing data have been deposited at Zenodo (https://doi.org/10.5281/zenodo.7768108) and analysis notebooks have been uploaded to GitHub (https://github.com/kasperlab/Gallini_et_al_2023_Nature).

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

**Acknowledgements** The authors thank all members of the Greco and Kasper laboratories, B. Slobodan, S. Regot, K. Politi, B. Hu, L. Gonzalez and A. Andersen for critical feedback on the manuscript; M. Mandar for the LSL-*Kras*^G12D/+ mice; B. De Kumar and G. Wang for their support in scRNA-seq experiments and data analyses; and W. Damsky for the acquisition of H&E staining slides. Research reported in this publication was supported by an 55108527 HHMI Scholar award, the National Institute Of Arthritis and Musculoskeletal and Skin Diseases of the National Institutes of Health under award numbers R01AR063663 and R01AR072668, and the National Institute on Aging of the National Institutes of Health under award number DP1AG066590. The content is solely the responsibility of the authors and does not necessarily represent the official views of the National Institutes of Health. S.G. was supported by the Human Frontiers Science Program Long-term Postdoctoral fellowship (LT000051_2017-L). M.K. and K.A. were supported by grants from Cancerfonden (CAN 21 1821 Pj), Vetenskapsrådet (VR2018-02963), Karolinska Institutet (2-2111/2019; 2016-00206) and StratRegen (SFO at Karolinska Institutet).

**Author contributions** S.G. and V.G. designed experiments and wrote the manuscript. S.G. performed two-photon imaging, epidermal preparation immunostaining, mouse genetics, molecular assays, image analyses, designed the scRNA-seq experiment and assisted with analyses. D.G.G. assisted with IMARIS and MatLab analyses. N.-T.R., K.A. and M.K. performed scRNA-seq analyses and interpreted the data. K.A. and M.K. provided continuous feedback on the manuscript and assisted in manuscript writing. S.Y. assisted with IMARIS data analysis, LV-CreER in utero injection and provided feedback on the manuscript. C.M.-M. assisted with IMARIS data analysis, epidermal preparation immunostaining and provided feedback on the manuscript. E.L. and T.X. assisted with whole-mount tissue and OCT immunostaining. K.C.S. assisted with clinical diagnosis of histological images and provided feedback on the manuscript.

**Funding** Open access funding provided by Karolinska Institute.

**Competing interests** The authors declare no competing interests.

**Additional information**
**Correspondence and requests for materials** should be addressed to Maria Kasper or Valentina Greco.

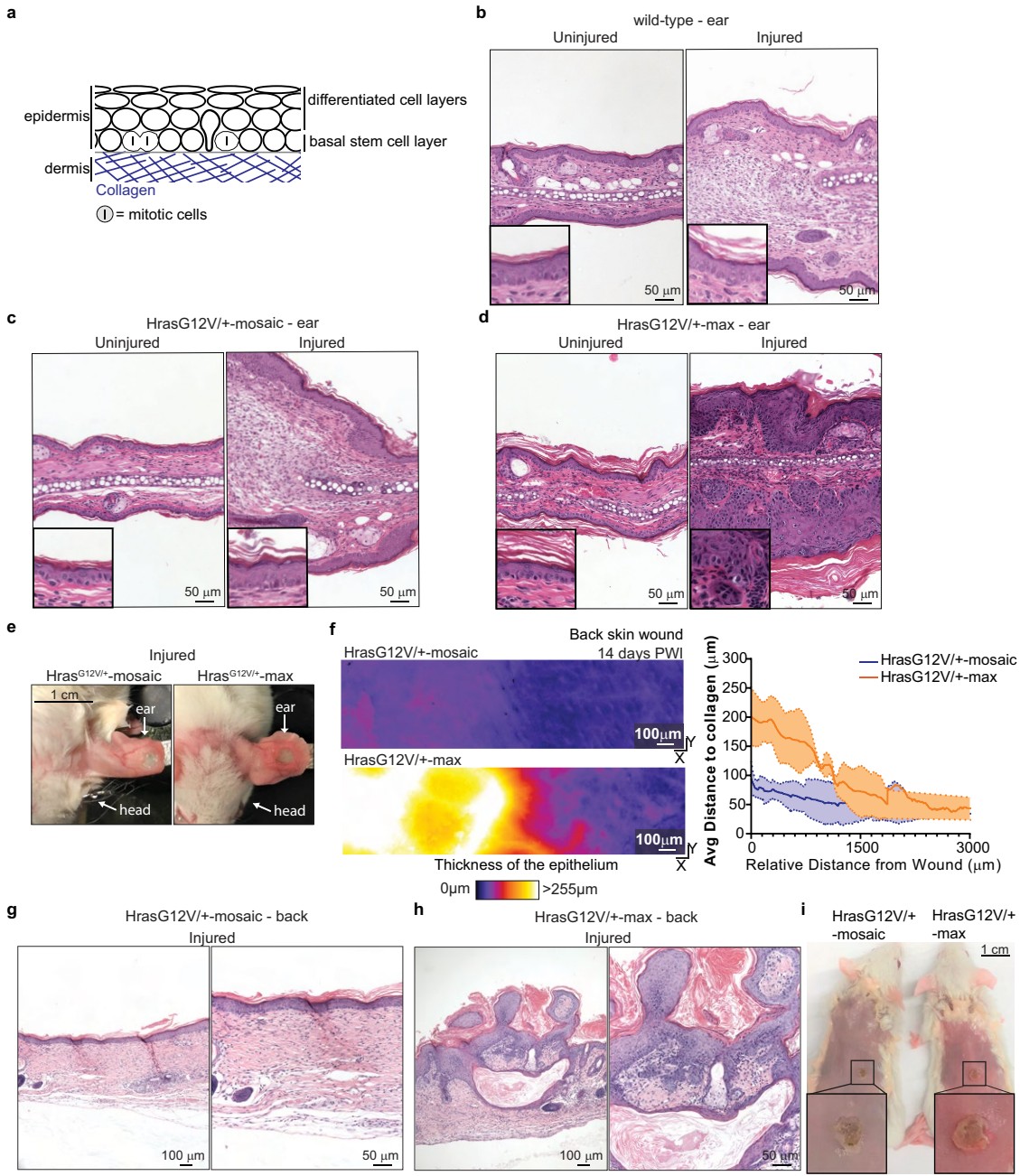

**Extended Data Fig. 1 | Injury induces tumors only in Hras$^{G12V/+}$-max models.**
**a)** Skin tissue architecture cartoon schematic. **b, c, d)** H&E staining of uninjured
and injured ear skin within 2 weeks PWI with magnified insets. n = 5 wild-type,
n = 5 Hras$^{G12V/+}$-max and n = 4 Hras$^{G12V/+}$-mosaic. b) Wild-type tissue exhibits
normal histology (*left*) and injured tissues (*right*) demonstrates mixed
inflammatory infiltrate and increased fibroblast number consistent with early
scar tissue. c) Hras$^{G12V/+}$-mosaic uninjured tissue (*left*) shows normal histology.
Injured ear (*right*) exhibits dermal fibrosis consistent with normal scar tissue.
d) Hras$^{G12V/+}$-max uninjured tissue (*left*) with normal epithelial thickness and
differentiation with mild orthokeratotic hyperkeratosis (stratum corneum
thickening with normal basket-weave appearance). In the Hras$^{G12V/+}$-max injured
ear (*right*) adjacent to the wound there is notable acanthosis and
hypergranulosis of the epithelium with compact orthokeratotic
hyperkeratosis. Cytologic atypia is present. **e)** Aberrant growth in the

injured-Hras$^{G12V/+}$-max model in contrast to the normal epithelium
in the injured-Hras$^{G12V/+}$-mosaic model (day-14 PWI). **f)** (*right*) Heat maps of
epithelial thickness in the top-down (*x-y*) view of representative two-photon
images adjacent to injury in the back skin of Hras$^{G12V/+}$-mosaic and Hras$^{G12V/+}$-max
identifying aberrant growth. Scale bar, 100 µm. (*left*) Quantification of average
epithelial thickness 14 days PWI at different distances from wound edge in
Hras$^{G12V/+}$-mosaic and Hras$^{G12V/+}$-max. n = 3 mice (mean±s.d.). **g, h)** H&E staining
of injured back skin within two weeks PWI. n = 3 mice. g) Hras$^{G12V/+}$-mosaic
injured back shows skin dermal fibrosis with vertically oriented blood vessels
consistent with scar. The overlying epidermis is normal thickness with
orthokeratosis. h) Hras$^{G12V/+}$-max injured back skin with dermal scar and the
overlying epidermis is characterized by hyperkeratosis and papillomatosis.
**i)** Aberrant growth around the wound of the Hras$^{G12V/+}$-max model in contrast
to the normal epithelium of the Hras$^{G12V/+}$-mosaic model.

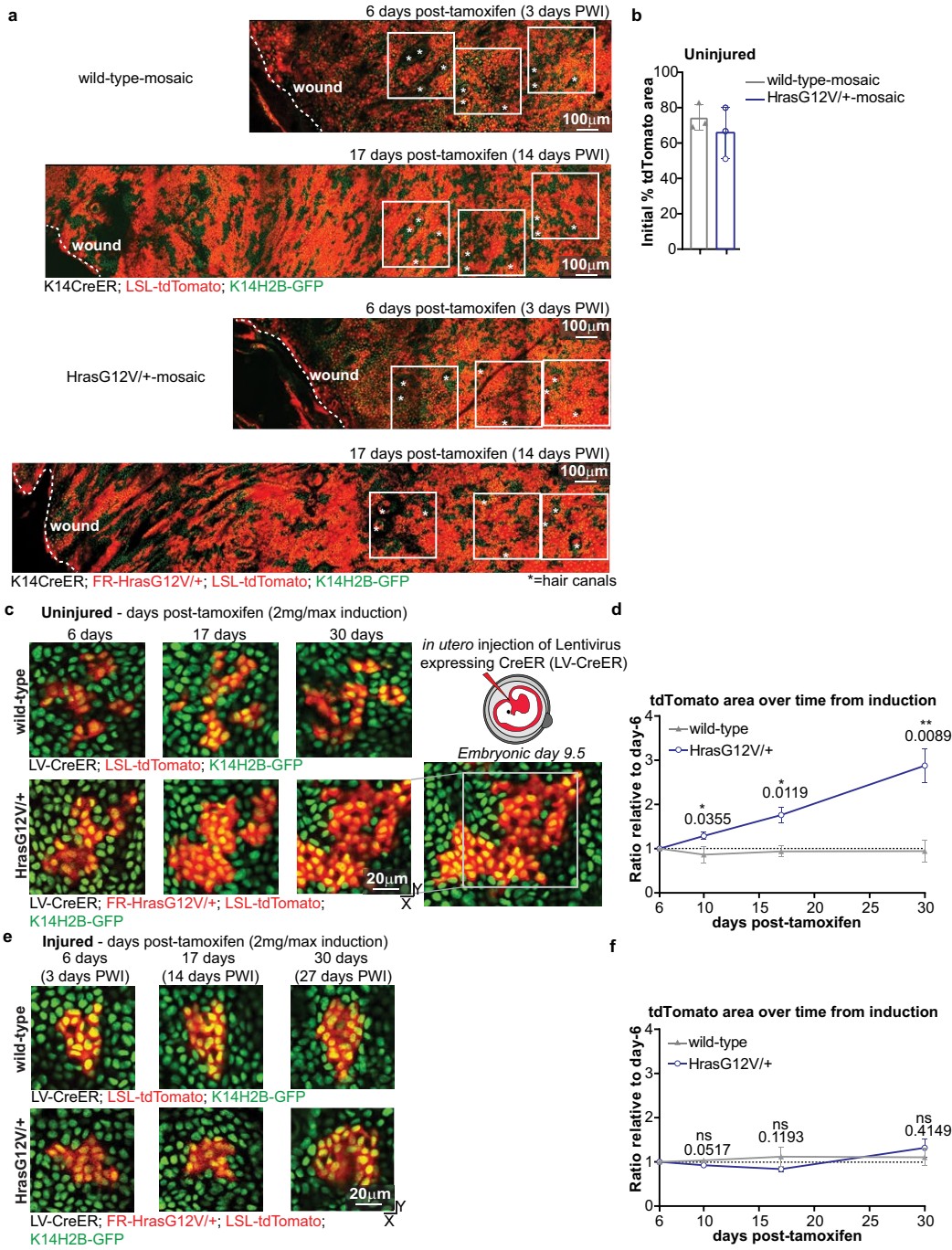

**Extended Data Fig. 2 | Injury suppresses Hras^G12V/+ cell expansion in mosaic skin with a low mutational burden. a)** Two-photon revisit images of the same area of the basal stem cell layer of the epidermis in wild-type-mosaic and Hras^G12V/+-mosaic at 3 and 14 days PWI. Dashed lines, wound edge; white squares, three representative regions tracked and quantified to measure the percent tdTomato+ area. The hair follicle pattern was used to revisit the exact same area of the skin (*=hair canals). Scale bar, 100 μm. **b)** Quantification of the initial percent tdTomato+ area 6 days post-tamoxifen injection in the uninjured condition. n = 3 mice. **c)** Representative two-photon revisit images of the same tdTomato+ clone revisited in the basal stem cell layer of the epidermis infected with lentiviral(LV)-CreER. **d)** Quantification of the relative ratio of the

tdTomato+ area at different time points compared to the initial clone size at 6 days post-tamoxifen injection. Relative ratio was used to consider the different initial sizes of the analyzed clones. The initial clone size was between 2 to ~50 cells. n = 3 mice. **e)** Representative two-photon revisit images of the same tdTomato+ clone in the basal stem cell layer of the epidermis infected with LV-CreER. **f)** Same quantification as in d). n = 3 mice. (d, f) Statistics: Unpaired, two-tailed *t-test* between wild-type and mutant mice at different time points in uninjured and injured conditions. Exact p-value reported on the figure. ns indicates not statistically significant. At least three independent tdTomato+ clones were analysed for each mouse. Data are represented as means and standard deviations. Scale bar, 20 μm.

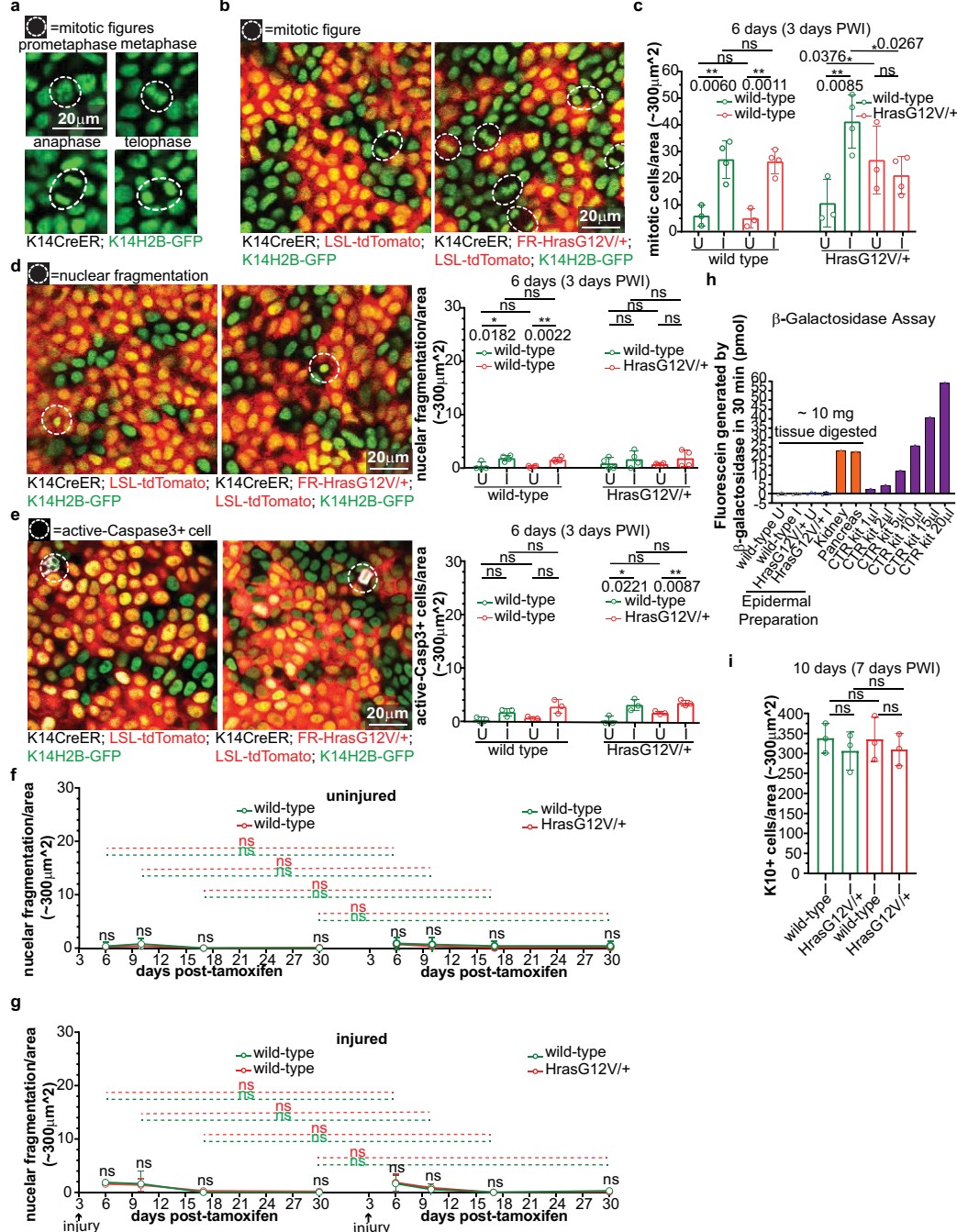

**Extended Data Fig. 3 | Selective increase of wild-type cell proliferation in the acute phase of injury-repair of mosaic skin. a, b)** Representative two-photon images of mitotic figures (white circles). **c)** Quantification of mitotic figures in tdTomato+ and tdTomato− areas in injured (I; n = 4 mice) and uninjured (U; n = 3 mice) wild-type and Hras[G12V/+]-mosaic. **d)** Representative two-photon images of apoptotic bodies (white circles) and quantification in tdTomato+ and tdTomato− areas in injured (I) and uninjured (U) wild-type and Hras[G12V/+]-mosaic. n = 4 mice. **e)** Representative two-photon images of the epidermal preparation immunofluorescence for active-Caspase-3 (white circle) and quantification of active-Caspase3+ cells in tdTomato+ and tdTomato− areas in injured (I) and uninjured (U) wild-type and Hras[G12V/+]-mosaic. n = 3 mice. **f)** Quantification of apoptotic bodies over time in tdTomato+ and tdTomato− areas in uninjured ears. Statistic between wild-type-mosaic and

Hras[G12V/+]-mosaic in tdTomato− or tdTomato+ areas are represented with green or red dotted lines. **g)** Same quantification as in f) in injured ears. (f, g) n = 4 mice. **h)** β-galactosidase activity assay of epidermal preparations of uninjured and injured wild-type and Hras[G12V/+]-mosaic and positive controls: pancreas, kidney and provided by the kit. n = 3 mice for epidermal preparations. **i)** Quantification of cells expressing Keratin10 in tdTomato+ and tdTomato− areas in injured (I) wild-type and Hras[G12V/+]-mosaic. n = 3. Statistics: Paired, two-tailed *t-test*, comparing tdTomato+ and tdTomato− areas in the same group of mice. Unpaired, two-tailed *t-test* comparing tdTomato+ and tdTomato− populations in different groups of mice. Exact p-value reported on the figure. ns indicates not statistically significant. At least three independent areas of approximately 300 μm² were analysed for each mouse (*see* Methods). Data are represented as means and standard deviations. Scale bar, 20 μm.

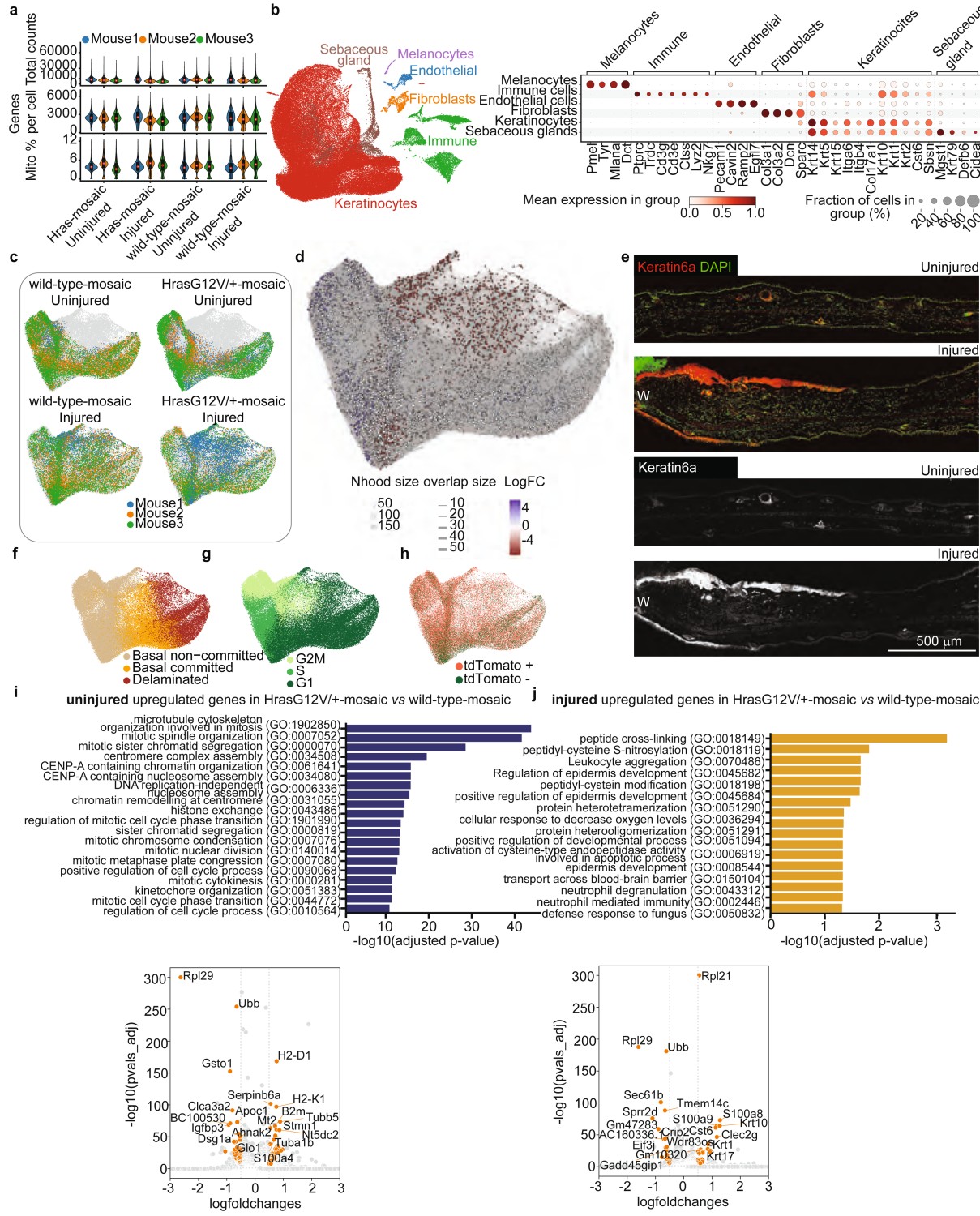

**Extended Data Fig. 4** | See next page for caption.

**Extended Data Fig. 4 | Characterization of scRNA-seq datasets of wild-type-mosaic and Hras$^{G12V/+}$-mosaic cells in uninjured or injured condition.**
**a)** Violin plots showing quality control metrics for each sample. All the box plots within violin plots denote the 25%, 50% and 75% quartiles with whiskers depicting the minima and maxima of the data, excluding outliers that are beyond 1.5x interquartile range. **b)** UMAP displaying the main cell populations of the integrated dataset (*left*), and dot-plot showing characteristic marker-gene expression (*right*). **c)** UMAPs showing the distribution of interfollicular epidermal (IFE) keratinocytes of the different conditions, coloured according to biological replicates. Grey cells denote all keratinocytes. **d)** Abstracted graph of neighbourhoods superimposed on IFE UMAP, showing differential abundance testing results from *MiloR*[68]. Node sizes represent the size of the neighbourhood and edges indicate number of cells common between neighbourhoods. Neighbourhoods displaying significant differential abundance are coloured according to the log-foldchange of the differential abundance testing. **e)** Micrographs showing Keratin6a protein expression in uninjured and injured ear tissue. W – wound, scale bar, 500 μm. n = 3 mice. **f, g, h)** UMAPs showing cell classification into basal non-committed, basal committed, and delaminated populations (f), cell cycle phase (g) and cell classification based on tdTomato expression (h). **i, j)** Bar plots showing top GO terms (*upper*) for differentially up-regulated genes comparing Hras$^{G12V/+}$ and wild-type-mosaic cells in uninjured (i) or injured (j) condition. Respective differential gene expression analysis is shown on volcano plots (*lower*) with log2 fold-changes on x-axis and -log10(adjusted p-values) on y-axis. Genes were considered differentially expressed (orange dots) when they were expressed in >25% of cells in each of the compared biological replicate, had absolute log-foldchange > 0.5 and adjusted p-value < 0.05 (Wilcoxon rank-sum test with Benjamini-Hochberg correction). a-d, f-j) n = 12 independently sequenced mice (3 mice per condition and genotype).

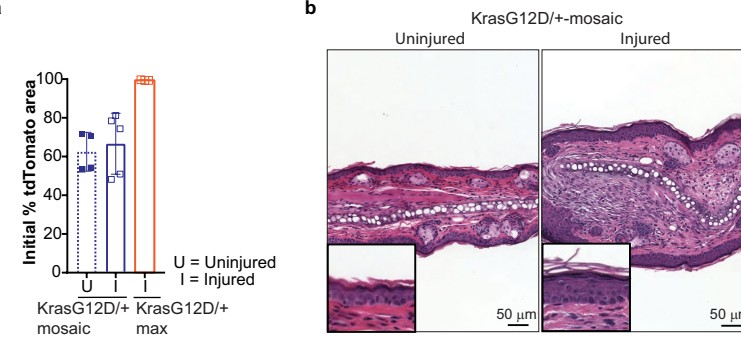

**a**

**b** KrasG12D/+-mosaic

**c** KrasG12D/+-max

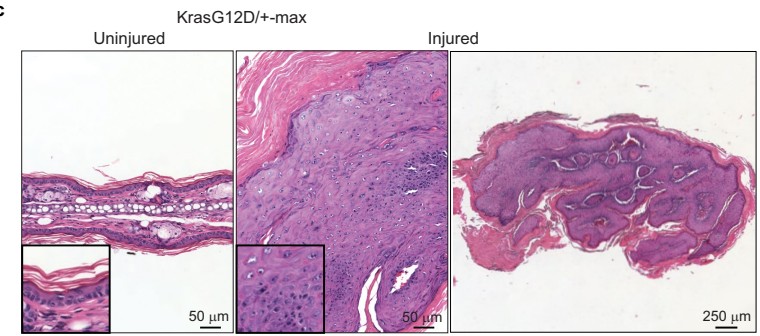

**d**

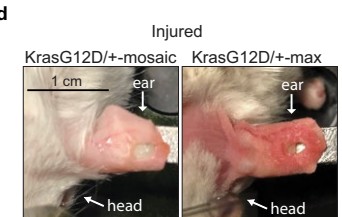

**Extended Data Fig. 5 | Kras[G12D/+]-max models display rapid oncogenic growth after injury. a)** Quantification of the initial percent tdTomato+ area in the first revisit of Kras[G12D/+]-mosaic in uninjured (n = 4 mice) and injured (n = 5 mice) conditions and Kras[G12D/+]-max in injured (n = 3 mice) condition. At least three independent areas of approximately 300 μm² were analysed for each mouse (Methods). Data are represented as means and standard deviations. **b, c)** Histopathologic examination via H&E staining of uninjured and injured ear skin within 2 weeks PWI. n = 4 Kras[G12D/+]-mosaic, n = 5 Kras[G12D/+]-max. b) Kras[G12D/+]-mosaic (*left*) in uninjured ear skin with normal epithelium, dermis and cartilage and injured ear (*right*) showing mixed inflammatory cell infiltrate and increased number of fibroblasts consistent with early scar tissue. c) Kras[G12D/+]-max uninjured ear *(right)* showing normal epithelial thickness and architecture. This is in comparison to the injured ear (*left*) showing significant expansion of the epithelial layer with hypergranulosis, focal parakeratosis alternative to the compact hyperkeratin. Focal cytologic atypia is present. Scale bars indicated on the figure. Magnified insets of the epidermis in the lower left corner of each image. **d)** Macroscopic image of the aberrant growth around the wound of the Kras[G12D/+]-max model in contrast to the normal epithelium of the Kras[G12V/+]-mosaic model.

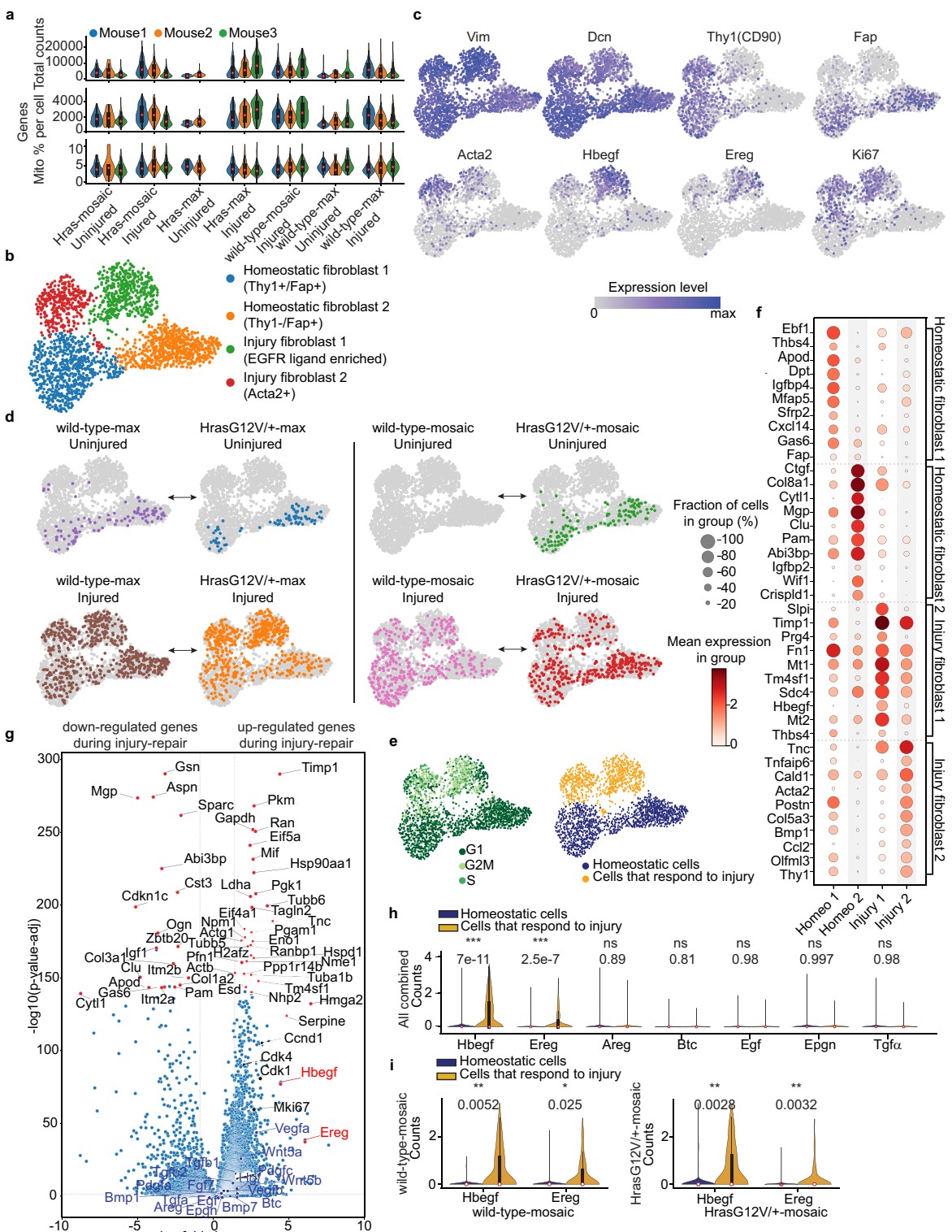

**Extended Data Fig. 6** | See next page for caption.

**Extended Data Fig. 6 | scRNA-seq fibroblast subcluster analysis reveals increased EGFR-ligand expression upon injury. a)** Violin plots showing quality control metrics for each individual sample. **b)** Unsupervised clustering of sampled fibroblasts[74]. **c)** Marker gene expression in fibroblast subclusters overlaid on UMAP. **d)** Distribution of cells from different mouse models among fibroblast clusters (n = 3 mice for each condition). Grey dots denote all cells. Hras$^{G12V/+}$ and wild-type models have a similar distribution of cells in four clusters, depending on the tamoxifen treatment and the presence or not of the wound. **e)** UMAP showing cell-cycle classification (*left*) and injury response status (*right*). **f)** Dot-plot showing marker gene expression for the fibroblast subclusters. **g)** Volcano plot of differential gene expression profiles between fibroblasts from homeostatic and injured conditions showing the magnitude on the x-axis (Log2 fold change) and significance on the y-axis (-Log10 adjusted p-value, Wilcoxon rank-sum test with Benjamini-Hochberg correction). Red dots mark the 50 highest differentially expressed genes. Red names highlight the highest differentially expressed growth factors that affect epithelial cell behaviors. Blue dots with blue names represent other growth factors that take part in injury-repair and black dots with black names indicate genes involved in cell proliferation. **h)** Violin plots of EGFR ligands that affect epithelial cell behaviors[75]. Analysis is based on all mouse models combined. **i)** Violin plots that compare homeostatic and injury-responsive cells for the expression of EGFR ligands with a significantly different expression in (h). n = 3 mice per group. (a, h, i) Internal box plots denote the 25%, 50% and 75% quartiles with whiskers depicting the minima and maxima of the data, excluding outliers that are beyond 1.5x interquartile range. Statistics: two-tailed *t-test* comparing the averages of biological replicates according to conditions. (a-h) n = 24 independently sequenced mice (3 mice per condition and genotype, note that some samples did not contain fibroblasts; 'mosaic samples' are the same as in Extended Data Fig. 4).

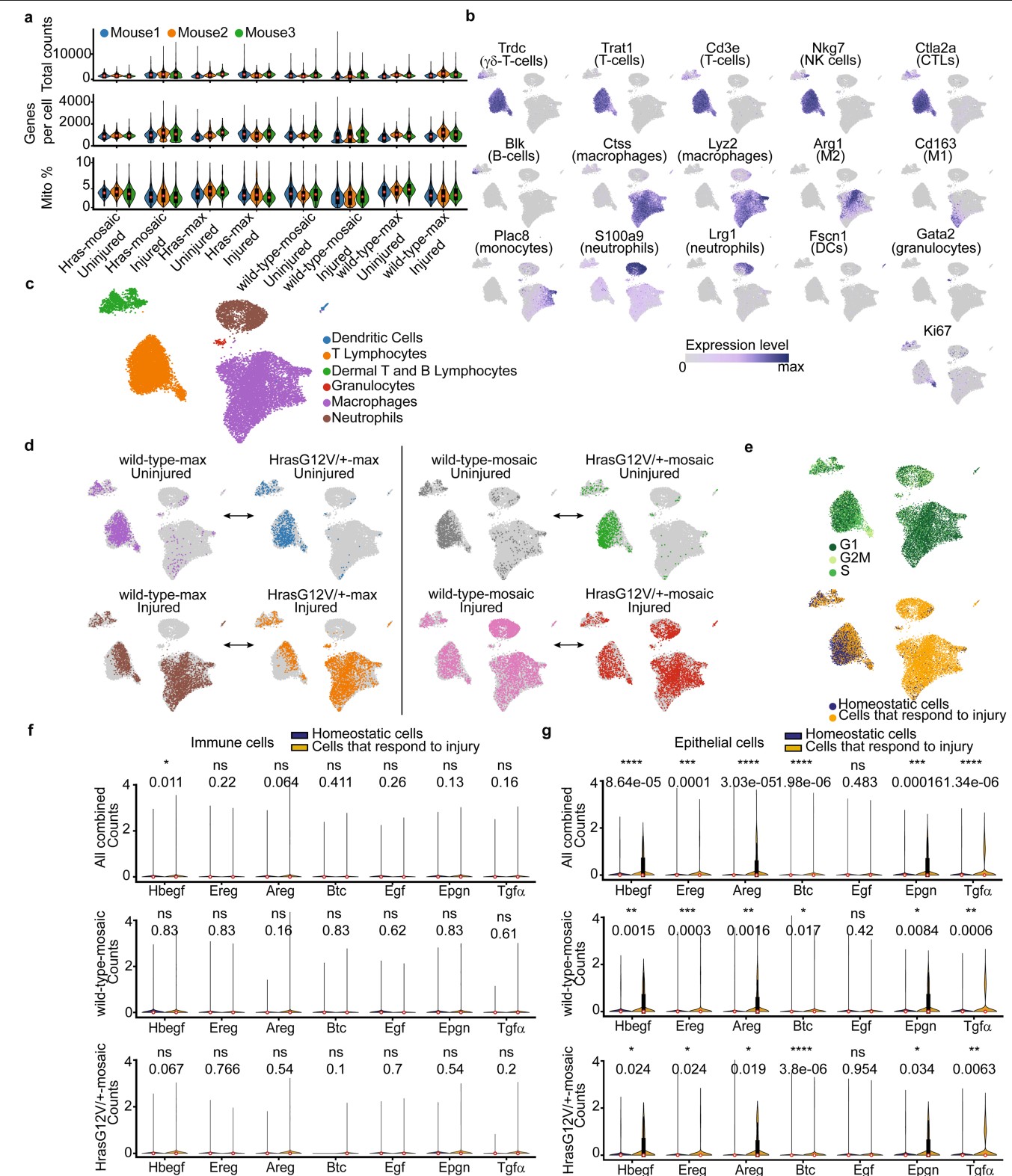

**Extended Data Fig. 7 | scRNA-seq immune cell subcluster analysis shows limited EGFR-ligand expression upon injury. a)** Violin plots showing quality control metrics for each individual sample. **b)** Annotation of cluster identities based on marker gene expression[76]. **c)** Unsupervised clustering of sampled immune cells (*see* Methods)[76]. **d)** Distribution of cells from different mouse models among the identified immune cell clusters (n = 3 mice for each condition). **e)** UMAP representations of cell cycle classification (*top*) and injury status of the dataset *(bottom)*. **f)** Violin plots showing immune cell EGFR ligand expression for all datasets combined (n = 24 mice, *top*) and comparison between wild-type-mosaic and Hras[G12V/+]-mosaic datasets (n = 6 mice per group, *bottom*). **g)** Violin plots showing epidermal (i.e.: IFE) EGFR ligand expression for all datasets combined (n = 12 mice, *top*) and comparison between wild-type-mosaic and Hras[G12V/+]-mosaic datasets (n = 6 mice per group, *bottom*). (a, f, g) Internal box plots denote the 25%, 50% and 75% quartiles with whiskers depicting the minima and maxima of the data, excluding outliers that are beyond 1.5x interquartile range. (f, g) Statistics: two-tailed *t-test* comparing the means of biological replicates according to conditions. (a-f) n = 24 independently sequenced mice (3 mice per condition and genotype; 'mosaic samples' are the same as in Extended Data Figs. 4, 6).

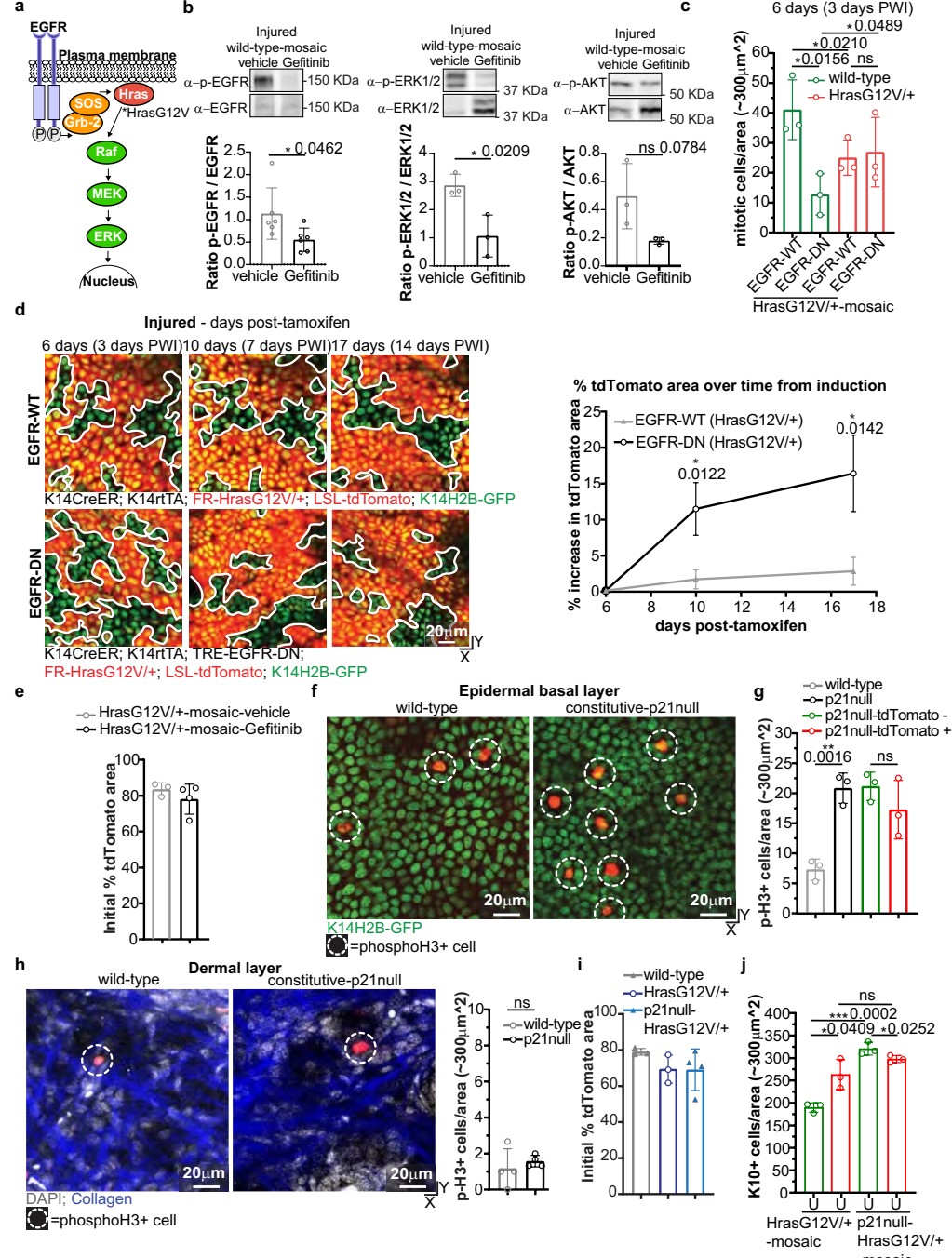

**Extended Data Fig. 8 | EGFR/Ras pathway is required to selectively increase wild-type cell proliferation after injury. a)** Schematic representation of the EGFR signaling pathway. Hras[G12V] is constitutively active and therefore less dependent on upstream activation of EGFR. **b)** Western blot analysis of p-EGFR(phospho-Tyr1068) normalized on total-EGFR (n = 6 mice) and of p-ERK1/2(phospho-Thr202/Tyr204) and p-AKT(phospho-Ser473) normalized on total-ERK1/2 and total-AKT (n = 3 mice). Unpaired, two-tailed *t-test*. **c)** Mitotic figure quantification in tdTomato+ and tdTomato− areas in Hras[G12V/+]-mosaic without or with EGFR-Dominant Negative (DN) expression (EGFR-WT or EGFR-DN). n = 3 mice. Unpaired or Paired two-tailed *t-test* for comparison between different or the same groups of mice. **d)** Revisit images of the same area of the basal stem cell layer. (*left*) The increase of tdTomato+ area. n = 3 mice. Unpaired, two-tailed *t-test*. **e)** Initial percent tdTomato+ area in the first revisit of injured-Hras[G12V/+]-mosaic treated with vehicle (n = 3 mice) or Gefitinib (n = 4 mice). **f)** Epidermal preparation immunofluorescence for phospho-Histone3 in wild-type (*left*) and constitutive-p21[null] (*right*) at postnatal-day-25. **g)** Phospho-Histone3+ cell quantification in wild-type, constitutive-p21[null] with or without LSL-tdTomato at postnatal-day-25 (n = 3 mice). Unpaired, two-tailed *t-test*. **h)** Whole mount immunofluorescence for phospho-Histone3 in wild-type (*left*) and constitutive-p21[null] (*right*) at postnatal-day-25. Phospho-Histone3+ dermal cell quantification (n = 4 mice). Unpaired, two-tailed *t-test*. **i)** Initial percent tdTomato+ area in the first revisit of uninjured wild-type-mosaic (n = 3 mice), Hras[G12V]-mosaic (n = 3 mice) and constitutive-p21[null]-Hras[G12V/+]-mosaic (n = 4 mice). **j)** Keratin10+ cell quantification in tdTomato+ and tdTomato− areas in Hras[G12V/+]-mosaic and constitutive-p21[null]-Hras[G12V/+]-mosaic in uninjured(U) ears (n = 3 mice). Unpaired or Paired two-tailed *t-test* for comparison between different or the same groups of mice. Exact p-values reported on the figure. ns indicates not statistically significant. At least three independent areas of approximately 300 µm² were analysed for each mouse (Methods). Data are represented as means and standard deviations. Scale bar, 20 µm.

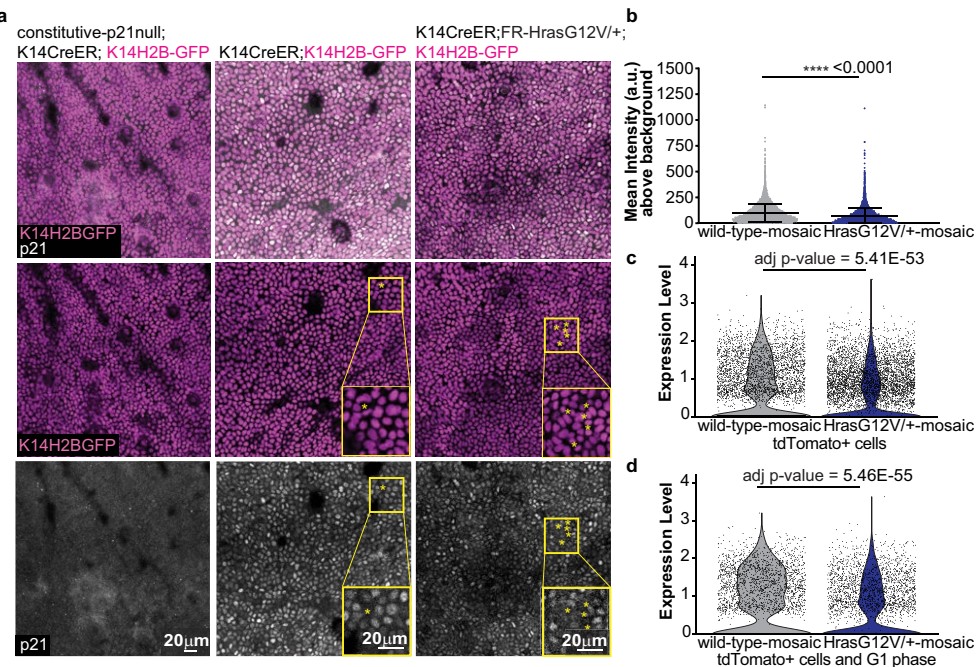

**Extended Data Fig. 9 | p21 expression is significantly reduced in Hras$^{G12V/+}$-mosaic models. a)** Representative two-photon images of the epidermal preparation immunofluorescence for p21 (white) in constitutive-p21$^{null}$ (*left*), wild-type-mosaic (*center*) and Hras$^{G12V/+}$-mosaic (*right*). The basal cells expressing K14H2B-GFP are marked in magenta. Representative images show both the overlay (*top*) and separate channels for K14H2B-GFP expression (*middle*) and p21 immunolabeling (*bottom*). *=mitotic cells negative for p21 (background signal). **b)** Quantification of p21 immunolabeling in the nucleus of the basal cells in wild-type-mosaic and Hras$^{G12V/+}$-mosaic (n = 3 mice). Average of approximately 3,000 cells were analysed for each mouse. Unpaired, two-tailed *t-test* for the nuclear intensity of p21. Data are represented as means and standard deviations. **c)** p21 level quantification in the basal cells by using scRNA-seq datasets of wild-type-mosaic and Hras$^{G12V/+}$-mosaic with the selection of only tdTomato+ cells. **d)** Same quantification as in c) but with the selection of non-cycling and tdTomato+ cells. (c, d) n = 3 mice. Statistics on scRNA-seq analysis of p21 expression is based on the non-parametric Wilcoxon rank sum test performed by the Seurat package[62]. Adjusted p-value is based on Bonferroni correction using all features in the dataset.

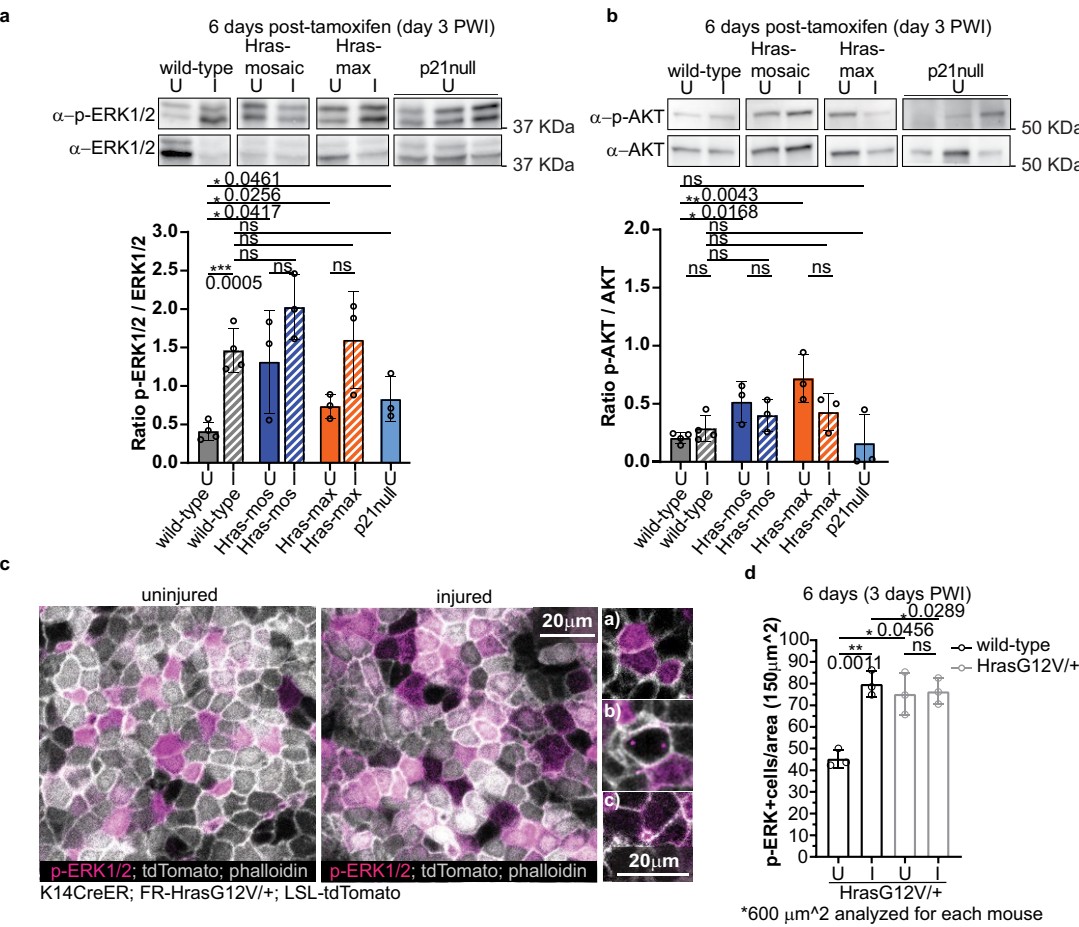

**a** 6 days post-tamoxifen (day 3 PWI)

**b** 6 days post-tamoxifen (day 3 PWI)

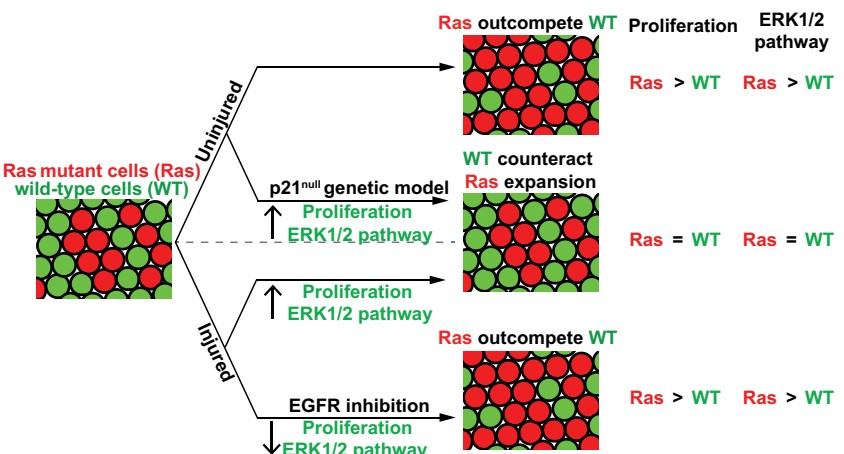

**c**

uninjured     injured

20μm

a)

b)

c)

20μm

p-ERK1/2; tdTomato; phalloidin     p-ERK1/2; tdTomato; phalloidin
K14CreER; FR-HrasG12V/+; LSL-tdTomato

**d** 6 days (3 days PWI)

**e**

**Injury-repair increases wild-type cell divisions to prevent abnormal oncogenic Ras growth**

Ras mutant cells (Ras)
wild-type cells (WT)

Uninjured

Injured

p21^null genetic model
Proliferation
ERK1/2 pathway

Proliferation
ERK1/2 pathway

EGFR inhibition
Proliferation
ERK1/2 pathway

Ras outcompete WT     Proliferation     ERK1/2 pathway

Ras > WT     Ras > WT

WT counteract
Ras expansion

Ras = WT     Ras = WT

Ras outcompete WT

Ras > WT     Ras > WT

**Extended Data Fig. 10** | See next page for caption.

**Extended Data Fig. 10 | Injury-repair or p21$^{loss}$ specifically increases phospho-ERK1/2 in wild-type cells. a)** Western blot analysis of p-ERK1/2(phospho-Thr202/Tyr204) normalized on total-ERK1/2. **b)** Western blot analysis of p-AKT(phospho-Ser473) normalized on total-AKT. (a, b) wild-type (n = 4 mice), Hras$^{G12V/+}$-mosaic (n = 3 mice), Hras$^{G12V/+}$-max (n=3 mice) and constitutive-p21$^{null}$ (n = 3 mice). Unpaired, two-tailed *t-test*. Data are represented as means and standard deviations. **c)** Confocal representative images of the epidermal preparation immunofluorescence for p-ERK1/2 and phalloidin in Hras$^{G12V/+}$-mosaic. (*left*) Insets of p-ERK1/2+ cells: **a)** cells in interphase **b)** mitotic cell and **c)** footprint of a cell that is departing from the basal layer. **d)** Quantification of p-ERK1/2+ cells in tdTomato+ and tdTomato− areas in Hras$^{G12V/+}$-mosaic in injured/3 days PWI(I) and uninjured(U) ears. At least four independent 150 μm$^2$ areas were analysed for each mouse. (c, d) n = 3 mice. Unpaired or Paired two-tailed *t-test* for comparison between different or the same groups of mice. ns indicates not statistically significant. Data are represented as means and standard deviations. Scale bar, 20 μm. (a, b, c, d) Exact p-values reported on the figure. **e)** Final model. In the uninjured mosaic skin epidermis, Ras cells integrate and expand, outcompeting wild-type neighbors. During injury-repair of mosaic skin, the competitive advantage of Ras cells is suppressed, and oncogenic growths do not develop. EGFR signaling pathway is crucial for this selective increase of wild-type cell divisions that prevents Ras cell expansion during injury-repair. Inducing proliferation, via constitutive p21 loss, mimics the injury condition in uninjured skin, counteracting the competitive advantage of Ras cells. Our data support a model whereby injury-repair and p21 loss increase the activity of ERK1/2, a Ras downstream pathway that controls cell proliferation. This leads to comparable ERK1/2 levels between wild-type and Ras cells, resulting in an increase of dividing wild-type cells that effectively prevent Ras mutant cell expansion.

# Reporting Summary

## Statistics

For all statistical analyses, confirm that the following items are present in the figure legend, table legend, main text, or Methods section.

| n/a | Confirmed | |
|---|---|---|
| ☐ | ☒ | The exact sample size (*n*) for each experimental group/condition, given as a discrete number and unit of measurement |
| ☐ | ☒ | A statement on whether measurements were taken from distinct samples or whether the same sample was measured repeatedly |
| ☐ | ☒ | The statistical test(s) used AND whether they are one- or two-sided<br>*Only common tests should be described solely by name; describe more complex techniques in the Methods section.* |
| ☒ | ☐ | A description of all covariates tested |
| ☐ | ☒ | A description of any assumptions or corrections, such as tests of normality and adjustment for multiple comparisons |
| ☐ | ☒ | A full description of the statistical parameters including central tendency (e.g. means) or other basic estimates (e.g. regression coefficient) AND variation (e.g. standard deviation) or associated estimates of uncertainty (e.g. confidence intervals) |
| ☐ | ☒ | For null hypothesis testing, the test statistic (e.g. *F*, *t*, *r*) with confidence intervals, effect sizes, degrees of freedom and *P* value noted<br>*Give P values as exact values whenever suitable.* |
| ☒ | ☐ | For Bayesian analysis, information on the choice of priors and Markov chain Monte Carlo settings |
| ☒ | ☐ | For hierarchical and complex designs, identification of the appropriate level for tests and full reporting of outcomes |
| ☒ | ☐ | Estimates of effect sizes (e.g. Cohen's *d*, Pearson's *r*), indicating how they were calculated |

*Our web collection on statistics for biologists contains articles on many of the points above.*

## Software and code

Policy information about availability of computer code

| | |
|---|---|
| Data collection | Image stacks were acquired with a LaVision TriM Scope II (LaVision Biotec) laser scanning microscope equipped with a tunable Two-photon Chameleon Vision II (Coherent) Ti:Sapphire laser and Two-photon Chameleon Discovery (Coherent) Ti:Sapphire laser. ImSpector v7.5.2 (LaVision Biotec) for 3D image acquisition.<br>Image stacks of epidermal preparation immunolabeled against phosphoERK1/2 and phalloidin were acquired with confocal microscope Zeiss LSM 980 with Software ZEN (blue edition).<br>To prepare the single-cell library, the cellular suspensions were counted and diluted to a final concentration of 1200 cells/µl in PBS/0.04% BSA and then loaded on a Chromium Controller to generate single-cell gel bead emulsions, targeting 3'. Single-cell 3' RNA-seq libraries were generated according to the manufacturer's instructions (Chromium Single Cell 3' Reagent v3 Chemistry Kit, 10X Genomics, Inc.). Libraries were sequenced to an average depth of ?20.000 reads per cell on an Illumina Novaseq 6000 system. |
| Data analysis | Statistical analyses were performed using Prism (version 9) as indicated in the figure legends. Raw two-photon image stacks were analyzed in ImageJ (1.53c, NIH Image) or IMARIS (version 9.9.1, Oxford Instruments). To quantify the thickness of the skin epithelium, we used IMARIS and MatLab (version R2018a). To quantify the mean fluorescence intensity of the p21 signal within each individual nucleus and background we used IMARIS. Single cell RNA-sequencing analysis was performed in Scanpy (1.6-1.9), SoupX (https://github.com/constantAmateur/SoupX), Seurat (3 - https://satijalab.org/seurat/index.html), bbknn (1.4.1), DoubletFinder package (https://github.com/chris-mcginnis-ucsf/DoubletFinder), scikit-learn (0.24.2), miloR (1.2.0) and GSEAPY package (v 0.12). Further details are provided in methods and analysis notebooks are uploaded to github: (https://github.com/kasperlab/Gallini_et_al_2023_Nature). |

For manuscripts utilizing custom algorithms or software that are central to the research but not yet described in published literature, software must be made available to editors and reviewers. We strongly encourage code deposition in a community repository (e.g. GitHub). See the Nature Portfolio guidelines for submitting code & software for further information.

## Data

Policy information about [availability of data](availability of data)

All manuscripts must include a [data availability statement](data availability statement). This statement should provide the following information, where applicable:

- Accession codes, unique identifiers, or web links for publicly available datasets
- A description of any restrictions on data availability
- For clinical datasets or third party data, please ensure that the statement adheres to our [policy](policy)

All data from this study are available from the authors on request. The raw data files of the scRNA-seq analyses reported in the manuscript are uploaded to Gene Expression Omnibus (GSE195892). Previously published scRNA-seq data that were used for reference are available under accession codes GSE152044, GSE129218, and GSE67602. Annotated and analysed sequencing data have been deposited in Zenodo: https://doi.org/10.5281/zenodo.7768108 and analysis notebooks are uploaded to GitHub: https://github.com/kasperlab/Gallini_et_al_2023_Nature.

## Human research participants

Policy information about [studies involving human research participants and Sex and Gender in Research.](studies involving human research participants and Sex and Gender in Research.)

| | |
|---|---|
| Reporting on sex and gender | N/A |
| Population characteristics | N/A |
| Recruitment | N/A |
| Ethics oversight | N/A |

Note that full information on the approval of the study protocol must also be provided in the manuscript.

# Field-specific reporting

Please select the one below that is the best fit for your research. If you are not sure, read the appropriate sections before making your selection.

☒ Life sciences   ☐ Behavioural & social sciences   ☐ Ecological, evolutionary & environmental sciences

For a reference copy of the document with all sections, see [nature.com/documents/nr-reporting-summary-flat.pdf](nature.com/documents/nr-reporting-summary-flat.pdf)

# Life sciences study design

All studies must disclose on these points even when the disclosure is negative.

| | |
|---|---|
| Sample size | No statistical method were used to pre-determined sample sizes but our sample sizes are similar to those reported in previous publications (Rompolas, Mesa et al., 2016, Brown, Pineda et al., 2017, Mesa, Kawaguchu, Cockburn et al., 2018, Joost et al., 2016, Joost, Annusver et al., 2020, Cockburn Annusver et al., 2022). |
| Data exclusions | No data were excluded from the analysis |
| Replication | All the experiments were performed in at least 3 biologically independent replicates. All replicates reported in the manuscript are biological replicates. All attempts at replication of the results were successful. |
| Randomization | Mouse models were chosen based on correct genotypes: K14CreER/FR-HrasG12V/tdTomato/K14H2B-GFP; K14CreER/tdTomato/K14H2B-GFP; K14CreER/LSL-KrasG12D/tdTomato/K14H2B-GFP; K14CreER/FR-HrasG12V/p21null/tdTomato/K14H2B-GFP; K14CreER; K14rtTA; FR-HrasG12V/+; LSL-tdTomato; TRE-EGFR-DN; K14H2B-GFP.<br>Mice were always induced with tamoxifen (Sigma T5648-5G ) at 19 days after birth and wound was induce at 21 days after birth. Each experiment contained animals from at least 2 different litters. |
| Blinding | The investigator were not blinded . Blinding was not possible as the same investigator processed the animals and analyzed the data. |

# Reporting for specific materials, systems and methods

We require information from authors about some types of materials, experimental systems and methods used in many studies. Here, indicate whether each material, system or method listed is relevant to your study. If you are not sure if a list item applies to your research, read the appropriate section before selecting a response.

## Materials & experimental systems

| n/a | Involved in the study |
|-----|----------------------|
| ☐ | ☒ Antibodies |
| ☒ | ☐ Eukaryotic cell lines |
| ☒ | ☐ Palaeontology and archaeology |
| ☐ | ☒ Animals and other organisms |
| ☒ | ☐ Clinical data |
| ☒ | ☐ Dual use research of concern |

## Methods

| n/a | Involved in the study |
|-----|----------------------|
| ☒ | ☐ ChIP-seq |
| ☒ | ☐ Flow cytometry |
| ☒ | ☐ MRI-based neuroimaging |

## Antibodies

**Antibodies used**

For Immunofluorescence the following antibodies were used:
Primary antibodies: active-Caspase3 (AF835-R&D Systems, Inc.) 1:300, phospho-Histone3 (06-570-Millipore) 1:300, Keratin6A (905701-BioLegend) 1:500, p21/Cdkn1a (ab188224-Abcam) 1:50; Phospho-p44/42 MAPK(Erk1/2) (4310-Cell Signaling) 1:300, Alexa Fluor™ 647 Phalloidin (A22287-Thermofisher) 1:200 and Keratin10 (03-GP-K10-ARP) 1:200 diluted in blocking buffer. Secondary antibodies: Alexa Fluor 633, A-21071-Invitrogen-Goat anti-Rabbit IgG (H+L) Secondary Antibody, Alexa Fluor 633, A-21105-Invitrogen-Goat anti-Guinea Pig IgG (H+L) Highly Cross-Adsorbed Secondary Antibody, Alexa Fluor™ 488, A-21206, Donkey anti-Rabbit IgG (H+L) Highly Cross-Adsorbed, and Alexa Fluor 568, A10042-Donkey anti-Rabbit IgG (H+L) Highly Cross-Adsorbed Secondary Antibody) were diluted 1:300.
For Western Blot the following rabbit primary antibodies were used at the given concentrations: phospho-p44/42 MAPK (ERK1/2) (Thr202/Tyr204) (1:500, Cell Signaling-9101), p44/42 MAPK (ERK1/2) (1:500, Cell Signaling-4695), phosphoEGFR (Tyr1068) (1:100, Cell Signaling-2234), EGFR (1:100 Cell Signaling-4267 - Extended Data Figure 8b), EGFR (1:100, Cell Signaling-2232 - Figure 5b), phosphoAKT (Ser473) (1:200, Cell Signaling-4060), AKT (1:200, Cell Signaling-9262) and GAPDH (14C10) (1:500 - Cell Signaling-2118). An anti-rabbit IgG HRP (1:500, Cell Signaling-7074) secondary was used.

**Validation**

Antibodies validation information can be found on manufacturers' website. We used protocols and recommendations of the manufacturer on validated species.

Primary Antibodies:
1. active-Caspase3 (AF835-R&D Systems, Inc - https://www.rndsystems.com/products/human-mouse-active-caspase-3-antibody_af835)
2. phospho-Histone3 (06-570-Millipore - https://www.emdmillipore.com/US/en/product/Anti-phospho-Histone-H3-Ser10-Antibody-Mitosis-Marker,MM_NF-06-570)
3. Keratin6A (905701-BioLegend - https://www.biolegend.com/en-us/search-results/purified-anti-mouse-keratin-6a-antibody-11459)
4. p21/Cdkn1a (ab188224-Abcam - https://www.abcam.com/products/primary-antibodies/p21-antibody-epr18021-ab188224.html)
5. Phospho-p44/42 MAPK (Erk1/2) (Thr202/Tyr2 - 04) (D13.14.4E) XP® Rabbit mAb #4370 - https://www.cellsignal.com/products/primary-antibodies/phospho-p44-42-mapk-erk1-2-thr202-tyr204-d13-14-4e-xp-rabbit-mab/4370
6. Keratin10 (03-GP-K10-ARPv - https://www.arp1.com/anti-keratin-k10-polyclonal-antibody-serum-03-gp-pp2.htmlv / Cockburn, Annusver et al., 2022)
7. phospho-p44/42 MAPK (ERK1/2) (Thr202/Tyr204) (Cell Signaling-9101 - https://www.cellsignal.com/products/primary-antibodies/phospho-p44-42-mapk-erk1-2-thr202-tyr204-antibody/9101)
8. p44/42 MAPK (ERK1/2) (Cell Signaling-4695 - https://www.cellsignal.com/products/primary-antibodies/p44-42-mapk-erk1-2-137f5-rabbit-mab/4695)
9. phosphoEGFR (Tyr1068) (Cell Signaling-2234 - https://www.cellsignal.com/products/primary-antibodies/phospho-egf-receptor-tyr1068-antibody/2234)
10. EGFR (Cell Signaling-2232 - https://www.cellsignal.com/products/primary-antibodies/egf-receptor-antibody/2232)
11. EGFR (Cell Signaling-4267 -https://www.cellsignal.com/products/primary-antibodies/egf-receptor-d38b1-xp-rabbit-mab/4267)
12. phosphoAKT (Ser473) (Cell Signaling-4060 - https://www.cellsignal.com/products/primary-antibodies/phospho-akt-ser473-d9e-xp-rabbit-mab/4060)
13. AKT (Cell Signaling-9262 - https://www.cellsignal.com/products/primary-antibodies/akt-antibody/9272)
14. GAPDH (14C10) (Cell Signalling-2118 - https://www.cellsignal.com/products/primary-antibodies/gapdh-14c10-rabbit-mab/2118)

Secondary Antibodies:
1. Alexa Fluor 633, A-21105-Invitrogen-Goat anti-Guinea Pig IgG (H+L) Highly Cross-Adsorbed Secondary Antibody - https://www.thermofisher.com/antibody/product/Goat-anti-Guinea-Pig-IgG-H-L-Highly-Cross-Adsorbed-Secondary-Antibody-Polyclonal/A-21105
2. Alexa Fluor 633, A-21071-Invitrogen-Goat anti-Rabbit IgG (H+L) Secondary Antibody (https://www.thermofisher.com/antibody/product/Goat-anti-Rabbit-IgG-H-L-Highly-Cross-Adsorbed-Secondary-Antibody-Polyclonal/A-21071)
3. Alexa Fluor™ 647 Phalloidin (A22287-Thermofisher - https://www.thermofisher.com/order/catalog/product/A22287v)
4. Alexa Fluor™ 488, A-21206, Donkey anti-Rabbit IgG (H+L) Highly Cross-Adsorbed (https://www.thermofisher.com/antibody/product/Donkey-anti-Rabbit-IgG-H-L-Highly-Cross-Adsorbed-Secondary-Antibody-Polyclonal/A-21206)
5. Alexa Fluor 568, A10042-Donkey anti-Rabbit IgG (H+L) Highly Cross-Adsorbed Secondary Antibody (https://www.thermofisher.com/antibody/product/Donkey-anti-Rabbit-IgG-H-L-Highly-Cross-Adsorbed-Secondary-Antibody-Polyclonal/A10042)
6. An anti-rabbit IgG HRP (1:1000, Cell Signaling-7074 - https://www.cellsignal.com/products/secondary-antibodies/anti-rabbit-igg-hrp-linked-antibody/7074v)

# Animals and other research organisms

| | |
|---|---|
| Laboratory animals | An outbred mouse strain background (CD1) from post-natal day 19 to post-natal day 49 was used in this study. K14CreER (Vasioukhin et al., 1999) and "Flox and Replace"-HrasG12V/+ (Chen et al., 2009), constitutive p21 (Cdkn1a) loss of function (Deng et al., 1995; JAX stock #016565), LoxSTOPLox-tdTomato (Madisen et al., 2010; JAX stock #007909), K14H2B-GFP (Tumbar et al.,2004), K14rtTA (Nguyen et al., 2006; (JAX stock #008099)), TRE-EGFR-DN (Roh et al., 2001; (JAX stock #010575)) and and LoxSTOPLox-KrasG12D/+ (Kackson et al., 2001). All the animals used were grown in mixed albino background (CD1) to allow two-photon imaging experiments. To induce genetic recombination CreER-driven all mice were subjected to an intraperitoneal tamoxifen injection at 19 days after birth. At post-natal day 21, mice were anaesthetized by intraperitoneal injection of ketamine and xylazine cocktail mix (100mg kg and 10mg kg, respectively in phosphate-buffered saline). Once the anesthetized mouse did not physically respond to a noxious stimulus, a punch biopsy was performed using a 4-mm-diameter punch biopsy tool on the dorsal side of a mouse ear or in back skin. For recovery from the wound procedure Meloxicam(Metacam® Loxicom®) was administered via subcutaneous injection (0.3 mg Kg). For in vivo imaging, mice were anaesthetized by intraperitoneal injection of ketamine and xylazine cocktail mix (100mg kg and 10mg kg, respectively in phosphate-buffered saline) and then anesthesia was maintained throughout the course of the experiment with the delivery of vaporized isoflurane by a nose cone. K14CreER (Vasioukhin et al., 1999) and "Flox and Replace"-HrasG12V/+ (Chen et al., 2009), constitutive p21 (Cdkn1a) loss of function (Deng et al., 1995; JAX stock #016565), LoxSTOPLox-tdTomato (Madisen et al., 2010; JAX stock #007909), K14H2B-GFP (Tumbar et al.,2004), K14rtTA (Nguyen et al., 2006; (JAX stock #008099)), TRE-EGFR-DN (Roh et al., 2001; (JAX stock #010575)) and and LoxSTOPLox-KrasG12D/+ (Kackson et al., 2001). All the animals used were grown in mixed albino background to allow two-photon imaging experiments. To induce genetic recombination CreER-driven all mice were subjected to an intraperitoneal tamoxifen injection (Sigma T5648-5G in corn oil) at 19 days after birth. To induce rtTA-driven induction of EGFR-DN, mice were administered 2% of Doxycycline (Sigma D9891) and 2% sucrose (Sigma S9378) in drinking water. All time courses began 6 days post-tamoxifen injection. Gefitinib (ZD1839-Selleckchem) was resuspended in water with 0.5% (w/v) methylcellulose and 0.2% (v/v) Tween-80 (vehicle) and was administered orally (200 mg/kg body weight) starting 2 days before wound induction until 14 days post-wound induction. At post-natal day 21, mice were anaesthetized by intraperitoneal injection of ketamine and xylazine cocktail mix (100mg kg and 10 mg kg, respectively in phosphate-buffered saline). Once the anesthetized mouse did not physically respond to a noxious stimulus, a punch biopsy was performed using a 4-mm-diameter punch biopsy tool  (Integra™ Miltex™ Standard Biopsy Punches) on the dorsal side of a mouse ear or in back skin. For in vivo imaging, mice were anaesthetized by intraperitoneal injection of ketamine and xylazine cocktail mix (100mg kg and 10mg kg, respectively in phosphate-buffered saline) and then anesthesia was maintained throughout the course of the experiment with the delivery of vaporized isoflurane by a nose cone.<br><br>Housing condition: five mice per cage<br>Dark/light cycle: light from 7 AM to 7 PM<br>Ambient temperature and humidity: 68-79°F and humidity 30-70%. |
| Wild animals | This study did not involve wild animals. |
| Reporting on sex | Sex-specific differences were minimized by including both male and female animals in the replicates. |
| Field-collected samples | The study did not involve samples collected from the field. |
| Ethics oversight | All procedures involving animal subjects were performed under the approval of the Institutional Animal Care and Use Committee (IACUC) of the Yale School of Medicine. The mice were sacrificed if tumor reached 1cm3, not allowed by IACUC, or if mice presented signs of distress or weight loss. The tumor size limit was not exceeded in any of the experiments. |

Note that full information on the approval of the study protocol must also be provided in the manuscript.

