## [Peer Review File · Nature]

Manuscript Title: Injury prevents Ras mutant cell expansion in mosaic skin

Redactions – unpublished data

Reviewer Comments & Author Rebuttals

Reviewer Reports on the Initial Version:

Referee #1 (Remarks to the Author):

The manuscript by Gallini et al., focuses on characterizing the cellular behavior of HRAS and KRAS mutant cells within the context of the skin epithelium and the competitive relationship between the mutant and WT cells in the context of uninjured and injured skin. In vivo imaging is a powerful and elegant approach to characterize cellular behavior, however, there are a number of concerns about the interpretation of the results that are described in detail below.

Specific comments

The major point that wildtype cells following injury outcompete mutant cells carrying the mutated version of Ras (both HRas and KRas) constitutes a conundrum. Most researchers who have worked with these oncogenes activated by various cre drivers observe that a major obstacle for these studies is the development of papillomas in and around the mouth area linked with constant skin abrasion/wounding (e.g. Lapouge et al., 2011, PNAS; van der Weyden et al., 2011, J Pathol) even when only small populations of cells are targeted (Page et al., 2013, Cell Stem Cell). Moreover, following wounding it is clear that cells expressing oncogenic Ras even from the hair follicle are rapidly mobilized to migrate into the epidermis and form papillomas (e.g. Page et al., 2013, Cell Stem Cell). It consequently appears counter intuitive based on the published literature that cells expressing oncogenic Ras should be outcompeted by WT, and the authors need to address this point experimentally.

The authors perform their experiments on mice, which are 3 weeks of age. A major assumption in the analysis performed by the authors is the ability to alter homeostatic cell behavior, yet at 3-weeks the ear has not reached its final size and they are consequently modelling the effects of the mutation not only in the context of tissue growth but also wounding. It remains unknown whether this impacts their analysis.

The authors explore how mutant and wild type HRAS cells compete with one another when present in two different proportions of cells in the epidermis. They are able to demonstrate in line with previous observations that aberrant tissue morphology is induced by injury in mouse models with a high burden of HRASG12V mutations in basal cells (~100%). This is not observed in a model with a lower recombination efficiency (~65%) referred to as the “mosaic” model. The results of these experiments are however difficult to interpret. The authors use tdTomato as a proxy for HRASG12V mutations. Given that the two loci are located at distinct genomic sites and will require two independent recombination events, this remains an assumption and should be determined. This means that Ras mutants might be tdTomato negative and a fraction of the tdTomato positive cells are likely not mutants for Ras. The authors need to determine the recombination efficiency at the single cell level. Without evidence that directly demonstrates that tdTomato + cells also contain the

HRas mutation, it is impossible to conclude “Collectively, these data show that HrasG12V/+ cells break homeostatic tissue architecture during injury-repair only when nearly all of the basal cells express HrasG12V/+. (Line 101-103).

The authors also argue that they study effects of different degrees of recombination (65 vs 100% according to tdTomato+ cells), but none of these regimens actually simulate what happens, when rare mutated cells with a Ras mutation are placed under homeostatic or injury conditions. Is there a certain threshold percentage, where mutant cells behave differently or is only a small fraction of Ras mutants that go on to behave differently e.g. by changing their environment?

The authors argue that “Intriguingly, HrasG12V/+ mosaic models behaved similarly to wild-type mosaic models (Fig. 1c, d” (Line 95-96). However, although the differences might not be significant, it is a tall order to argue that mosaic wt and mosaic Ras models behave similarly. There are obvious differences between the two types of samples.

In the competition assays between WT and Ras mutant cells (Figure 2), it is unclear why the authors observe such a steep increase in the tdTomato +ve area 10 days after induction in the uninjured area. After this initial burst the increase in area covered is not nearly as pronounced. Are there systemic effects from the wounding specifically affecting the tdTomato +ve cells? Moreover, the results presented for the WT cells in figure 2c are surprising, since they show a steady increase in the percentage of tomato positive cells over time in intact skin. This is inconsistent with an equipotent basal progenitor model originally published for the tail epidermis (Clayton et al., 2007, Nature) and subsequently adopted for the ear epithelium by the Jones (Doupe et al., 2010, Dev Cell) and Greco labs (Rompolas et al., 2016, Science). In an equipotent model the number of labeled cells should remain constant. This may be a direct consequence of the small area analyzed in the study and it would consequently be important to provide robustness by increasing the areas analyzed. According to the figure legend, one area was imaged for these quantifications, and adding the independent measurements will clearly assist in the interpretation of robustness and the evaluation of whether the statistical tests used are appropriate.

To explain the observed increase in competitive drive for the WT cells the authors performed analyzes of the frequency of mitosis observed in HRASG12V mutant cells vs their WT counterparts in intact and injured skin. Here there are a number of observations that are left unclear. Why is there only a transient increase in mitotic events in the HRASG12V/tdTomato population in the uninjured tissue? Also, why is there around 5 times as many proliferating WT cells in the uninjured skin in the animal with HRas mutation (3g vs h), and similarly many more proliferating WT cells in the injured condition when compared to the controls? The increase in competitiveness of wt cells is interesting and warrant further exploration as to the specific mechanisms inducing these changes, and is left completely un-explored. The authors argue that loss of p21 could be the mechanism providing the WT cells with their selective advantage during tissue regeneration, yet they provide no evidence for this.

Much more sophisticated analyses are required to address mechanistically the interplay between WT and mutant cells. What is the role of the underlying mesenchyme? Are there some kind of quorum sensing in the culture, where a certain percentage of mutant cells provides one effect

whereas a higher or lower fraction will induce different changes? These are all very pertinent questions that the authors never touch upon, which would be expected based on the title and abstract.

Referee #2 (Remarks to the Author):

Gallini and colleagues examined the fitness of epidermal stem cells with mutational RAS activation (HRAS or KRAS) and the impact of injury on this fitness. The use of both HRAS and KRAS models, with mutational activation at endogenous loci, helps substantiate their results (these models are elegant and powerful). A key feature of their experiments is that the impact of injury on the malignant progression of skin basal cells expressing activated HRAS-G12V or KRAS-G12D is assessed in a mosaic context where WT cells remain, contrasting with previous studies in homogeneous models. They obtained the opposite result from that expected by previous studies in homogeneous models – injury actually promoted the ability of WT cells to outcompete the mutant ones, and malignant growth was not observed. In contrast, when they induced HRAS activation in ~100% of cells, injury induced malignant growth. Thus, homeostatic architecture is only disrupted by injury when nearly all cells express mutant HRAS. They had previously shown that RAS mutations provided a competitive advantage to skin cells in mosaic contexts (recapitulated here), so it is interesting that injury would abrogate this advantage (with the mutant cells no different than WT).

They further showed that the competing WT cells exhibited enhanced mitotic activity post-injury, while injury did not increase mitoses in the RAS activated cells, which already had higher mitoses (but less than WT competitors in the injured context). In contrast, in uninjured tissue the HRAS (or KRAS) activated cells exhibited greater mitotic activity, consistent with their competitive advantage in uninjured skin.

They then showed that global loss of the CDK inhibitor p21 suppressed the competitive advantage of HRAS mutant cells in uninjured skin. It thus appears that losing p21 “levels the playing field” by selectively increasing proliferation of the WT cells, taking away the edge that HRAS mutant cells normally have. They further show that MAPK is activated by both p21 loss and by injury, as it is by HRAS mutation (as expected).

Data are supportive of their conclusions and are robust. Experiments have sufficient numbers of repetitions (minimally 3) and are properly analyzed statistically. The manuscript is well written and their points are conveyed clearly and fairly.

The discoveries in this manuscript are exciting and very novel, and could be fundamentally important, revealing a new mechanism whereby injury could be tumor suppressive. The authors are correct that their data support a different approach to cancer prevention: specifically boosting the fitness of WT cells, rather than suppressing all cells as with traditional chemo. Still, there are major concerns regarding the generality and physiological relevance of their discovery. In addition, these studies do not provide much mechanistic insight into how injury alters competitive dynamics.

Major concerns:

1) They are inducing recombination to generate activated HRAS in 60-80% of cells. Of course, oncogenesis begins with mutations to single cells. While this is harder to model, it would be important to determine the fitness effects of HRAS activation and the impact of injury under the more physiological context of a much lower fraction of oncogenically activated epidermal stem cells (e.g. activation of HRAS in 1-5% of cells, where most cells with recombinations will be surrounded by WT cells). Such a modification is particularly relevant given the many demonstrations that cell competition is dependent on ratios of winners and losers, with a dominance of winners being required to eliminate a loser (which can be a cell with an oncogenic mutation) (1-3).

2) These data derive from a single context of injury (ear punch) with two very similar models (activation of a RAS gene using similar GEMM), raising concerns about the general relevance. The demonstration that injury is tumor suppressive and might help eliminate pre-transformed cells would be very important, and thus they need to substantiate that injury induced elimination of initiated cells happens physiologically (either in humans or in mice). What is important is that this manuscript indicates that such elimination CAN happen in this particular mouse model. What they need to show is that it DOES happen in some natural context, or at least some additional contexts.

3) They show that WT competitors have greater mitotic activity than RAS activated cells in the skin specifically in the context of injury. What is not clear is why the WT cells do not outcompete the RAS activated cells in the injured skin, given the greater mitotic activity of the former. Some insight into this quandary should be provided.

4) It is not clear how far the suppression of cells with activated RAS extends from the injury (ear punch). From the images in Suppl Fig 2, the impact appears to be at least a couple of mm from the injury. Such an understanding should help inform a search for the mechanism, which appears to be likely mediated by diffusible factors. A quick test of whether these effects are systemic would be to analyze competition in the uninjured ear from the same mouse.

5) This work does not shed much light on the underlying mechanism for how injury promotes mitoses specifically in the WT cells, particularly when in competition with RAS activated cells. Have the authors analyzed gene expression in the WT cells with or without RAS competitors and with or without injury? Are there potential soluble mediators that could be tested as mediators?

6) It is also not clear why the p21 null competitors, which have about 2.5X greater mitotic indices than the p21 null HRAS mutant cells, do not have a competitive advantage. In addition, we do not know if p21 is playing a role in the reversal of the fitness advantage of mutant cells following injury – for example, is p21 lost post-injury in the WT cells?

7) While MAPK activity is promoted by p21 loss, injury and HRAS activation, they do not show whether this MAPK induction is contributing to injury promoted alterations in the competitive dynamics. While it seems likely that MAPK increases are important for the competitive advantage of RAS activated cells in uninjured tissue, it is not shown whether MAPK plays a role in injury-induced promotion of WT competitor fitness.

Minor:

1) Fig 3: It is interesting that WT competitor cell proliferation appears further boosted by the presence of HRAS mutant competitors in the injured setting, which was apparently not seen when both competing populations are WT. However, this conclusion needs to be substantiated by statistical comparisons [compare day 6 WT mitotic indices (green) in Fig 3g and 3h].

2) A more comprehensive study of mutations in normal skin (from the same group as their reference #1) is the recent Fowler et al paper (4).

1. Liu N, Matsumura H, Kato T, Ichinose S, Takada A, Namiki T, et al. Stem cell competition orchestrates skin homeostasis and ageing. *Nature*. 2019;568(7752):344-50.
2. Ballesteros-Arias L, Saavedra V, Morata G. Cell competition may function either as tumour-suppressing or as tumour-stimulating factor in *Drosophila*. *Oncogene*. 2014;33(35):4377-84.
3. Kajita M, Fujita Y. EDAC: Epithelial defence against cancer-cell competition between normal and transformed epithelial cells in mammals. *Journal of biochemistry*. 2015;158(1):15-23.
4. Fowler JC, King C, Bryant C, Hall M, Sood R, Ong SH, et al. Selection of oncogenic mutant clones in normal human skin varies with body site. *Cancer Discovery*. 2020:CD-20-1092.

Referee #3 (Remarks to the Author):

This manuscript reports an interesting observation of potential important pathological implication. Cancer is often compared to a non-healing wound, and wounding/non healing conditions may accelerate e.g. skin cancer. Indeed, e.g. in dystrophic epidermolysis bullosa patients, who carry mutations in collagen VII, the continuous induction of skin blistered promotes a chronic wound conditions, including extensive inflammation, and this has been linked to SCC in these patients, with most patients dying of invasive skin cancer before their 50s. Moreover, wounding may accelerate overgrowth and tumor formation in mouse models with a homogenous expression in which the oncogene, often H-RasV12, is homogeneously expressed, although it is important to note that this may depend where in the epidermis cells are expressed, as early studies suggested that suprabasal expression of mutant Ras using K1 or K10 promoters accelerate tumor formation near sites of scratching or wounding whereas no such relation was observed when expressed basally under K5 conditions (Brown et al., *Current Biology* 1998).

In contrast, normal skin tolerates the presence of epidermal cells with carcinogenic mutations without apparent tumor formation, even if underlying mechanisms are still unclear. The authors now asked whether acute wounding would promote the outgrowth of these cells when in a mosaic condition with wt cells. Interestingly, using either H-RASG12V or the more aggressive K-RasG12 mutations, wounding of the ear under mosaic conditions does not result in further overgrowth of mutant cell, unlike when there is over 95% of mutant cells present. The authors then go on and

show that unlike the mutant cells, wt cells show a strong increase in proliferation upon wounding, most pronounced after 3 days after wounding, whereas mutant cells show a similar proliferation profile in non-wounded versus wounded conditions, thus suggesting that wt cells early after wounding “outcompete” mutant cells. Using a p21 null mutant that releases cell cycle inhibition, they then ask whether induction of proliferation in wt cells is sufficient to prevent outgrowth of mutant cells under non-wounding conditions, and find that this is indeed sufficient. Finally they link this to, not very surprising, MAP kinase signaling.

Overall, these are potential interesting albeit rather preliminary findings that do not yet convincingly show that increased proliferation of wt cells is indeed key to suppress overgrowth of mutant cells under wounding conditions. They only show that increasing proliferation in wt cells is sufficient to keep mutant cell expansion in check under non-wounding conditions, showing that under these conditions mutant cell overgrowth is driven by a proliferative advantage of these cells. This observation although interesting and nicely fitting with the correlative data on the proliferation measurements after wounding, is insufficient to show that this mechanism also keeps mutant cells in check after wounding. This would require (and I agree not a trivial task) the reverse experiment and show that suppression of wt cell growth upon wounding results in outgrowth of mutant cells under mosaic conditions.

Most importantly, the authors do not carefully take into account the role of several other key differences in terms of tissue confinement, tissue structure, and inflammation in normal versus wounded skin, which may play an essential role in how wounding impacts how control cells keep mutant cells in check:

(a) As they make a full thickness wound they basically break the basement membrane, an important barrier in keeping cancer cells at check. Proper healing also requires restoration of epidermal tissue confinement, by secretion of ECM molecules, adhesion to these ECM and ultimately basement membrane reformation that restores tissue compartments. The ability to repair this mechanical and signaling barrier or not repair this barrier and restoring determined by the presence of a sufficient number of wt cells may therefore be essential to prevent outgrowth and ingrowth into the dermis, which is basically what the mutant max cells are doing.

(b) along the same lines, mosaic versus mostly mutant Ras cells may impact on fibroblasts and on the dermal extracellular matrix rearrangements necessary for proper healing that in return also impacts on the epidermis.

(b) ruling out the role of Inflammation that may be different upon wounding (as well as a major driver of the overgrowth or not) depending on whether the tissue is wt, mosaic or has mostly Ras mutant cells (Ras max).

(c) although the authors rule out the role of cell death, they only score cell death at day 3, whereas the readout for tissue thickness as measured to average distance to collagen is done at day 14. Cell death may thus also play an important role later in the process.

Importantly for all of the above points, it seems to me, as also indicated in many reviews, that the mutant Ras cells are in a state of continuous activation, and in that sense behave as a “non-healing” activated keratinocytes (hence their Map kinase activity is different because sustained activation), whereas the wt cells are still able to respond to wounding. These different states may actually evoke a non-healing response in mutant Ras max conditions versus mosaic conditions, where control cells

may advance healing conditions including but very likely not exclusively driven by the ability to proliferate but also by controlling inflammatory responses and coordinating the ability to restore homeostatic tissue architecture.

Is this a general phenomena regardless of the local differences in skin architecture between e.g ear, tail or back skin? As a proof of more general principle does wounding of mosaic mice also suppresses outgrowth when wounded in the back skin, which has more HFs and may thus result in altered proliferative response (and likely a somewhat different wounding response also in terms of inflammation)? The authors do not have to do any in vivo tracing but could show an endpoint in terms of thickness.

The real competitive difference of proliferation of wt above Ras mutant under mosaic conditions is only seen at Day 3 (2-fold) , after that the two have very similar proliferative curves (Fig.3G). Can this 2-fold short term proliferative advantage explain clone sizes, using modeling? Vice versa, as proliferation does not change compared to the non-wounded situation , why not see a similar relative increase in thickness in the unwounded situation when compared to Mutant max wounded condition?

The data on Map kinase are very limited and somewhat simplistic and only provide minimal insight into why wt cells are able to respond or not. The map kinase activity is rather dynamic and often biphasic upon growth factor stimulation. Taking one point is not very helpful, also as the activation under all three conditions seems to be very similar upon wounding regardless whether wt, mosaic or Ras mutant cell max. Normalization should also be done with respect to total map kinase. Western blot analysis also does not allow to examine whether Map kinase activity is higher in wt versus Ras mutant cells under mosaic conditions, which one would predict.

The authors claim in the text that the wt cells are more proliferative under mosaic conditions than under wt conditions only (Figure 3G, right two upper (control mosaic) and lower panel (wt/mutant mosaic). However, this difference is rather minimal and not directly compared within the same graph applying stringent statistics. Looking at all of the data, it seems that wounding induces a similar proliferative response in wt cells regardless whether Ras Mutant cells are present in mosaic conditions.

Some of the statistics is not clear from the legend. Sometimes the authors refer to N= X mice but sometimes just to N=X, in case of the latter what does that mean? Some of the data is also based on n=2 mice, which does not help to show biological variation (n=3 would be the absolute minimum).

Referee #4 (Remarks to the Author):

The authors use sophisticated imaging and molecular genetics to study the role of wounds in tumorigenesis. They use an oncogenic allele of Hras (and Kras) to stimulate squamous tumorigenesis with and without wounding. It is known that chronic wounds are associated with tumorigenesis in the skin, but whether the effect of the wound is on wildtype or mutated cells is not clear.

Interestingly, the authors use imaging to follow the progression of both wildtype and mutant clones of cells in the skin, and found that wildtype cells preferentially expand over mutant cells upon wounding. On the other hand, in the absence of a wound, Hras mutant cells expand. They go on to show that p21, a cell cycle inhibitory protein plays a role in this effect. They summarize their observations to conclude that wildtype cells actually suppress the expansion of Hras mutant cells in response to wounding as a means to suppress tumorigenesis. They suggest that this will lead to improved treatments for cancer via the promotion of wildtype cells in wound repair. The methods employed are elegant and the data collected appear robust. While there are certainly some interesting observations presented, the data raise many questions, and many of the assertions are not well defined. The relation of this study to cancer is also not clear, which raises the issue of impact of the study. If this is not a cancer study is it really more relevant to cell competition? wound healing?

1, the authors refer to tumorigenesis throughout the manuscript, but never show the formation of a tumor. do they mean hyperplasia? benign cyst or growth? papilloma? There are no macroscopic images to show what is really happening.

2, The authors should address the possibility of oncogene induced senescence (OIS). One way to interpret the data is that the Hras-mutant cells are out-competed due to senescence driven by constitutive activity of the Hras allele. Large tumors typically only form in this model when coupled to mutations in p53, which allows mutant cells to overcome OIS.

3, According to data from Phil Jones and others, sun exposed human skin keratinocytes almost universally are mutated for Ras, Notch etc, so it is not clear if the data presented here are relevant to human data since SCCs only really form in sun exposed skin. In other words, how could one promote the wildtype cells to suppress mutant cells, where there are no wildtype cells?

4, The authors probably should have shown that mutations in Hras or Kras using the strategy employed here coupled with injury actually leads to tumorigenesis, since the study is based on this assumption.

5, The authors restricted their analysis to ear punches as a relatively straightforward approach for both uniformity (and probably imaging?). However, in these GEMM models, tumors rarely ever form on the ear, so i am curious if choosing to study the ear has limited the interpretation of the data? can the authors replicate the findings on the backskin? In addition, there are strains of mice that heal ear punches very quickly, it might be worthwhile to study those to see if their proposed mechanism is supported.

6, This is perhaps a minor point, but the authors refer to Kras as an "aggressive" allele, but I am not familiar with this distinction with regard to this particular model system, can the authors show macroscopically that the animals with Kras have a more severe phenotype than those with Hras?

Author Rebuttals to Initial Comments:

Referee #1 (Remarks to the Author)

Expertise skin stem cells, cancer models, lineage tracing

The manuscript by Gallini et al., focuses on characterizing the cellular behavior of HRAS and KRAS mutant cells within the context of the skin epithelium and the competitive relationship between the mutant and WT cells in the context of uninjured and injured skin. In vivo imaging is a powerful and elegant approach to characterize cellular behavior, however, there are a number of concerns about the interpretation of the results that are described in detail below.

Specific comments

R1.1) The major point that wildtype cells following injury outcompete mutant cells carrying the mutated version of Ras (both HRas and KRas) constitutes a conundrum. Most researchers who have worked with these oncogenes activated by various cre drivers observe that a major obstacle for these studies is the development of papillomas in and around the mouth area linked with constant skin abrasion/wounding (e.g. Lapouge et al., 2011, PNAS; van der Weyden et al., 2011, J Pathol) even when only small populations of cells are targeted (Page et al., 2013, Cell Stem Cell).

We thank the reviewer for asking us to reconcile our new findings with previous literature. Indeed, several previous works have shown that wounding can induce papilloma formation from a small targeted stem cell population carrying H- or K-Ras mutation (e.g. Page et al. 2013). Based on our study setup, differences to these previous studies were expected, which we failed to bring out in our original submission and have now brought up early in the main text and methods.

Previous studies on Kras induced tumorigenesis have used mouse strains that are highly sensitive to tumor induction and observed higher frequency of tumors after wounding. In this work, we utilized outbred CD1 mice, which have a generic heterogeneity similar to wild-caught mice and display a lower susceptibility to H/K-ras induced tumors compared to more inbred mice used in previous studies (Aldinger KA et al., 2009, Page et al. 2013). This model allowed us to

REDACTED

study a scenario where limited recombination efficiency in our mosaic mice would not, but **when nearly all epithelial basal cells express the Ras mutation** in the max induction models would lead to tumors upon wounding (**Manuscript Fig. 1, 4, S1, S6**). Importantly, consistent with previous reports, in our model of *homogenous* Kras mutation (Kras^{G12D/+}-max), we also observed rapid development of tumors in body surfaces subjected to constant abrasion, such as mouth and anus, two months after tamoxifen-mediated recombination (timeframe beyond the ones analyzed in the manuscript - **Rebuttal Fig. 1**).

Here it is worth noting that the modality of performed injury is critical: for instance, shallow wounds such those resulting from abrasions have been shown to increase specifically participation of junctional stem cells (Lrig-1 models used in Page et al., 2013) to support injury-repair compared to deeper wounds where repair is more extensively driven by epithelial cells in the interfollicular epithelium (Gonzalez KAU et al., 2021).

A second critical distinction of our study in comparison to prior ones is the approach we have used – intravital imaging – which allows to directly connect the initially labeled number of cells to the resulting

phenotype later on by tracking those same cells over time. This contrasts with some studies that relied on sacrificing a different animal at a different time point and infer connections between these separate temporal snapshots. Additionally, these static approaches cannot control neither for leakiness nor for ectopic tdTomato recombination that occurs as a result of abrasion, induced by topical applications of compounds (i.e. TPA, topical tamoxifen treatment).

R1.2) Moreover, following wounding it is clear that cells expressing oncogenic Ras even from the hair follicle are rapidly mobilized to migrate into the epidermis and form papillomas (e.g. Page et al., 2013, Cell Stem Cell). It consequently appears counter intuitive based on the published literature that cells expressing oncogenic Ras should be outcompeted by WT, and the authors need to address this point experimentally.

We agree with the reviewer that, without the upfront explanation stated above (R1.1), the results of our model would appear counterintuitive.

To experimentally address this reviewer's comment in the context of the wound approaches used in this manuscript, we utilized a Cre inducible mouse model (Lgr5CreER) that targets hair follicle stem cells. Lgr5CreER combined with LSL-tdTomato allele effectively labels hair follicle stem cells following maximal dose of tamoxifen injection at P19 (**Rebuttal 2B**). We adopted this strategy to have a complementary approach to our K14CreER model that targets interfollicular epidermis, junctional zone: infundibulum and isthmus, and sebaceous glands, but only a small proportion of bulge and hair germ. We would like to highlight that our K14CreER models also target the cell population in the junctional zone as in Page et al. 2013. We show two-photon z-stacks to illustrate that the K14CreER model includes targeting of the Junctional Zone and that Lgr5CreER targets the hair germ and bulge as expected (**Rebuttal Fig. 2A**). Using the same strategy we adopted for the other models

in the manuscript, we performed a 4 mm wound in one ear of Lgr5CreER models at P24. Then, we tracked over time the repairing tissue and followed the contribution of tdTomato-positive/Lgr5-positive stem cells in wild-type or Hras^{G12V/+} models (Lgr5CreER-LSL-tdTomato and Lgr5CreER-FR-Hras^{G12V/+}-LSL-tdTomato mouse models respectively). In both models we observed that a small percentage (10% or less) of tdTomato-positive/Lgr5-positive stem cells localize in the interfollicular epidermis up to 1.1 mm away from the wound at 1-month post-injury-induction (**Rebuttal Fig. 2C**). Furthermore, we did not observe abnormal growth in Lgr5CreER; FR-Hras^{G12V/+} models even in the maximal tamoxifen-induction conditions. Collectively, our data indicate that the level of participation of hair follicle stem cells (specifically the bulge/hair germ) to injury-repair of deep wounds is minor and they do not trigger injury-induced aberrant oncogenic growth when carrying a Ras-oncogene mutation.

All together the above additional experiments and data reveal that in our outbred CD1 model *injury triggers papilloma only in the maximal induction of the Ras mutation in epidermis, isthmus and sebaceous gland* and *never when only bulge and hair germ alone are targeted*.

R1.3) The authors perform their experiments on mice, which are 3 weeks of age. A major assumption in the analysis performed by the authors is the ability to alter homeostatic cell behavior, yet at 3-weeks the ear has not reached its final size and they are consequently modelling the effects of the mutation not only in the context of tissue growth but also wounding. It remains unknown whether this impacts their analysis.

We thank the reviewer for bringing up the point of whether possible organ/organism growth may create a variable that was not touched upon. As part of our experimental designs, we always compared Hras^{G12V/+} to wild-type littermates within the same experimental framework (i.e., same alleles minus Hras, same Tamoxifen regimen, etc.) and same stage of development (i.e. 3 weeks of age). With that said, to

REDACTED

determine the extent of tissue growth, we measured the increase of the area between hair follicles over time in the regions tracked in wild-type and Hras^{G12V/+}-mosaic models in both uninjured and injured conditions. ***Our analysis revealed a similar tissue expansion when comparing injured and uninjured mosaic mice either with or without the Hras^{G12V} mutation (Rebuttal Fig 3)***. These data therefore show that the ability of wild-type cells to outcompete Hras^{G12V/+} cells in the wound scenario is not a consequence of tissue expansion due to developmental growth given it is a shared phenomenon across all analyzed models including those uninjured.

R1.4) The authors explore how mutant and wild type HRAS cells compete with one another when present in two different proportions of cells in the epidermis. They are able to demonstrate in line with previous observations that aberrant tissue morphology is induced by injury in mouse models with a high burden of HRASG12V mutations in basal cells (~100%). This is not observed in a model with a lower recombination efficiency (~65%)

referred to as the “mosaic” model. The results of these experiments are however difficult to interpret. The authors use tdTomato as a proxy for HRASG12V mutations. Given that the two loci are located at distinct genomic sites and will require two independent recombination events, this remains an assumption and should be determined. This means that Ras mutants might be tdTomato negative and a fraction of the tdTomato positive cells are likely not mutants for Ras. The authors need to determine the recombination efficiency at the single cell level. Without evidence that directly demonstrates that tdTomato + cells also contain the HRas mutation, it is impossible to conclude “Collectively, these data show that HrasG12V/+ cells break homeostatic tissue architecture during injury-repair only when nearly all of the basal cells express HrasG12V/+. (Line 101-103).

We appreciate the reviewer’s point that current state-of-the-art tools are limited in that. They rely on fluorescent reporters inserted in a different genomic site that therefore recombines independently of the gene of interest. However, prior studies from our and other labs showed that tdTomato along with other fluorescent reporters serve as a good approximation of mutant cells including the knock-in gene Hras^{G12V/+} (Deschene, Myung et al., 2014, Pineda et al., 2019, Beronja et al., 2013; Sandoval et al., 2021). To test the extent of reliability of tdTomato expression to identify Hras^{G12V/+} cells, we have previously published the use of a constitutive Cre that ensures recombination of all LoxP sites present in the genome of a cell (Pineda et al., 2019). In that paper, we documented the same behavioral phenotype of Hras^{G12V/+} cells by which during homeostasis they expand over wild-type cells with the same percentage increase showed in the current manuscript.

REDACTED

Nevertheless, we experimentally addressed this reviewer comment and resolved the proportion of Hras mutation in the tdTomato^{positive} population at the single cell level by next generation sequencing. Specifically, we utilized mice carrying K14CreER; FR-Hras^{G12V/+}; LSL-tdTomato; K14H2B-GFP genotype and sorted GFP^{positive};

tdTomato^{positive} and GFP^{positive}; tdTomato^{negative} cells. Next, we conducted next generation sequencing of the PCR products of the part of the Hras knock-in gene that contains the mutation, and which is actively transcribed (**Rebuttal Fig. 4A**). This experimental design allowed us to determine the percentage of floxed-Hras^{G12V/+} in Hras^{G12V/+}-mosaic and -max models. Of note, the reverse primers contained unique identifiers (UMI) that allowed us to be quantitative in the analysis of the mutant or wild-type form of the Hras gene.

A caveat we identified was that we isolated a lower percentage of tdTomato^{positive} cells after sorting than the percentage we typically observed when we tracked with two-photon imaging despite the mice being treated identically (**Rebuttal Fig. 4B, Manuscript Fig. 1B**). This was due to quenching of the fluorescence intensity of tdTomato that resulted from the single cell isolation and the FACS (Fluorescence-activated Cell Sorting) leading to recombined cells appearing now as false-positive in the tdTomato^{negative} population. Specifically, we were able to recover after sorting: 30-38% of tdTomato^{positive} cells for Hras^{G12V/+}-mosaic and 48-61% for Hras^{G12V/+}-max (**Rebuttal Fig. 4B**). These percentages were lower than 65-80% and 99% of tdTomato^{positive} cells observed with two-photon microscopy for Hras^{G12V/+}-mosaic and Hras^{G12V/+}-max, respectively (**Manuscript Fig. 1B**). Therefore, this approach allowed us to reveal the recombination efficiency of Hras^{G12V/+} only in the tdTomato^{positive} population.

Specifically, we tracked three single point mutations, including the G12V mutation, that are present only in the Hras^{G12V/+} mutant knock-in gene and not in the wild-type form and we found an average of 58.91% of recombination in the Hras^{G12V/+}-mosaic model and 86.16% in the Hras^{G12V/+}-max model (**Rebuttal Fig. 4C, D**). Considering the lower percentage of recombination of Hras knock-in compared to tdTomato and the difference in the length of the floxed fragment in the Hras locus (2.4 kb) and the tdTomato locus (only a STOP codon), we infer that the tdTomato negative population should not have events of recombination in Hras knock-in gene. However, given that the phenotype of Hras^{G12V/+} expansion is unambiguous, the possibility that the tdTomato population may have false negatives would actually strengthen our discoveries as it implies the phenotype may even be more pronounced than what we have documented.

Collectively, these data demonstrate that *the minimum threshold necessary for mutant cells to behave differently and generate oncogenic growths upon injury is when ~86% of the cells express Hras^{G12V/+}*.

R1.5) The authors also argue that they study effects of different degrees of recombination (65 vs 100% according to tdTomato+ cells), but none of these regimens actually simulate what happens, when rare mutated cells with a Ras mutation are placed under homeostatic or injury conditions. Is there a certain threshold percentage, where mutant cells behave differently or is only a small fraction of Ras mutants that go on to behave differently e.g. by changing their environment?

To address this reviewer question, we used an approach that allows for the induction of rare Hras^{G12V/+} cells in uninjured and injured conditions. In short, we transduced the epidermis of mid-gestation embryos (E9.5) of FR-Hras^{G12V/+}; LSL-tdTomato mouse models with lentivirus containing inducible Cre-recombinase (LV-CreER), using an ultrasound-guided intra-amniotic injection (Beronja et al., 2013; Beronja et al., 2010; Ying et al., 2018; Pineda et al., 2019). Previously, we have successfully used this technique to demonstrate the competitive advantage of Hras^{G12V/+} cells over wild-type cells in the uninjured setting (Pineda et al., 2019).

Given lentiviral CreER titer can be adjusted to achieve a low degree of mosaicism, we effectively infected only a small number of epidermal cells by performing in utero injections in both pregnant moms carrying pups with FR-Hras^{G12V/+}; LSL-tdTomato; K14H2B-GFP genotype or control LSL-tdTomato; K14H2B-GFP genotype. To ensure complete recombination within these rare cells, we treated these mice with a maximal dose of tamoxifen at 3 weeks old (P19) followed by injury in one ear 3 days later (4 mm diameter punch biopsy down to the cartilage) consistent with the methods used in the manuscript. First, we confirmed previous discoveries that the Hras^{G12V/+}/tdTomato^{positive} population steadily expanded in the uninjured Hras^{G12V/+}-mosaic epithelium, in contrast to the tdTomato^{positive} population in the uninjured wild-type-mosaic epithelium where the tdTomato clones remain unchanged (**Manuscript Fig. S2C, D**). Second, upon injury, the tdTomato^{positive} clones maintained

the same size in both Hras^{G12V/+}- and wild-type-mosaic epithelia. Of note this was the case for both clones containing as low as 2 cells and up to ~50 tdTomato^{positive} cells. This finding indicates that Hras^{G12V/+} cells failed to outcompete wild-type cells and expand after injury-repair of Hras^{G12V/+}-mosaic mice also in the presence of a small percentage of Hras^{G12V/+} basal stem cells (**Manuscript Fig. S2E, F**). Collectively, these data demonstrate that the **threshold where mutant cells generate oncogenic growths upon injury is when nearly all cells in the basal stem cell layer express Hras^{G12V/+}** confirming the discoveries reported in the manuscript.

R1.6) The authors argue that “Intriguingly, HrasG12V/+-mosaic models behaved similarly to wild-type-mosaic models (**Fig. 1c, d**” (Line 95-96). However, although the differences might not be significant, it is a tall order to argue that mosaic wt and mosaic Ras models behave similarly. There are obvious differences between the two types of samples.

We appreciate the review’s remark. We would like to clarify that the message we liked to convey is that Hras^{G12V/+}-mosaic mice did not develop injury-induced tumors which was similar to wild-type models and in contrast to Hras^{G12V/+}-max mice. However, the reviewer is correct that for instance we did observe an increase in the thickness that was proportional to the amount of tdTomato^{positive} cells during injury-repair in Hras^{G12V/+}-mosaic. A similar observation was also made in our previous paper in the uninjured context (Pineda et al., 2019). We changed the text accordingly to both provide further explanations on this point and highlight the observed significant difference in the resistance of the Hras^{G12V/+}-mosaic model to induce oncogenic growth upon wounding when compared to the Hras^{G12V/+} max model.

R1.7) In the competition assays between WT and Ras mutant cells (Figure 2), it is unclear why the authors observe such as steep increase in the tdTomato +ve area 10 days after induction in the uninjured area. After this initial burst the increase in area covered is not nearly as pronounced. Are there systemic effects from the wounding specifically affecting the tdTomato +ve cells?

We thank the reviewer for the interesting point. Based on our data and Sandoval et al., 2021, the difference in the steep increase of the tdTomato area 10 days after induction is due to the larger size and proliferative rate of Hras^{G12V/+} clones compared to wild-type neighbors and the presence of areas still occupied by wild-type cells that facilitate Hras^{G12V/+} cell expansion. These data are further supported by our *in utero* injection experiment described in the answer to point 1.5 that allowed us to follow small Hras^{G12V/+} clones. In this condition, Hras^{G12V/+} cells steadily expanded over time without reaching a plateau or any steep increase at a particular time point in the month analyzed (**Manuscript Fig. S2C, D**). To further address potential systemic effects on the observed phenotypes, we collected replicates of the contralateral, unwounded ear from Hras^{G12V/+}-mosaic that have or have not been wounded. The comparison between those conditions showed that systemic effects from the wound do not affect Hras^{G12V/+}/tdTomato^{positive} cell expansion in the uninjured, contralateral ear of injured Hras^{G12V/+}-mosaic models and its significant difference compared to the injured ear (**Rebuttal Fig. 5**).

REDACTED

R1.8) Moreover, the results presented for the WT cells in figure 2c are surprising, since they show a steady increase in the percentage of tomato positive cells over time in intact skin. This is inconsistent with an equipotent basal progenitor model originally published for the tail epidermis (Clayton et al., 2007, Nature) and subsequently adopted for the ear epithelium by the Jones (Doupe et al., 2010, Dev Cell) and Greco labs (Rompolas et al., 2016, Science). In an equipotent model the number of labeled cells should remain constant.

We appreciate this point. The number of labeled cells did not remain constant in wild-type-mosaic mice because *the CreER/LoxP system exhibits a certain degree of tamoxifen-independent “leaky” Cre activity* which is a known characteristics of this tool (**Rebuttal Fig. 6**; Alvarez-Aznar et al., 2020; Lapouge et al., 2011; Fish et al., 2020; Kong et al., 2016). Indeed, littermates that did not harbor the CreER transgene did not show recombination of the targeted genes such as tdTomato, whereas littermates that did harbor the CreER transgene accumulated a small number of tdTomato^{positive} cells even in the absence of tamoxifen. This phenomenon is happening at different degrees in all Cre lines in the field, but it is often neglected or not documented. One advantage the live imaging approach provides us with is that it allows us to track the same cells over time and monitor the basal activity of CreER. Furthermore, to control for the leakiness of the system in our experimental design and correctly interpret the data, we always compared wild-type and Hras^{G12V/+}-mosaic models. We have added a comment in *Materials and Methods* to document the leakiness of the CreER system.

Rebuttal Fig. 6. *Top:* Top down (x-y) views of two-photon images of the skin epithelium at 6 days post-tamoxifen injection of LSL-tdTomato; K14H2B-GFP wild-type or Hras^{G12V/+} mouse. *Bottom:* Top down (x-y) views of two-photon images of the skin epithelium at the same time frame as the top images but without tamoxifen injection of K14CreER; LSL-tdTomato; K14H2B-GFP wild-type or Hras^{G12V/+} mouse. Scale bar = 20 µm.

R1.9) This may be a direct consequence of the small area analyzed in the study and it would consequently be important to provide robustness by increasing the areas analyzed. According to the figure legend, one area was imaged for these quantifications, and adding the independent measurements will clearly assist in the interpretation of robustness and the evaluation of whether the statistical tests used are appropriate.

We thank the review for highlighting the lack of clarity in the figure legends. We have modified them to reflect the three independent areas of 300 µm² that we analyzed for each mouse (*see Material and Method-900 µm² for each mouse*). The areas were selected to allow the revisit of the exact same region of the uninjured and injured skin epithelia over time by following the same hair follicle pattern.

R1.10) To explain the observed increase in competitive drive for the WT cells the authors performed analyzes of the frequency of mitosis observed in HRASG12V mutant cells vs their WT counterparts in intact and injured skin. Here there are a number of observations that are left unclear. Why is there only a transient increase in mitotic events in the HRasG12V/tdTomato population in the uninjured tissue?

The reviewer is correct that Hras^{G12V/+} cells were significantly more proliferative than wild-type cells only from 6 days to 17 days post-tamoxifen injection. This resulted in mutant cell expansion reaching high coverage of the basal layer which we believe may contribute to why proliferation is not sustained at similar level 2 weeks later. Of note, at 1-month post-tamoxifen injection, *Hras^{G12V/+} cells and wild-type cells displayed similar proliferative potential, suggesting that the two populations reach a point of near homeostasis where*

Hras^{G12V/+} cells are tolerated by the skin epithelium. This is consistent with a previous report (Sandoval et al., 2021) that showed that in the presence of broad area of Hras^{G12V} induction, similar to our experimental setting at 1-month post-tamoxifen, Hras^{G12V/+} cells reduced their rate of cell division to the wild-type level and increased their differentiation capacity through Rassf5 activity. This slows down Hras^{G12V/+} clone expansion. All together, these data point to a model where the equal proliferative rate of wild-type and Hras^{G12V/+} cells ultimately reached in the uninjured Hras^{G12V/+}-mosaic model is a defense mechanism adopted by the skin epidermis to contrast an excessive expansion of Hras^{G12V/+} cells. This established new homeostatic equilibrium leads to skin epidermal oncogenic tolerance (Sandoval et al., 2021, **Manuscript Fig 3C**). We infer that the injury context accelerates the establishment of this new homeostatic equilibrium potentiating wild-type cell proliferative capacity to surpass first and equalize later Hras^{G12V/+} cell proliferative potential.

We have modified the graph in **Figure 3C, D** of the manuscript to show appropriate statistical analysis and facilitate the comparison of tdTomato^{positive} and ^{negative} cells between Hras^{G12V/+} and wild-type-mosaic models in uninjured and injured conditions.

R1.11) Also, why is there around 5 times as many proliferating WT cells in the uninjured skin in the animal with HRas mutation (3g vs h), and similarly many more proliferating WT cells in the injured condition when compared to the controls? The increase in competitiveness of wt cells is interesting and warrant further exploration as to the specific mechanisms inducing these changes, and is left completely un-explored.

As we mention in the answer 1.10 above, we have modified the graph in Figure 3C, D (former Fig. 3G, H) to show appropriate statistical analysis and allow the comparison between Hras^{G12V/+} and wild-type-mosaic models. Our data showed that wild-type cells increased their proliferative capacity and initially surpass and later equalize Hras^{G12V/+} cell proliferation during injury-repair. These behaviors allowed the establishment of a proliferative balance between mutant and wild-type clones by 7 days post-injury induction (10 days post-tamoxifen injection). A similar equalizing of wild-type and Hras^{G12V/+} proliferation rates was also documented at one-month post-tamoxifen in uninjured condition. The molecular mechanisms explaining this equalization of wild-type and Hras^{G12V/+} cell proliferation *in uninjured tissue* were explored by Sandoval et al., 2021, as we commented in the R1.10 point above.

To address the reviewer's request, we explore further the mechanisms underlying wild-type competitiveness *in injured tissue* and we found that wild-type cells are more responsive to autocrine and paracrine secretion of EGF ligands than Hras^{G12V/+} cells (**Manuscript Fig. S7, S8, S9C**) (*for more details please see the comment R1.13 below*). To test the possible role of EGFR ligands in promoting a selective proliferative advantage of wild-type cells in the injured Hras^{G12V/+}-mosaic model, we chemically and genetically inhibited EGFR with Gefitinib treatment or the expression of EGFR-DN (dominant negative) in the basal stem cells layer. EGFR inhibition did not affect Hras^{G12V/+} cell proliferation during injury-repair but, conversely and excitingly, it selectively reduced wild-type cell proliferation during injury-repair. This was sufficient to reestablish the competitive advantage of Hras^{G12V/+} cells over wild-type neighbors upon wounding (**Manuscript Fig. 5A, B, C Fig. S9D, E, F**). This result supports a mechanism by which the selectively increase in wild-type cell proliferation mediated by EGFR ligands secreted during injury-repair ultimately suppress the expansion of Hras^{G12V/+} cells.

In the original manuscript we also examined whether increased proliferation of wild-type cells is sufficient to suppress the competitive advantage of Ras-mutant cells even in the uninjured context. To this end, we combined the p21^{null} model with the tamoxifen-inducible Hras^{G12V/+} model (K14CreER; LSL-Hras^{G12V/+}; LSL-tdTomato; constitutive p21^{null}; K14H2B-GFP). Excitingly, p21 loss increased the proliferation of wild-type cells but not of Hras^{G12V/+} cells in p21^{null}-Hras^{G12V/+}-mosaic mice, mimicking the selective increase in wild-type cell proliferation during injury-repair of Hras^{G12V/+}-mosaic mice (**Manuscript Fig. 5D, E**). This was sufficient to suppress the competitive advantage of Hras^{G12V/+} cells over time in uninjured mice, recapitulating the effects of injury-repair in Hras^{G12V/+}-mosaic mice (**Fig. 5F, G, Fig. S9G**). In the revisited manuscript, we expanded on the molecular details behind the specific effect of the global knock-out of p21 on wild-type cell proliferation. We found that p21 expression both at mRNA and at protein level was already downregulated in Hras^{G12V/+} cells in

the uninjured mosaic skin epithelium when compared to wild-type cells (*for more details please see the comment R1.12 below*).

Overall, the new data provide molecular insights in the key role of wild-type cell proliferation in preventing Hras^{G12V/+} cell expansion in Hras^{G12V/+}-mosaic models to establish a new homeostatic equilibrium and oncogenic tolerance.

R1.12) The authors argue that loss of p21 could be the mechanism providing the WT cells with their selective advantage during tissue regeneration, yet they provide no evidence for this.

We thank the reviewer for this important point. We agree that providing molecular insights on the role of p21 in the cell competition between wild-type and Hras^{G12V/+} cells will further develop our model.

In the skin epidermis, the G1/S cell-cycle inhibitor p21 is expressed in G1-cells of the basal stem cell layer (Haensel et al., 2020). Based on the homeostatic function of p21, we hypothesized that Hras^{G12V/+} cells downregulate p21 expression to promote proliferation. To this end, we quantified the levels of p21 mRNA (newly generated scRNA-seq datasets, **Manuscript Fig. S4, Fig. S10C, D, E**) and protein (epidermal preparation immunostaining, **Manuscript Fig. S10A, B**) in the basal stem cells of uninjured Hras^{G12V/+} and wild-type-mosaic models. We showed that both p21 mRNA and protein were significantly downregulated in Hras^{G12V/+}-mosaic models compared to wild-type-mosaic models (**Manuscript Fig. S10B, C**). Furthermore, we compared non-cycling tdTomato+ cells in the Hras^{G12V/+}-mosaic and wild-type-mosaic models to exclude cycling cells, which physiologically downregulate p21 expression. We observed the same significant difference in p21 expression levels between non-cycling wild-type cells in the wild-type-mosaic and non-cycling mutant cells in the Hras^{G12V/+}-mosaic (**Manuscript Fig. S8D, E**). These data confirm that Hras^{G12V/+} cells express a lower level of p21, and that knocking out p21 in all cells is an effective approach to selectively manipulate wild-type cells. Consistent with this, the p21^{null} model showed increased proliferation of wild-type cells but not Hras^{G12V/+} cells, mimicking our observations of cell proliferation behaviors during injury-repair (**Manuscript Fig. 5D, E**). To understand whether p21 is

REDACTED

playing a role in increasing wild-type cell proliferation following injury, we measured the protein level of p21 in wild-type models in uninjured and injured conditions and observed a downregulation of p21 expression upon injury-induction. Considering the role of p21 as a cell cycle inhibitor, we conclude that its downregulation is required to increase the proliferative capacity of wild-type cells during injury-repair (**Rebuttal Fig. 7**).

These insights further expand the significance of the p21^{null} experiments we had in our manuscript. Specifically, the low level of p21 expression in Hras^{G12V/+} cells allowed us to explore a competition scenario wherein *p21 loss in the wild type cells of the Hras^{G12V/+}-mosaic model selectively increased their proliferation resulting in the prevention of Hras^{G12V/+} cell expansion even without injury* (Manuscript Fig. 5D, E, F, G).

R1.13) Much more sophisticated analyses are required to address mechanistically the interplay between WT and mutant cells. What is the role of the underlying mesenchyme?

These are fascinating questions of great interest to us. To experimentally address this reviewer's comment, we performed single-cell RNA-sequencing (scRNA-seq) of skin cells in the wild-type, Hras^{G12V/+} -mosaic, and Hras^{G12V/+} -max models at day 3 PWI, time point at which the competitive scenario in Hras^{G12V/+} mosaic models changes during injury-repair. Specifically, we performed epidermal cell isolation, which enriches for keratinocytes, while still containing other closely attached cell types (fibroblasts, immune cells etc), that we used to generate 10X Genomics scRNA-seq libraries. Analyses of the resulting datasets were performed as our previous work (Cockburn and Annusver et al., 2021) in close collaborations with Dr Maria Kasper and Karl

REDACTED

Annusver (Karolinska Institute) as well as Dr Nur Rahman (Yale). Focusing on fibroblasts, we identified 4 distinct fibroblast clusters, 2 clusters in the uninjured condition and 2 additional clusters that emerge in the injured condition for all the models analyzed. Intriguingly, the observed clustering was not affected by the induction status (mosaic or max) of $Hras^{G12V/+}$ (**Manuscript Fig. S7**). We also did not see model-specific differences between either the overall transcriptional pattern of the injury-repair associated clusters nor in the transcripts involved in deposition/remodeling of extracellular matrix (ECM) and basement membrane (**Rebuttal Fig. 8A, B**).

Next, we looked at soluble factors secreted by fibroblasts that are known to influence epithelial cell behaviors and found that one of the injury-associated fibroblast clusters displayed significantly upregulated levels of the two EGFR ligands: Hbegf and Ereg compared to other growth factors involved in injury-repair (**Manuscript Fig. S7**). Specifically, the upregulation of Hbegf upon injury was present in all different mouse models analyzed (**Rebuttal Fig. 9**).

Notably, EGFR is one of the best characterized upstream activators of the Ras pathway and previous studies have shown that EGFR signaling promotes epithelial cell proliferation during injury-repair within wild-type skin epithelium (Chen X, et al., 2019, Berlanga-Acosta J, et al. 2009, Repertinger SK, et al., 2004, Takazawa Y, et al., 2015). In agreement with this, our western blot analyses and quantification from wild-type mice showed increased EGFR activation (phosphoEGFR) in injured versus uninjured conditions (**Manuscript Fig. S9C**).

We expanded our analysis on EGFR ligands to two other populations involved in injury-repair: immune and epithelial cells (**Manuscript Fig. S8**). While immune cells displayed an increase in EGFR ligand expression in a minor number of cells, epithelial cells had a significant increase in Hbegf, Ereg, Epgn and Tgf-alpha (**Manuscript Fig S8**), overall appointing fibroblasts and epithelial cells as sources of EGFR ligands during injury-repair. Of note, we also performed ELISA assays on a panel of 32 soluble factors (Bio-Plex-Pro Mouse Cytokine Standard 23-Plax Group I and 9-Plax Group II - Bio-rad). This strategy allowed us to test whether cytokines prevalently associated with inflammation were differentially expressed in injured wild-type, $Hras^{G12V/+}$ -mosaic, and $Hras^{G12V/+}$ -max models. Our data did not show a significant increase in any of the factors analyzed upon injury (**Rebuttal Fig. 10**). Collectively, these data suggested that *epithelial cell proliferation is regulated by paracrine and autocrine secretion of EGFR ligands*.

To test the role of EGFR ligands in the selective proliferative advantage of wild-type cells in the injured mosaic skin epithelium, we inhibited the EGF-Receptor both chemically and genetically. Treatment with the EGFR inhibitor Gefitinib has been documented to down-regulate downstream targets of EGFR/Ras, such as ERK1/2, in different epithelial models, including the skin epithelium (Buday L, et al., 1993, Ono M, et al., 2004, Amann J, et al., 2005, Chen X, et al., 2019). In contrast, several cancers with persistent activation of the Ras

REDACTED

pathways are resistant to EGFR inhibitors, including Gefitinib (Chen J, et al. 2013, Qin B, et al., 2006, Janmaat ML, et al., 2003). Thus, we hypothesized that Gefitinib treatment may lead to inhibition of proliferation specifically in wild-type cells.

REDACTED

We determined the optimal dose of the inhibitor in wild-type-mosaic mice by gavage administration of 100 or 200 mg/kg Gefitinib/body weight, for 6 days, starting treatment 2 days before injury and continuing until 3 days after injury. We found that the 6-day 200 mg/kg of Gefitinib treatment inhibited EGFR and ERK1/2 activation at day-3 post-injury (**Manuscript Fig. S9B**). Additionally, the 200 mg/kg of Gefitinib treatment significantly reduced the cell proliferation during injury-repair of wild-type-mosaic mice, as expected, without increasing cell death events or causing overall weight loss (data not shown).

Next, we treated Hras^{G12V/+}-mosaic models with vehicle (0.5% methylcellulose and 0.2% Tween-80 in water) or 200 mg/kg of Gefitinib, as described above. We found that EGFR inhibition did not affect Hras^{G12V/+} cell proliferation during injury-repair of Hras^{G12V/+}-mosaic mice, consistent with proliferation being independent of EGFR activation due to constitutively active Ras (**Manuscript Fig. 5A and Fig. S9A**). Conversely and excitingly, ***EGFR inhibition selectively reduced wild-type cell proliferation in Hras^{G12V/+}-mosaic models during injury-repair (Manuscript Fig. 5A)***, suggesting that their increased proliferation requires EGFR/Ras activation. This is consistent with our western-blot data that showed a reduced EGFR protein levels in the skin epithelium that expresses constitutively active Hras^{G12V}, as previously shown *in vitro*, which further corroborate the reduced sensitivity of Hras^{G12V/+} cells to injury and EGFR activation (**Manuscript Fig. S9C**, Derer S, et al., 2012).

We also tracked the percentage of surface coverage of Hras^{G12V/+} cells (tdTomato-positive) at 6-, 10-, 17-days post-tamoxifen injection (3-, 7-, 14-days post-injury) (**Manuscript Fig. 5B, C**). Strikingly, we discovered that ***EGFR inhibition, followed by a reduced wild-type cell proliferation post-injury, reestablishes the competitive advantage of Hras^{G12V/+} cells over wild-type neighbors in Hras^{G12V/+}-mosaic models during injury-repair.***

Given that drug treatments do not specifically target the epithelium but can affect additional skin cell types, we complemented with genetic approaches. Specifically, we utilized a tet-inducible dominant negative form of EGFR (TRE-EGFR-DN) that does not express the cytoplasmic tyrosine kinase domain (Roh M, et al., 2001 – JAX:010575), whose expression can be specifically driven in basal stem cell layer of the skin epidermis (Keratin14rtTA). We then combined the doxycycline-inducible EGFR-DN mouse model with the tamoxifen-inducible Hras^{G12V/+} model (K14CreER; K14rtTA; FR-Hras^{G12V/+}; LSL-tdTomato; TRE-EGFR-DN; K14H2B-GFP). Our quantifications showed that the expression of EGFR-DN selectively inhibited the proliferation of wild-type cells, but not of Hras^{G12V/+} cells, in Hras^{G12V/+}-mosaic models after injury, consistent with the Gefitinib treatment (**Manuscript Fig. S9D**). To assess how EGFR-DN affects cell competition during injury-repair, we tracked the percentage of surface coverage of Hras^{G12V/+} cells (GFP+/tdTomato+) at 3-, 7- and 14-days PWI (6-, 10-, 17- days

post-tamoxifen-induced mosaicism). Strikingly, we discovered that the specific EGFR inhibition in the basal stem cell compartment reestablishes the competitive advantage of $Hras^{G12V/+}$ cells over their wild-type neighbors during injury-repair of mosaic mice (**Manuscript Fig. S9E, F**). Collectively, these data further confirmed the result obtained with the inhibition of EGFR by Gefitinib treatment (**Manuscript Fig. 5A, B, C**).

All together, these experiments demonstrate that *EGFR-mediated wild-type cell proliferation counteracts the expansion of $Hras^{G12V/+}$ cells in mosaic skin after injury*.

Collectively, our data demonstrate that the activation of the EGFR/Ras signaling pathway through the secretion of EGFR ligands from fibroblasts and epithelial cells is required for the selective increase in wild-type cell proliferation to prevent $Hras^{G12V/+}$ cell expansion in $Hras^{G12V/+}$ -mosaic models upon injury.

R1.14) Are there some kind of quorum sensing in the culture, where a certain percentage of mutant cells provides one effect whereas a higher or lower fraction will induce different changes? These are all very pertinent questions that the authors never touch upon, which would be expected based on the title and abstract.

In the manuscript we showed that the different percentage of *$Hras^{G12V/+}$ cells in mosaic versus max induction does make a difference at the tissue level* regarding the formation of oncogenic growth around the wound site. Here, we have expanded our analyses to cover a very low recombination rate that represents a more physiological scenario. First, we would like to specify that all experiments are done *in vivo* and not in cell culture. As we reply to the comment R1.5 above, *in utero* injection of LV-CreER allowed us to induce a low number of mutation events and in this scenario, we showed that the $Hras^{G12V/+}/tdTomato^{positive}$ population steadily expanded in uninjured $Hras^{G12V/+}$ -mosaic epithelium, in contrast to the $tdTomato^{positive}$ population in uninjured wild-type-mosaic epithelium where the $tdTomato$ clones do not expand (**Manuscript Fig. S2C, D**). Upon injury, $tdTomato^{positive}$ cell behaviors were similar in both $Hras^{G12V/+}$ - and wild-type-mosaic epithelia, $Hras^{G12V/+}$ cells failed to outcompete wild-type cells and expand after injury-repair of $Hras^{G12V/+}$ -mosaic mice also in the presence of a small percentage of $Hras^{G12V/+}$ basal stem cells (**Manuscript Fig. S2E, F**). Collectively, these data demonstrate that the overall *threshold where mutant cells behave differently and generate tumors upon injury is when nearly all cells in the basal stem cell layer express $Hras^{G12V/+}$* (~86% of mutant cells based on the Next Generation Sequencing data in comment R1.4).

Referee #2 (Remarks to the Author):

Expertise cell competition in cancer

Gallini and colleagues examined the fitness of epidermal stem cells with mutational RAS activation (HRAS or KRAS) and the impact of injury on this fitness. The use of both HRAS and KRAS models, with mutational activation at endogenous loci, helps substantiate their results (these models are elegant and powerful). A key feature of their experiments is that the impact of injury on the malignant progression of skin basal cells expressing activated HRAS-G12V or KRAS-G12D is assessed in a mosaic context where WT cells remain, contrasting with previous studies in homogeneous models. They obtained the opposite result from that expected by previous studies in homogeneous models – injury actually promoted the ability of WT cells to outcompete the mutant ones, and malignant growth was not observed. In contrast, when they induced HRAS activation in ~100% of cells, injury induced malignant growth. Thus, homeostatic architecture is only disrupted by injury when nearly all cells express mutant HRAS. They had previously shown that RAS mutations provided a competitive advantage to skin cells in mosaic contexts (recapitulated here), so it is interesting that injury would abrogate this advantage (with the mutant cells no different than WT).

They further showed that the competing WT cells exhibited enhanced mitotic activity post-injury, while injury did not increase mitoses in the RAS activated cells, which already had higher mitoses (but less than WT competitors in the injured context). In contrast, in uninjured tissue the HRAS (or KRAS) activated cells exhibited

greater mitotic activity, consistent with their competitive advantage in uninjured skin.

They then showed that global loss of the CDK inhibitor p21 suppressed the competitive advantage of HRAS mutant cells in uninjured skin. It thus appears that losing p21 “levels the playing field” by selectively increasing proliferation of the WT cells, taking away the edge that HRAS mutant cells normally have. They further show that MAPK is activated by both p21 loss and by injury, as it is by HRAS mutation (as expected).

Data are supportive of their conclusions and are robust. Experiments have sufficient numbers of repetitions (minimally 3) and are properly analyzed statistically. The manuscript is well written and their points are conveyed clearly and fairly.

The discoveries in this manuscript are exciting and very novel, and could be fundamentally important, revealing a new mechanism whereby injury could be tumor suppressive. The authors are correct that their data support a different approach to cancer prevention: specifically boosting the fitness of WT cells, rather than suppressing all cells as with traditional chemo.

We thank the reviewer for their assessment of the novelty of our work and for the thoughtful and helpful suggestions, which we have addressed below in detail. We agree with the reviewer that our data will support the development of new cancer prevention therapies through the potentiation of the natural mechanism of defense such as the fitness of wild-type cells.

Still, there are major concerns regarding the generality and physiological relevance of their discovery. In addition, these studies do not provide much mechanistic insight into how injury alters competitive dynamics.

Major concerns:

R2.1) They are inducing recombination to generate activated HRAS in 60-80% of cells. Of course, oncogenesis begins with mutations to single cells. While this is harder to model, it would be important to determine the fitness effects of HRAS activation and the impact of injury under the more physiological context of a much lower fraction of oncogenically activated epidermal stem cells (e.g. activation of HRAS in 1-5% of cells, where most cells with recombinations will be surrounded by WT cells). Such a modification is particularly relevant given the many demonstrations that cell competition is dependent on ratios of winners and losers, with a dominance of winners being required to eliminate a loser (which can be a cell with an oncogenic mutation) (1-3).

To address this Reviewer 2 question (also shared by Reviewer 1 point 1.5), we used an approach that allows for the induction of rare Hras^{G12V/+} cells in uninjured and injured conditions. In short, we transduced the epidermis of mid-gestation embryos (E9.5) of FR-Hras^{G12V/+}; LSL-tdTomato mouse model with lentivirus containing inducible Cre-recombinase (LV-CreER), using an ultrasound-guided intra-amniotic injection (Beronja et al., 2013; Beronja et al., 2010; Ying et al., 2018; Pineda et al., 2019). Previously, we have successfully used this technique to demonstrate the competitive advantage of Hras^{G12V/+} cells over wild-type cells in the uninjured setting (Pineda et al., 2019).

Given lentiviral CreER titer can be adjusted to achieve a low degree of mosaicism, we effectively infected only a small number of epidermal cells by performing in utero injections in both pregnant moms carrying pups with FR-Hras^{G12V/+}; LSL-tdTomato; K14H2B-GFP genotype or control LSL-tdTomato; K14H2B-GFP genotype. At 3 weeks old (P19), we treated them with a maximal dose of tamoxifen followed by injury in one ear 3 days later (4 mm diameter punch biopsy down to the cartilage) consistent with the methods used in the genetic models described in the manuscript. First, we confirmed previous discoveries that the Hras^{G12V/+}/tdTomato^{positive} population steadily expanded in uninjured Hras^{G12V/+}-mosaic epithelium, in contrast to the tdTomato^{positive} population in the uninjured wild-type-mosaic epithelium where the tdTomato clones remain unchanged

(**Manuscript Fig. S2C, D**). Second, upon injury, the tdTomato^{positive} clones maintained the same size in both Hras^{G12V/+} and wild-type-mosaic epithelia. Of note this was the case for both clones containing as low as 2 cells and up to ~50 tdTomato^{positive} cells. This finding indicates that Hras^{G12V/+} cells failed to outcompete wild-type cells and expand after injury-repair of Hras^{G12V/+}-mosaic mice also in the presence of a small percentage of Hras^{G12V/+} basal stem cells (**Manuscript Fig. S2E, F**). Collectively, these data demonstrate that *the dominance of winners, wild-type cells, is not sufficient to eliminate the losers, Hras^{G12V/+} cells*.

R2.2) These data derive from a single context of injury (ear punch) with two very similar models (activation of a RAS gene using similar GEMM), raising concerns about the general relevance. The demonstration that injury is tumor suppressive and might help eliminate pre-transformed cells would be very important, and thus they need to substantiate that injury induced elimination of initiated cells happens physiologically (either in humans or in mice). What is important is that this manuscript indicates that such elimination CAN happen in this particular mouse model. What they need to show is that it DOES happen in some natural context, or at least some additional contexts.

We thank the reviewer for their comment and wanted to clarify that we did not show elimination of Hras^{G12V/+}/Kras^{G12D/+} cells but rather containment of oncogenic mutant behaviors. We think that our findings represent a key discovery with clinical relevance for surgical treatment of tumor that contain Ras oncogene, providing valuable insights into the protective mechanisms against Ras-driven tumor formation and relapse. We can speculate that similar behavior may be occurring in natural contexts that human skin experiences, and with other mutations in addition to Ras such as the loss of function of p53 (Fernandez-Antoran D et al. 2019). This is consistent with the skin ability to maintain homeostasis despite continuous injuries and the high burden of somatic mutations (Fowler et al., 2020; Martincorena et al., 2015). We are not generalizing our discoveries to all the oncogenic mutations but specifically went in great depth with one of the major drivers of skin cancer – Ras oncogene - that confers a competitive advantage to mutant cells in the normal skin epidermis. Additionally, the revised manuscript provides important molecular and functional insights underpinning our discoveries (see answers R2.5, new **Manuscript Fig. 5A-C, Fig. S9A-F**) and repeating all these experiments with an entirely new genetic model would be beyond the scope of this manuscript. Importantly, we think that our demonstration that the wound context, which is normally associated with tumorigenesis, can instead contain expansion of Ras oncogenic cells changes the way we think about the consequences of injury for future works. These conclusions were reflected on both the title and the manuscript.

R2.3) They show that WT competitors have greater mitotic activity than RAS activated cells in the skin specifically in the context of injury. What is not clear is why the WT cells do not outcompete the RAS activated cells in the injured skin, given the greater mitotic activity of the former. Some insight into this quandary should be provided.

We thank the reviewer for pointing out the lack of clarity in describing the results. We documented a transient spike in wild-type cell proliferation on day-3 post-wound induction (PWI). However, as we tracked the tissue and cells over time, we observed that at later time points (day-10, day-17 and 1-month PWI), the rate of mitotic events between wild-type and Hras^{G12V/+} cells became similar (**Manuscript Fig. 3D**). This equal proliferative capacity supports the observed constant number of Hras^{G12V/+} and wild-type cells over time and the lack of outcompetition of wild-type cells over Hras^{G12V/+} cells. In addition, after injury, both Hras^{G12V/+} and wild-type cells increase their rate of differentiation (**Manuscript Fig. 3H, Fig. S3H**). As a consequence, *during injury-repair, wild-type cells ultimately suppress the expansion of Hras^{G12V/+} clones observed in the uninjured skin epithelium*.

We infer that *the increase in differentiation of both wild-type and Hras^{G12V/+} cells after injury accommodates the increase in wild-type cell proliferation in the basal layer* (Rebuttal Table 1). During injury-

repair, wild-type cells have an increased contribution in replenishing cells lost to differentiation, constraining the expansion of Hras^{G12V/+} cells and allowing efficient repair of the wound.

At later time point (7 days PWI) both proliferation and differentiation capacity of mutant and wild-type cells is equal that support the absence of the expansion of the one population at the expenses of the other (**Manuscript Fig. 3D, Fig. S3I**).

To test the cause-effect relationship of wild-type cell proliferation in counteracting Hras^{G12V/+} cell expansion, we reduced the proliferation of wild-type cells during injury-repair (*please see comment 2.5 below for more details*). Specifically, we employed the EGFR inhibitor Gefitinib, and showed that it repressed the activity of EGFR and ERK1/2 (downstream target of Ras signaling) during injury-repair of wild-type-mosaic mice (**Manuscript Fig. S9A, B**), and selectively inhibited the proliferation of wild-type cells, but not of Hras^{G12V/+} cells (**Manuscript Fig. 5A**). We identified that the reduced sensitivity of Hras^{G12V/+} cells to EGFR inhibition was due to a low protein level of EGFR in the skin epithelium that expresses constitutively active Hras^{G12V}, as previously shown *in vitro* (**Manuscript Fig. S9C**, Derer S, et al., 2012). Our data showed that ***the reduction of the proliferation of wild-type cells but not Hras^{G12V/+} cells achieved with Gefitinib treatment suppressed the competitive advantage of wild-type cells and allowed Hras^{G12V/+} cells to expand during injury-repair.*** Of note, we obtained the same result by expressing the Dominant Negative form of EGFR (EGFR-DN) in only the Keratin14-positive basal cells of the skin epithelium to exclude the effect of the drug treatment on other cell types in the skin epithelium (**Manuscript Fig. S9D, E, F**).

In addition, we demonstrated that the ***selective increase of wild-type cell proliferation by deletion of the cell cycle inhibitor p21, already downregulated in Hras^{G12V/+} cells, prevented the expansion of Hras^{G12V/+} cells even in the absence of injury*** (**Manuscript Fig. 5D, E, F, G, Fig. S10**).

Collectively, our data explain behaviorally and molecularly how the competition between wild-type and Hras^{G12V/+} cells is orchestrated during injury-repair to counteract Hras^{G12V/+} cell expansion and abnormal growth.

R2.4) It is not clear how far the suppression of cells with activated RAS extends from the injury (ear punch). From the images in Suppl Fig 2, the impact appears to be at least a couple of mm from the injury. Such an understanding should help inform a search for the mechanism, which appears to be likely mediated by diffusible factors. A quick test of whether these effects are systemic would be to analyze competition in the uninjured ear from the same mouse.

We thank the reviewer for the valuable suggestion and followed this recommendation to identify a possible systemic effect generated by the wound which might influence the competition between wild-type and Hras^{G12V/+} cells in the uninjured contralateral ear. In response to this comment, and comment R1.7 of Reviewer 1, we collected replicates of the **contralateral, unwounded ear from Hras^{G12V/+}-mosaic and wild-type-mosaic mice** that have or have not been wounded. The comparison between those four conditions showed that **systemic effects from the wound do not affect Hras^{G12V/+}/tdTomato^{positive} cell expansion** in the uninjured, contralateral ear of injured Hras^{G12V/+}-mosaic models (**Rebuttal Fig. 5**).

R2.5) This work does not shed much light on the underlying mechanism for how injury promotes mitoses specifically in the WT cells, particularly when in competition with RAS activated cells. Have the authors analyzed

	Injury	
	wild-type cells	Hras ^{G12V/+} cells
Proliferation (Gain of cells)	++	=
Differentiation (Loss of cells)	+	+
	Net gain (compared to WT cells in uninjured ear)	Net loss (compared to Hras ^{G12V/+} cells in uninjured ear)

Table 1. Comparison of proliferation and differentiation rates between wild-type and Hras^{G12V/+} cells in injured Hras^{G12V/+}-mosaic models. The fold change differences in cells positive for phospho-Histone3 (proliferation) and Keratin10 (differentiation) staining in the **Manuscripts Fig. 3H** are used to calculate the gain or loss of cells in Hras^{G12V/+}-mosaic skin epithelium.

gene expression in the WT cells with or without RAS competitors and with or without injury? Are there potential soluble mediators that could be tested as mediators?

We agree with the reviewer that analyzing the effects of potential soluble mediators on the competition between Hras^{G12V/+} and wild-type cells could give further insights on the molecular mechanism. The same interest was shared with reviewer 1 in comment R1.13

To experimentally address this reviewer's comment, we performed single-cell RNA-sequencing (scRNA-seq) of skin cells in the wild-type, Hras^{G12V/+}-mosaic, and Hras^{G12V/+}-max models at day 3 PWI, time point at which the competitive scenario in Hras^{G12V/+} mosaic models changes during injury-repair. Specifically, we performed epidermal cell isolation, which enriches for keratinocytes, while still containing other closely attached cell types (fibroblasts, immune cells etc), that we used to generate 10X Genomics scRNA-seq libraries. Analyses of the resulting datasets were performed as our previous work (Cockburn and Annusver et al., 2021) in close collaborations with Dr Maria Kasper and Karl Annusver (Karolinska Institute) as well as Dr Nur Rahman (Yale).

The generated data sets allowed us to first compare the differentially expressed genes in wild-type (tdTomato^{negative}) basal stem cells in the wild-type-mosaic and in Hras^{G12V/+}-mosaic models to understand if the secretion of soluble factors was influenced by Hras^{G12V/+} neighbors in uninjured and injured context. We did not identify soluble mediators in the differentially expressed gene list of wild-type cells with or without the presence of Ras-competitors and with or without injury. Interestingly, however, after injury, wild-type cells in the Hras^{G12V/+} mosaic respond differently from wild-type cells in the wild-type mosaic. They specifically increased the expression of genes associated with proliferation, growth, and differentiation (**Rebuttal Fig. 11**).

To identify the source of possible soluble mediators that influence epithelial cell behaviors, we first focused on fibroblasts due to the well-known reciprocal communication with epithelial neighbors during injury-repair. The generated data sets allowed us to identify 4 distinct fibroblast clusters, 2 clusters in the uninjured condition and 2 additional clusters that emerge in the injured condition for all the models analyzed. Intriguingly, the observed clustering was not affected by the induction status (mosaic or max) of Hras^{G12V/+} (**Manuscript Fig. S7**). We also did not see model-specific differences between either the overall transcriptional pattern of the injury-repair associated clusters nor in the transcripts involved in deposition/remodeling of extracellular matrix (ECM) and basement membrane (**Rebuttal Fig. 8A, B**).

Next, we looked at soluble factors secreted by fibroblasts that are known to influence epithelial cell behaviors and found that one of the injury-associated fibroblasts clusters displayed significantly upregulated levels of the two EGFR ligands: Hbegf and Ereg compared to other growth factors involved in injury-repair (**Manuscript Fig. S7**). Specifically, the upregulation of Hbegf upon injury was present in all different mouse models analyzed (**Rebuttal Fig. 9**).

Notably, EGFR is one of the best characterized upstream activators of the Ras pathway and previous studies have shown that EGFR signaling promotes epithelial cell proliferation during injury-repair within the wild-type skin epithelium (Chen X, et al., 2019, Berlanga-Acosta J, et al. 2009, Repertinger SK, et al., 2004, Takazawa Y, et al., 2015). In agreement with this, our western blot analyses and quantification from wild-type mice showed increased EGFR activation (phosphoEGFR) in injured versus uninjured conditions (**Manuscript Fig. S9C**).

We expanded our analysis on EGFR ligands to two other populations involved in injury-repair: immune and epithelial cells (**Manuscript Fig. S8**). While immune cells displayed an increase in EGFR ligand expression in a minor number of cells, epithelial cells had a significant increase in Hbegf, Areg, Epgn and Tgf-alpha (**Manuscript Fig S8**), overall appointing fibroblasts and epithelial cells as sources of EGFR ligands during injury-repair. Of note, we also performed ELISA assays on a panel of 32 soluble factors (Bio-Plex-Pro Mouse Cytokine Standard 23-Plax Group I and 9-Plax Group II - Bio-rad). This strategy allowed us to test whether cytokines prevalently associated with inflammation were differentially expressed in injured wild-type, Hras^{G12V/+}-mosaic, and Hras^{G12V/+}-max models. Our data did not show a significant increase in any of the factors analyzed upon injury (**Rebuttal Fig. 10**). Collectively, these data suggested that ***epithelial cell proliferation is regulated by paracrine and autocrine secretion of EGFR ligands.***

To test the role of EGFR ligands in the selective proliferative advantage of wild-type cells in the injured mosaic skin epithelium, we inhibited the EGF-Receptor both chemically and genetically. Treatment with the EGFR inhibitor Gefitinib has been documented to down-regulate downstream targets of EGFR/Ras, such as ERK1/2, in different epithelial models, including the skin epithelium (Buday L, et al., 1993, Ono M, et al., 2004, Amann J, et al., 2005, Chen X, et al., 2019). In contrast, several cancers with persistent activation of the Ras pathways are resistant to EGFR inhibitors, including Gefitinib (Chen J, et al. 2013, Qin B, et al., 2006, Janmaat ML, et al., 2003). Thus, we hypothesized that Gefitinib treatment may lead to inhibition of proliferation specifically in wild-type cells.

We determined the optimal dose of the inhibitor in wild-type-mosaic mice by gavage administration of 100 or 200 mg/kg Gefitinib/body weight, for 6 days, starting treatment 2 days before injury and continuing until 3 days after injury. We found that the 6-day 200 mg/kg of Gefitinib treatment inhibited EGFR and ERK1/2 activation at day-3 post-injury (**Manuscript Fig. S9B**). Additionally, the 200 mg/kg of Gefitinib treatment significantly reduced the cell proliferation during injury-repair of wild-type-mosaic mice, as expected, without increasing cell death events or causing overall weight loss (data not shown).

Next, we treated Hras^{G12V/+}-mosaic models with vehicle (0.5% methylcellulose and 0.2% Tween-80 in water) or 200 mg/kg of Gefitinib, as described above. We found that EGFR inhibition did not affect Hras^{G12V/+} cell proliferation during injury-repair of Hras^{G12V/+}-mosaic mice, consistent with proliferation being independent of EGFR activation due to constitutively active Ras (**Manuscript Fig. 5A and Fig. S9A**). Conversely and excitingly, ***EGFR inhibition selectively reduced wild-type cell proliferation in Hras^{G12V/+}-mosaic models during injury-repair (Manuscript Fig. 5A)***, suggesting that their increased proliferation requires EGFR/Ras activation. This is consistent with our western-blot data that showed a reduced EGFR protein levels in the skin epithelium that expresses constitutively active Hras^{G12V}, as previously shown *in vitro*, which further corroborate the reduced sensitivity of Hras^{G12V/+} cells to injury and EGFR activation (**Manuscript Fig. S9C**, Derer S, et al., 2012).

We also tracked the percentage of surface coverage of Hras^{G12V/+} cells (tdTomato-positive) at 6-, 10-, 17-days post-tamoxifen injection (3-, 7-, 14-days post-injury) (**Manuscript Fig. 5B, C**). Strikingly, we discovered that ***EGFR inhibition, followed by a reduced wild-type cell proliferation post-injury, reestablishes the competitive advantage of Hras^{G12V/+} cells over wild-type neighbors in Hras^{G12V/+}-mosaic models during injury-repair.***

Given that drug treatments do not specifically target the epithelium but can affect additional skin cell types, we complemented with genetic approaches. Specifically, we utilized a tet-inducible dominant negative form of EGFR (TRE-EGFR-DN) that does not express the cytoplasmic tyrosine kinase domain (Roh M, et al., 2001 - JAX:010575), whose expression can be specifically driven in basal stem cell layer of the skin epidermis (Keratin14rtTA). We then combined the doxycycline-inducible EGFR-DN mouse model with the tamoxifen-inducible Hras^{G12V/+} model (K14CreER; K14rtTA; FR-Hras^{G12V/+}; LSL-tdTomato; TRE-EGFR-DN; K14H2B-GFP). Our quantifications showed that the expression of EGFR-DN selectively inhibited the proliferation of wild-type cells, but not of Hras^{G12V/+} cells, in Hras^{G12V/+}-mosaic models after injury, consistent with the Gefitinib treatment (**Manuscript Fig. S9D**). To assess how EGFR-DN affects cell competition during injury-repair, we tracked the percentage of surface coverage of Hras^{G12V/+} cells (GFP+/tdTomato+) at 3-, 7- and 14-days PWI (6-, 10-, 17- days

post-tamoxifen-induced mosaicism). Strikingly, we discovered that the specific EGFR inhibition in the basal stem cell compartment reestablishes the competitive advantage of Hras^{G12V/+} cells over their wild-type neighbors during injury-repair of mosaic mice (**Manuscript Fig. S9E, F**). Collectively, these data further confirmed the result obtained with the inhibition of EGFR by Gefitinib treatment (**Manuscript Fig. 5A, B, C**).

All together, these experiments demonstrate that *EGFR-mediated wild-type cell proliferation counteracts the expansion of Hras^{G12V/+} cells in mosaic skin after injury*.

Collectively, our data demonstrate that the activation of the EGFR/Ras signaling pathway through the secretion of EGFR ligands from fibroblasts and epithelial cells is required for the selective increase in wild-type cell proliferation to prevent Hras^{G12V/+} cell expansion in Hras^{G12V/+}-mosaic models upon injury.

R2.6) It is also not clear why the p21 null competitors, which have about 2.5X greater mitotic indices than the p21 null HRAS mutant cells, do not have a competitive advantage. In addition, we do not know if p21 is playing a role in the reversal of the fitness advantage of mutant cells following injury – for example, is p21 lost post-injury in the WT cells?

We thank the reviewer for this important point. We observed a significant increase in cell proliferation of wild-type cells compared to Hras^{G12V/+} neighbors in Hras^{G12V/+}-p21^{null}-mosaic models (**Manuscript Fig. 5D, E**). To confirm this result, we increased the number of sample analyzed of p-H3+ staining in Hras^{G12V/+}-p21^{null}-mosaic models considering the high variability in the number of proliferative cells that we have already reported for the mitotic figure analysis in the manuscript (**Manuscript Fig. 5D**). The new replicates confirm a higher proliferation capacity of the tdTomato-negative (wild-type) population than the tdTomato-positive (Hras^{G12V/+}) cells in the Hras^{G12V/+}-p21^{null}-mosaic model. This allowed us to mimic the increase of wild-type cell proliferation in the uninjured context and highlight its key role in preventing Hras^{G12V/+} cell expansion (**Manuscript Fig. 5E**).

To investigate further the absence of tdTomato-negative (wild-type) cells expansion in Hras^{G12V/+}-p21^{null}-mosaic models despite their proliferative advantage, we analyzed the differentiation capacity. This cell behavior could be accelerated by the loss of function of p21. Indeed, we did not see a significant increase in the thickness of the epidermis in p21^{null} models despite the high proliferative potential of p21^{null} epithelial cells and it has been shown that p21 is highly expressed in the differentiated suprabasal cells (Haensel et al., 2020).

To evaluate possible changes in rate of differentiation we immunolabeled for the early differentiation marker Keratin10 (Schweizer et al., 1984; Braun et al., 2003, Cockburn and Annusver et al., 2021). Comparison of Keratin10-positive cells in uninjured Hras^{G12V/+}/p21^{null}-mosaic and Hras^{G12V/+}-mosaic mice showed that p21 loss increased the differentiation rate of wild-type cells, similarly at what we observed for proliferation (**Manuscript Fig. S9N**).

Considering the selective response of wild-type cells to p21 loss, we hypothesized that Hras^{G12V/+} cells downregulate p21 expression to promote the proliferative advantage in the uninjured condition. To this end, we quantified the levels of p21 mRNA (newly generated scRNA-seq datasets, **Manuscript Fig. S4, Fig. S10C, D, E**) and protein (epidermal preparation immunostaining, **Manuscript Fig. S10A, B**) in the basal stem cell layer of uninjured Hras^{G12V/+} and wild-type-mosaic models. We showed that both p21 mRNA and protein were significantly downregulated in Hras^{G12V/+}-mosaic models compared to wild-type-mosaic models (**Manuscript Fig. S10B, C**). Furthermore, we compared *non-cycling* tdTomato+ cells in the Hras^{G12V/+}-mosaic and wild-type-mosaic models to exclude cycling cells, which physiologically downregulate p21 expression. We observed the same significant difference in p21 expression levels between non-cycling wild-type cells in the wild-type-mosaic and non-cycling mutant cells in the Hras^{G12V/+}-mosaic (**Manuscript Fig. S9D, E**). These data confirm that Hras^{G12V/+} cells express a lower level of p21, and that knocking out p21 in all cells is an effective approach to selectively manipulate wild-type cells.

Following the reviewer's suggestion, we also documented a downregulation of p21 expression in the proliferative zone around the wound compared to the uninjured condition which we attribute to the increased

proliferative demand during injury-repair (**Rebuttal Fig. 8**). These data further demonstrate how p21 expression is modulated to allow for the proliferative response required to heal the wound.

These insights, further expand the significance of the p21^{null} experiments we had in our manuscript. Specifically, the low level of p21 expression in Hras^{G12V/+} cells allowed us to explore a competition scenario wherein *p21 loss in the wild-type cells of Hras^{G12V/+}-mosaic models selectively increased their proliferation and differentiation, mimicking injury-repair behaviors and resulting in the prevention of Hras^{G12V/+} cell expansion even without injury* (Manuscript Fig. 5D, E, F, G).

R2.7) While MAPK activity is promoted by p21 loss, injury and HRAS activation, they do not show whether this MAPK induction is contributing to injury promoted alterations in the competitive dynamics. While it seems likely that MAPK increases are important for the competitive advantage of RAS activated cells in uninjured tissue, it is not shown whether MAPK plays a role in injury-induced promotion of WT competitor fitness.

We thank the reviewer for this important point. To examine a potential role of ERK1/2 activation in the competition between wild-type and Hras^{G12V/+} cells, we performed phosphoERK1/2 immunostaining in Hras^{G12V/+}-mosaic epidermis in both uninjured and injured ears.

In the uninjured condition, we observed a higher number of phosphoERK1/2 positive cells in Hras^{G12V/+} clones compared to wild-type ones supporting the competitive and proliferative advantage of Hras^{G12V/+} cells over wild-type cells, as the reviewer mentioned (**Manuscript Fig S11C, D**). Interestingly, during injury-repair, phosphoERK1/2 immunostaining showed that the number of Hras^{G12V/+} cells positive for phosphoERK1/2 was similar in both uninjured and injured scenarios showing that Hras^{G12V/+} cells have already maximized their ERK1/2 activation before injury consequently limiting their ability to increase proliferation during injury-repair. On the other hand, the number of wild-type cells positive for phosphoERK1/2 increased together with the proliferative events upon injury (**Manuscript Fig S11C, D**). Collectively, the new phosphoERK1/2 immunostaining data at the single cell level in Hras^{G12V/+}-mosaic model supported our western-blot data in the original manuscript (**Manuscript Fig S11A**). Furthermore, we investigated the activation of another Ras-downstream pathway PI3K/AKT involved in cell proliferation, and it was not significantly affected by injury-repair (**Manuscript Fig. S11B**).

This reviewer asks whether ERK1/2 plays a role to promote the proliferative advantage of wild-type cells during injury. Independent lines of experimentation to address additional comments have led us to demonstrate the key role of EGFR signaling pathway which is the best characterized upstream activators of the Ras pathway and its downstream effector ERK1/2. In agreement with this, we showed that EGFR inhibitor Gefitinib repressed the activity of EGFR and ERK1/2 during injury-repair of wild-type-mosaic mice (**Manuscript Fig S9A, B**). EGFR pathway is also known to promote epithelial cell proliferation during injury-repair within the wild-type skin epithelium (Chen X, et al., 2019, Berlanga-Acosta J, et al., 2009, Repertinger SK, et al., 2004, Li B, et al., 2021). On the contrary, Hras^{G12V/+} cells are less dependent on the upstream activation of EGFR for Ras signaling, due to their expression of constitutively active Hras^{G12V} (**Manuscript Fig. S9A**). We now provide western blot analyses that showed an injury-induced activation of EGFR (phosphoEGFR) in wild-type-mosaic models, as expected from prior studies (**Manuscript Fig. S9C**, Takazawa Y, et al., 2015). In addition, the levels of total EGFR are reduced in Hras^{G12V/+} cells, further explaining their reduced sensitivity to EGFR activation (**Manuscript Fig. S9C**).

Thus, *the release of EGFR ligands upon wounding selectively promotes the proliferation of wild-type cells because they can activate EGFR-Ras signaling whereas Hras^{G12V/+} are already maximally activated*. Moreover, EGFR inhibition by Gefitinib treatment or the expression in the basal stem cell compartment of the EGFR dominant negative (DN) selectively reduced wild-type cell proliferation during injury-repair by preventing the activation of EGFR signaling in response to EGFR ligands. Conversely, Hras^{G12V/+} cell proliferation was unaffected by Gefitinib treatment or the EGFR-DN expression because it is less sensitive to EGFR activation (**Manuscript Fig. 5A, Fig. S9C, D**). Therefore, Hras^{G12V/+} cells re-acquired a competitive advantage over wild-type neighbors in Hras^{G12V/+}-mosaic models treated with Gefitinib or expressing EGFR-DN (**Manuscript Fig. 5B, C, Fig. S9E, F**).

Collectively, we identify the differential activation of ERK1/2 through EGFR/Ras signaling as the molecular pathway responsible for the selective increase of wild-type cell proliferation and their resulting ability to *counteract the expansion of Hras^{G12V/+} cells in mosaic skin after injury*.

Minor:

1) Fig 3: It is interesting that WT competitor cell proliferation appears further boosted by the presence of HRAS mutant competitors in the injured setting, which was apparently not seen when both competing populations are WT. However, this conclusion needs to be substantiated by statistical comparisons [compare day 6 WT mitotic indices (green) in Fig 3g and 3h].

We thank the reviewer for pointing out the lack of a clear statistical comparison between wild-type-mosaic models and Hras^{G12V/+}-mosaic models during injury repair. We changed Figure 3C and D (former Figure 3G and H) to allow for the comparison of tdTomato^{positive} and ^{negative} cell proliferation between wild-type and Hras^{G12V/+}-mosaic models in both uninjured and injured conditions. We also applied the appropriate statistical analysis described in the figure legend. However, the new graphical representation still substantiated our conclusion. The initially elevated wild-type cell proliferation observed in the injured Hras^{G12V/+}-mosaic model decreased over time, and eventually looked similar to Hras^{G12V/+} neighbors, but still greater than wild-type cell proliferation after injury of the wild-type-mosaic model (**Manuscript Fig. 3C, D**).

2) A more comprehensive study of mutations in normal skin (from the same group as their reference #1) is the recent Fowler et al paper (4).

1. Liu N, Matsumura H, Kato T, Ichinose S, Takada A, Namiki T, et al. Stem cell competition orchestrates skin homeostasis and ageing. *Nature*. 2019;568(7752):344-50.
2. Ballesteros-Arias L, Saavedra V, Morata G. Cell competition may function either as tumour-suppressing or as tumour-stimulating factor in *Drosophila*. *Oncogene*. 2014;33(35):4377-84.
3. Kajita M, Fujita Y. EDAC: Epithelial defence against cancer-cell competition between normal and transformed epithelial cells in mammals. *Journal of biochemistry*. 2015;158(1):15-23.
4. Fowler JC, King C, Bryant C, Hall M, Sood R, Ong SH, et al. Selection of oncogenic mutant clones in normal human skin varies with body site. *Cancer Discovery*. 2020:CD-20-1092.

We thank the reviewer for indicating another recent study on cell competition in normal human skin epithelium. We added Fowler et al., 2020 reference in the introduction of the manuscript.

Referee #3 (Remarks to the Author):

Expertise skin biology, injury, epithelial homeostasis

R3.1) This manuscript reports an interesting observation of potential important pathological implication. Cancer is often compared to a non-healing wound, and wounding/non healing conditions may accelerate e.g. skin cancer. Indeed, e.g. in dystrophic epidermolysis bullosa patients, who carry mutations in collagen VII, the continuous induction of skin blistered promotes a chronic wound conditions, including extensive inflammation, and this has been linked to SCC in these patients, with most patients dying of invasive skin cancer before their 50s. Moreover, wounding may accelerate overgrowth and tumor formation in mouse models with a homogenous expression in which the oncogene, often H-RasV12, is homogeneously expressed, although it is important to note that this may depend where in the epidermis cells are expressed, as early studies suggested that suprabasal expression of mutant Ras using K1 or K10 promoters accelerate tumor formation near

sites of scratching or wounding whereas no such relation was observed when expressed basally under K5 conditions (Brown et al., Current Biology 1998).

We appreciate the request to reconcile what we discovered in the skin epidermis with previous results in different transgenic mice expressing a mutated form of Hras. As mentioned by the reviewer, the transgenic mice expressing the Hras mutation under the promoter of Keratin10 developed papillomas at sites of injury (Bailleu, et al., 1990). In Brown et al., 1998, the expression of the Hras oncogene is under the control of a truncated form of the Keratin5 promoter which is activated along with the Keratin14 promoter. This leads to the development of papilloma and SCCs in areas that are not associated with injuries. Interestingly, literature from our and other lab showed that the basal layer of the epidermis contains a large fraction of Keratin14+/Keratin10+ cells (which are targeted by our Keratin14CreER model), which are primed and in the following days renew the suprabasal layers. As all delaminating cells (that produce the suprabasal layers) go through a Keratin10+ state, the suprabasal layer will consequently reach nearly a homogeneous setting (Schweizer et al., 1984, Cockburn and Annusver et al., 2021).

REDACTED

The differences observed in the localization of the tumors in regard to injury between the Keratin5 model in Bailleu, et al., 1990 and our Keratin14 model motivated us to highlight two different mouse models relevant to our study: the embryonic and constitutive expression of Hras^{G12V/+} (the *K14Cre* model) and the post-natal conditional expression of Hras^{G12V/+} upon tamoxifen injection (the *K14CreER* model, used in this manuscript) in the mouse skin epithelium. *K14Cre/FR-Hras^{G12V/+}* models develop premature sporadic papillomas similar to the published *Keratin5* model (Brown et al., 1998, **Rebuttal Fig. 12**). This model does not allow the visualization of the cooperation between injury and Hras^{G12V/+}, considering the early and sporadic appearance of aberrant growths, caused by an altered skin development at the embryonic stage. In contrast, our conditional model allows us to express Hras^{G12V/+} at a physiological level and in adult mice, a situation which mimics the appearance of somatic mutations in a temporal manner. Therefore, we would like to reiterate that both the works mentioned by the reviewer used transgenic mice expressing Ras mutation constitutively from embryonic development to adulthood at a supraphysiological level, scenarios which make it difficult to address which cells of the epidermis are competent to give rise to tumors.

R3.2) In contrast, normal skin tolerates the presence of epidermal cells with carcinogenic mutations without apparent tumor formation, even if underlying mechanisms are still unclear. The authors now asked whether acute wounding would promote the outgrowth of these cells when in a mosaic condition with wt cells. Interestingly, using either H-RASG12V or the more aggressive K-RasG12 mutations, wounding of the ear under mosaic conditions does not result in further overgrowth of mutant cell, unlike when there is over 95% of mutant cells present. The authors then go on and show that unlike the mutant cells, wt cells show a strong increase in proliferation upon wounding, most pronounced after 3 days after wounding, whereas mutant cells show a similar proliferation profile in non-wounded versus wounded conditions, thus suggesting that wt cells early after wounding “outcompete” mutant cells. Using a p21 null mutant that releases cell cycle inhibition, they then ask whether induction of proliferation in wt cells is sufficient to prevent outgrowth of mutant cells under non-wounding conditions, and find that this is indeed sufficient. Finally they link this to, not very surprising, MAP kinase signaling.

Overall, these are potential interesting albeit rather preliminary findings that do not yet convincingly show that increased proliferation of wt cells is indeed key to suppress overgrowth of mutant cells under wounding conditions. They only show that increasing proliferation in wt cells is sufficient to keep mutant cell expansion in

check under non-wounding conditions, showing that under these conditions mutant cell overgrowth is driven by a proliferative advantage of these cells. This observation although interesting and nicely fitting with the correlative data on the proliferation measurements after wounding, is insufficient to show that this mechanism also keeps mutant cells in check after wounding. This would require (and I agree not a trivial task) the reverse experiment and show that suppression of wt cell growth upon wounding results in outgrowth of mutant cells under mosaic conditions.

We thank this reviewer for the thoughtful perspective that was also shared with Reviewer 1 and Reviewer 2 in the comments 1.11, 1.13 and 2.5. We agree that a selective inhibition of wild-type proliferation during injury-repair would further support our model. To begin dissecting the mechanism behind the selective increase in wild-type cell proliferation during injury-repair, we performed single-cell RNA-sequencing (scRNA-seq) of skin cells in the wild-type, $Hras^{G12V/+}$ -mosaic, and $Hras^{G12V/+}$ -max models at day 3 PWI, time point at which the competitive scenario in $Hras^{G12V/+}$ mosaic models changes during injury-repair. Specifically, we performed epidermal cell isolation, which enriches for keratinocytes, while still containing other closely attached cell types (fibroblasts, immune cells etc), that we used to generate 10X Genomics scRNA-seq libraries. Analyses of the resulting datasets were performed as our previous work (Cockburn and Annusver et al., 2021) in close collaborations with Dr Maria Kasper and Karl Annusver (Karolinska Institute) as well as Dr Nur Rahman (Yale). Focusing on fibroblasts, we identified 4 distinct fibroblast clusters, 2 clusters in the uninjured condition and 2 additional clusters that emerge in the injured condition for all the models analyzed. Intriguingly, the observed clustering was not affected by the induction status (mosaic or max) of $Hras^{G12V/+}$ (**Manuscript Fig. S7**). We also did not see model-specific differences between either the overall transcriptional pattern of the injury-repair associated clusters nor in the transcripts involved in deposition/remodeling of extracellular matrix (ECM) and basement membrane (**Rebuttal Fig. 8A, B**).

Next, we looked at soluble factors secreted by fibroblasts that are known to influence epithelial cell behaviors and found that one of the injury-associated fibroblast clusters displayed significantly upregulated levels of the two EGFR ligands: Hbegf and Ereg compared to other growth factors involved in injury-repair (**Manuscript Fig. S7**). Specifically, the upregulation of Hbegf upon injury was present in all different mouse models analyzed (**Rebuttal Fig. 9**).

Notably, EGFR is one of the best characterized upstream activators of the Ras pathway and previous studies have shown that EGFR signaling promotes epithelial cell proliferation during injury-repair within the wild-type skin epithelium (Chen X, et al., 2019, Berlanga-Acosta J, et al. 2009, Repertinger SK, et al., 2004, Takazawa Y, et al., 2015). In agreement with this, our western blot analyses and quantification from wild-type mice showed increased EGFR activation (phosphoEGFR) in injured versus uninjured conditions (**Manuscript Fig. S9C**).

We expanded our analysis on EGFR ligands to two other populations involved in injury-repair: immune and epithelial cells (**Manuscript Fig. S8**). While immune cells displayed an increase in EGFR ligand expression in a minor number of cells, epithelial cells had a significant increase in Hbegf, Ereg, Epgn and Tgf-alpha (**Manuscript Fig S8**), overall appointing fibroblasts and epithelial cells as sources of EGFR ligands during injury-repair. Of note, we also performed ELISA assays on a panel of 32 soluble factors (Bio-Plex-Pro Mouse Cytokine Standard 23-Plax Group I and 9-Plax Group II - Bio-rad). This strategy allowed us to test whether cytokines prevalently associated with inflammation were differentially expressed in injured wild-type, $Hras^{G12V/+}$ -mosaic, and $Hras^{G12V/+}$ -max models. Our data did not show a significant increase in any of the factors analyzed upon injury (**Rebuttal Fig. 10**). Collectively, these data suggested that ***epithelial cell proliferation is regulated by paracrine and autocrine secretion of EGFR ligands.***

To test the role of EGFR ligands in the selective proliferative advantage of wild-type cells in the injured mosaic skin epithelium, we inhibited the EGF-Receptor both chemically and genetically. Treatment with the EGFR inhibitor Gefitinib has been documented to down-regulate downstream targets of EGFR/Ras, such as ERK1/2, in different epithelial models, including the skin epithelium (Buday L, et al., 1993, Ono M, et al., 2004,

Amann J, et al., 2005, Chen X, et al., 2019). In contrast, several cancers with persistent activation of the Ras pathways are resistant to EGFR inhibitors, including Gefitinib (Chen J, et al. 2013, Qin B, et al., 2006, Janmaat ML, et al., 2003). Thus, we hypothesized that Gefitinib treatment may lead to inhibition of proliferation specifically in wild-type cells.

We determined the optimal dose of the inhibitor in wild-type-mosaic mice by gavage of 100 or 200 mg/kg Gefitinib/body weight, for 6 days, starting treatment 2 days before injury and continuing until 3 days after injury. We found that the 6-day 200 mg/kg of Gefitinib treatment inhibited EGFR and ERK1/2 activation at day-3 post-injury (**Manuscript Fig. S9B**). Additionally, the 200 mg/kg of Gefitinib treatment significantly reduced the cell proliferation during injury-repair of wild-type-mosaic mice, as expected, without increasing cell death events or causing overall weight loss (data not shown).

Next, we treated Hras^{G12V/+}-mosaic models with vehicle (0.5% methylcellulose and 0.2% Tween-80 in water) or 200 mg/kg of Gefitinib, as described above. We found that EGFR inhibition did not affect Hras^{G12V/+} cell proliferation during injury-repair of Hras^{G12V/+}-mosaic mice, consistent with proliferation being independent of EGFR activation due to constitutively active Ras (**Manuscript Fig. 5A and Fig. S9A**). Conversely and excitingly, ***EGFR inhibition selectively reduced wild-type cell proliferation in Hras^{G12V/+}-mosaic models during injury-repair (Manuscript Fig. 5A)***, suggesting that their increased proliferation requires EGFR/Ras activation. This is consistent with our western-blot data that showed a reduced EGFR protein levels in the skin epithelium that expresses constitutively active Hras^{G12V}, as previously shown *in vitro*, which further corroborate the reduced sensitivity of Hras^{G12V/+} cells to injury and EGFR activation (**Manuscript Fig. S9C**, Derer S, et al., 2012).

We also tracked the percentage of surface coverage of Hras^{G12V/+} cells (tdTomato-positive) at 6-, 10-, 17-days post-tamoxifen injection (3-, 7-, 14-days post-injury) (**Manuscript Fig. 5B, C**). Strikingly, we discovered that ***EGFR inhibition, followed by a reduced wild-type cell proliferation post-injury, reestablishes the competitive advantage of Hras^{G12V/+} cells over wild-type neighbors in Hras^{G12V/+}-mosaic models during injury-repair.***

Given that drug treatments do not specifically target the epithelium but can affect additional skin cell types, we complemented with genetic approaches. Specifically, we utilized a tet-inducible dominant negative form of EGFR (TRE-EGFR-DN) that does not express the cytoplasmic tyrosine kinase domain (Roh M, et al., 2001 - JAX:010575), whose expression can be specifically driven in basal stem cell layer of the skin epidermis (Keratin14rtTA). We then combined the doxycycline-inducible EGFR-DN mouse model with the tamoxifen-inducible Hras^{G12V/+} model (K14CreER; K14rtTA; FR-Hras^{G12V/+}; LSL-tdTomato; TRE-EGFR-DN; K14H2B-GFP). Our quantifications showed that the expression of EGFR-DN selectively inhibited the proliferation of wild-type cells, but not of Hras^{G12V/+} cells, in Hras^{G12V/+}-mosaic models after injury, consistent with the Gefitinib treatment (**Manuscript Fig. S9D**). To assess how EGFR-DN affects cell competition during injury-repair, we tracked the percentage of surface coverage of Hras^{G12V/+} cells (GFP+/tdTomato+) at 3-, 7- and 14-days PWI (6-, 10-, 17- days post-tamoxifen-induced mosaicism). Strikingly, we discovered that the specific EGFR inhibition in the basal stem cell compartment reestablishes the competitive advantage of Hras^{G12V/+} cells over their wild-type neighbors during injury-repair of mosaic mice (**Manuscript Fig. S9E, F**). Collectively, these data further confirmed the result obtained with the inhibition of EGFR by Gefitinib treatment (**Manuscript Fig. 5A, B, C**).

All together, these experiments demonstrate that ***EGFR-mediated wild-type cell proliferation counteracts the expansion of Hras^{G12V/+} cells in mosaic skin after injury.***

Collectively, our data demonstrate that the activation of the EGFR/Ras signaling pathway through the secretion of EGFR ligands from fibroblasts and epithelial cells is required for the selective increase in wild-type cell proliferation to suppress Hras^{G12V/+} cell expansion in Hras^{G12V/+}-mosaic models upon injury.

R3.3) Most importantly, the authors do not carefully take into account the role of several other key differences in terms of tissue confinement, tissue structure, and inflammation in normal versus wounded skin, which may play an essential role in how wounding impacts how control cells keep mutant cells in check: (a) As they make a full thickness wound they basically break the basement membrane, an important barrier in

keeping cancer cells at check. Proper healing also requires restoration of epidermal tissue confinement, by secretion of ECM molecules, adhesion to these ECM and ultimately basement membrane reformation that restores tissue compartments. The ability to repair this mechanical and signaling barrier or not repair this barrier and restoring determined by the presence of a sufficient number of wt cells may therefore be essential to prevent outgrowth and ingrowth into the dermis, which is basically what the mutant max cells are doing.

(b) along the same lines, mosaic versus mostly mutant Ras cells may impact on fibroblasts and on the dermal extracellular matrix rearrangements necessary for proper healing that in return also impacts on the epidermis. (b) ruling out the role of Inflammation that may be different upon wounding (as well as a major driver of the overgrowth or not) depending on whether the tissue is wt, mosaic or has mostly Ras mutant cells (Ras max).

We agree with the reviewer's comment that interrogating the role of both fibroblasts and immune cells in the phenotype observed is a relevant point. As we mentioned in the comment above 3.2, we performed scRNA-seq of skin cells in the wild-type, Hras^{G12V/+}-mosaic, and Hras^{G12V/+}-max models at day 3 PWI, time point at which the competitive scenario in Hras^{G12V/+} mosaic models changes during injury-repair. This experiment allowed us to isolate epidermal cells and associated fibroblasts and immune cells and interrogate their molecular signatures.

We identified four different fibroblast clusters: 2 clusters in the uninjured condition and 2 additional clusters that emerge during injury-repair for wild-type, Hras^{G12V/+}-mosaic, and Hras^{G12V/+}-max models. Intriguingly, the observed clustering was not affected by the induction status (mosaic or max) of Hras^{G12V/+} (**Manuscript Fig. S7**). To address the reviewer question, we examined the overall transcriptional pattern of the injury-repair associated fibroblast clusters and specific transcripts involved in secretion and remodeling of ECM molecules, adhesion to ECM and basement membrane reformation but we did not find any significant difference between all the models analyzed (**Rebuttal Fig. 8A, B**). Furthermore, the expression of soluble factors that are involved in ECM deposition such as TGF-beta was not significantly different in any of the models analyzed (**Rebuttal Fig. 9**).

Our generated scRNA-seq datasets allowed us to also analyze immune cells. We found distinct immune cell clusters in all the conditions analyzed based on their distinctive molecular markers (**Manuscript Fig. S8**, Lie Y et al., 2020). Neutrophils and macrophages were only present in injured samples, as expected. Interestingly, the observed clustering was not affected by the induction status (mosaic or max) of Hras^{G12V/+} (**Manuscript Fig. S8**). To identify an underlying role for inflammation in the competition between Hras^{G12V/+} and wild-type cells during injury-repair, we performed ELISA assays on a panel of 32 soluble factors (Bio-Plex-Pro Mouse Cytokine Standard 23-Plax Group I and 9-Plax Group II - Bio-rad). This strategy allowed us to test whether cytokines prevalently associated with inflammation were differentially expressed in injured wild-type, Hras^{G12V/+}-mosaic, and Hras^{G12V/+}-max models. Our data did not show a significant increase in any of the factors analyzed upon injury (**Rebuttal Fig. 10**).

Collectively, we *did not discover any different molecular features of fibroblasts and immune cells upon wounding that depend on whether the tissue is wild-type, Hras^{G12V/+}-mosaic or -max* that could suggest a possible involvement in the oncogenic overgrowth that we observed in the Hras^{G12V/+}-max model.

R3.4) although the authors rule out the role of cell death, they only score cell death at day 3, whereas the readout for tissue thickness as measured to average distance to collagen is done at day 14. Cell death may thus also play an important role later in the process.

We agree with the reviewer that cell death could peak after the acute phase of injury-repair. To determine cell death events in both wild-type and Hras^{G12V/+} cells at later time points during injury-repair, we quantified the nuclear fragmentation events at day-3, day-7, day-14 and 1-month post-injury induction (**Manuscript Fig. S3F, G**). This analysis showed that *cell death does not significantly contribute to wild-type and Hras^{G12V/+} cell competition during injury-repair*.

R3.5) Importantly for all of the above points, it seems to me, as also indicated in many reviews, that the mutant Ras cells are in a state of continuous activation, and in that sense behave as a “non-healing” activated keratinocytes (hence their Map kinase activity is different because sustained activation), whereas the wt cells are still able to respond to wounding. These different states may actually evoke a non-healing response in mutant Ras max conditions versus mosaic conditions, where control cells may advance healing conditions including but very likely not exclusively driven by the ability to proliferate but also by controlling inflammatory responses and coordinating the ability to restore homeostatic tissue architecture.

We thank the reviewer for their thoughtful advice. To determine whether $Hras^{G12V/+}$ -mosaic and $Hras^{G12V/+}$ -max models have a different immune response after wounding, we analyzed immune cells and cytokine expression, as we mention in the comment 3.3 above. We used the newly generated scRNA-seq datasets to identify distinct immune cell clusters in all the conditions (**Manuscript Fig. S8**, Lie Y et al., 2020). When we compared $Hras^{G12V/+}$ -max or $Hras^{G12V/+}$ -mosaic with their respective control treated with the same dose of tamoxifen, we did not observe the appearance of additional immune cell clusters. We also performed ELISA assays on a panel of 32 soluble factors (Bio-Plex-Pro Mouse Cytokine Standard 23-Plax Group I and 9-Plax Group II - *Bio-rad*). This strategy allowed us to search for cytokines differentially expressed in injured wild-type, $Hras^{G12V/+}$ -mosaic, and $Hras^{G12V/+}$ -max models that could underlie a different immune response upon wounding. Interestingly, all the cytokines detected in the assays were not significantly different in any of the conditions analyzed (**Rebuttal Fig. 10**). Collectively, we **did not discover any difference in the inflammatory profile in wild-type, $Hras^{G12V/+}$ -mosaic or -max** during injury-repair.

REDACTED

We would like to also add that we do not think that $Hras^{G12V/+}$ cells in the $Hras^{G12V/+}$ -max model are in a “non-healing” state. In the wounded $Hras^{G12V/+}$ -max model, we observed extra differentiated and basal layers that participate in creating a thicker architecture that can be appreciated from the H&E staining in **Manuscript Fig. S1D**. This phenotype is typically associated with longer survival of differentiating cells along the journey, allowing the duplication of various suprabasal layers. To further support this point, we also documented that $Hras^{G12V/+}$ cells in $Hras^{G12V/+}$ -max increase both proliferation and differentiation by measuring the expression of markers of proliferation and differentiation, phospho-Histone3 and Keratin10 in the basal stem cell layer (**Rebuttal Fig. 13**).

In the $Hras^{G12V/+}$ -mosaic model, we showed that $Hras^{G12V/+}$ cells maintain an unchanged level of proliferation before and after the wound, but our analyses of Keratin10 expression and GO-terms of the scRNA-sequencing data indicated that $Hras^{G12V/+}$ cells increased their differentiation ability upon wounding. This behavior could contribute to the containment of the expansion of $Hras^{G12V/+}$ cells and accommodate their hyperproliferative wild-type neighbors (**Manuscript Fig. S5E, Fig. S3H, Rebuttal Table 1**). Similar conclusions on increase differentiation of $Hras^{G12V/+}$ cells were drawn in Sandoval et al., 2021 during the establishment of oncogenic tolerance in the uninjured context in the same experimental condition. Furthermore, when we chemically and genetically inhibited EGFR in the $Hras^{G12V/+}$ -mosaic model with Gefitinib treatment or the expression of EGFR-DN in the basal stem cells layer (*for more details please see the comment above R3.2*), we found a selective inhibition of wild-type cell proliferation but not $Hras^{G12V/+}$ cell proliferation. This was followed by a reestablishment of **the competitive advantage of $Hras^{G12V/+}$ cells over wild-type neighbors in $Hras^{G12V/+}$ -mosaic models during injury-repair** (Fig. 5A, B, C Fig. S9D, E, F). This demonstrates that $Hras^{G12V/+}$ cells are

capable of responding to injury with increased proliferation when released from the competition with wild-type neighbors.

R3.6) Is this a general phenomena regardless of the local differences in skin architecture between e.g ear, tail or back skin? As a proof of more general principle does wounding of mosaic mice also suppresses outgrowth when wounded in the back skin, which has more HFs and may thus result in altered proliferative response (and likely a somewhat different wounding response also in terms of inflammation)? The authors do not have to do any in vivo tracing but could show an endpoint in terms of thickness.

We thank the reviewer for their comment, and we addressed it by following the same experimental design adopted for the injury in the ear and performed a 4 mm punch biopsy in the back skin of both Hras^{G12V/+}-mosaic and -max models. At 14 days post-wound induction we observed abnormal oncogenic growth only in Hras^{G12V/+}-max models, as we documented for the wound in the ear (**Manuscript Fig. S1F, G, H, I**). Thus, despite of the local differences in the back and in the ear skin, wound promotes oncogenic growth only when nearly all the basal stem cell layer expressed Hras^{G12V/+}.

R3.7) The real competitive difference of proliferation of wt above Ras mutant under mosaic conditions is only seen at Day 3 (2-fold), after that the two have very similar proliferative curves (Fig.3G). Can this 2-fold short term proliferative advantage explain clone sizes, using modeling?

We agree with this assessment. We showed that wild-type cells become more competitive in their proliferative potential at 3 days post-injury but at later time points Hras^{G12V/+} and wild-type cells have equal proliferation that is consistent with a relatively steady size of wild-type and Hras^{G12V/+} clones during injury-repair.

To understand, how the transient increase of wild-type cell proliferation did not result in the outcompetition of Hras^{G12V/+} population at 6 days post-tamoxifen/3 days post-injury we directly analyzed other cellular behaviors that influence the size of wild-type and Hras^{G12V/+} clones in the basal stem cell layer. We agree that using modeling would have been elegant to test if this 2-fold short term proliferative advantage explain clone sizes, which we certainly considered. Unfortunately, ongoing difficulties of the pandemic and several technical setbacks during the revision time did not allow us to acquire the (extensive) number of samples and wound-conditions that would have been necessary given the complex balances of behaviors in a wound context (including daily revisit with the two-photon and immunostaining for behavioral markers). Instead, we performed in-depth analysis on angle of cell division, cell death and differentiation, which were highly informative.

Differentiation and cell death are mechanisms of cell loss that could influence the wild-type and Hras^{G12V} cell competition in the skin epidermis. First, we examined cell death by scoring for either nuclear fragmentation events or expression of an apoptotic marker, active-Capase3. The overall frequency of apoptosis was low, and we did not observe significant differences in cell death events of wild-type or Hras^{G12V/+} cells in mice with or without injury at 6 days post-tamoxifen and at later-time points (**Manuscript Fig. 3E, F, Fig S3D, E, F, G**).

Then, to test whether injury-repair leads to changes in the differentiation capacity of wild-type and Hras^{G12V/+} cells, we interrogated the expression of early differentiation markers by both protein and scRNA-sequencing. We discovered that both Hras^{G12V/+} and wild-type cells increase their rate of differentiation after injury (**Manuscript Fig. S3H, S5C**). We infer that ***the increase in differentiation rate of both wild-type and Hras^{G12V/+} cells after injury accommodates the increase in wild-type cell proliferation in the basal layer (Rebuttal Table 1)***. Consequently, ***during injury-repair, wild-type cells ultimately prevent the expansion of Hras^{G12V/+} clones observed in the uninjured skin epithelium***. During injury-repair, ***wild-type cells have an increased contribution in replenishing cells lost to differentiation, constraining the expansion of Hras^{G12V/+} cells and allowing efficient repair of the wound***. To comprehensively analyze the differentiation capacity, we quantified Keratin10 expression at day 7 post-injury in Hras^{G12V/+}-mosaic models and showed that both wild-type and Hras^{G12V/+} cells have the same differentiation rate, which in addition to similar proliferative capacities

justify the maintenance of the mutant and wild-type clone sizes at later time points during injury-repair (**Manuscript Fig. S31**).

We also quantified the mode of cell division of the basal stem cells in the skin epidermis, to determine its role in affecting the clone size and the competition between wild-type and Hras^{G12V/+} cells in the Hras^{G12V/+}-mosaic model. To achieve this goal, we measured the angle of division relative to the basal stem cell layer of tdTomato+ and tdTomato- cells in Hras^{G12V/+}-mosaic and wild-type-mosaic models, 6-days after tamoxifen-induced mosaicism, in both uninjured and post-injury settings (**Fig. 14A, B rebuttal**). The cell division orientation relative to the basal stem cell layer ranges from parallel to perpendicular, reflecting retention or exit from the stem cell pool, respectively. Our quantifications based on scoring of anaphases and telophases showed that ***all the cell divisions are planar*** (0-45°) in Hras^{G12V/+}-mosaic models with or without injury, with an average angle of division around 25° (**Fig. 14B, C rebuttal**). While we noticed a slight broadening of the planar group, we failed to observe any perpendicular division of the basal stem cells in Hras^{G12V/+}-mosaic *versus* wild-type-mosaic (**Fig. 14B, C, D rebuttal**). We infer that all the daughter cells remain within the basal stem cell layer after cell division in both the mosaic models, irrespective of injury. These results are consistent with ours and other prior studies, that showed that all cell divisions occur parallel to the basal stem cell layer in the adult skin epidermis to both maintain tissue homeostasis and regenerate full-thickness wounds (Ipponjima S. et al., 2016; Rompolas P, Mesa KR, et al., 2016, Mesa KR, Kawaguchi K, Cockburn K. et al., 2018, Cockburn K, Annusver K et al. 2021 and Liu N. et al., 2019). Collectively our data show

REDACTED

that *injury does not promote a perpendicular angle of division to the basal stem cell layer in mosaic wild-type or Hras^{G12V/+} models.*

Collectively, our data help to construct a model whereby the balance between Hras^{G12V/+} and wild-type cells is reached by the selective increase of wild-type cell proliferation after injury. The increase in wild-type cell proliferation is accommodated in the basal stem cell layer by an increase in differentiation of both wild-type and Hras^{G12V/+} cells.

R3.8) Vice versa, as proliferation does not change compared to the non-wounded situation, why not see a similar relative increase in thickness in the unwounded situation when compared to Mutant max wounded condition?

We were intrigued by the reviewer's comments, and we analyzed the thickness in unwounded Hras^{G12V/+}-mosaic mice where we can compare wild-type and Hras^{G12V/+} areas. Interestingly, we observed an increase in thickness in the area of the skin associated with large Hras^{G12V/+} clones in contrast to wild-type clone of the same size (**Rebuttal Fig. 15**). These data further support our measurement of the rate of differentiation performed in uninjured condition by staining for the early differentiation marker Keratin 10 as well as those reported in Sandoval et al., 2021 (**Manuscript Fig. 3H**).

R3.9) The data on Map kinase are very limited and somewhat simplistic and only provide minimal insight into why wt cells are able to respond or not. The map kinase activity is rather dynamic and often biphasic upon growth factor stimulation. Taking one point is not very helpful, also as the activation under all three conditions seems to be very similar upon wounding regardless whether wt, mosaic or Ras mutant cell max. Normalization should also be done with respect to total map kinase. Western blot analysis also does not allow to examine whether Map kinase activity is higher in wt versus Ras mutant cells under mosaic conditions, which one would predict.

We agree with the reviewer that ERK1/2 is characterized by an “on” and “off” activity in the minute range. Specifically, previous literature showed that bursts of activity happen more frequently in the areas where the skin cells were dividing (Hiratsuka T et al., 2015). Therefore, our western-blot analysis allowed us to have a better assessment of ERK1/2 activity related to cell proliferation that are more likely to be activated when we collected the samples. Our western-blot analyses were performed on ear samples quickly harvested from mice sacrificed during the course of an entire day - 3 days post-wound induction (PWI) - and from different litters. This experimental design should account for the fluctuation of ERK activity considering the different replicates were taken at different times of the day.

However, to experimentally address the reviewer's point, we analyzed additional time points to assess ERK1/2 activity at 4 and 7 days PWI. In the uninjured skin epithelium, wild-type mice exhibited a lower level of activated ERK1/2 when compared to Hras^{G12V/+}-mosaic mice in both 4 and 7 days PWI, due to Hras^{G12V} mutation that confers a proliferative advantage to mutant cells. In the injured condition, we observed two different outcomes at day-4 and at day-7 PWI. At 4 days PWI/7 days post-tamoxifen, we showed an increased levels of phosphoERK1/2 in wild-type mice that are undergoing injury-repair. In Hras^{G12V/+}-mosaic, phosphoERK1/2 level was not significantly different between uninjured and injured conditions. Moreover, phosphoERK1/2 levels

were similar after injury in all three models (**Rebuttal Fig. 16A**). Similar conclusions were obtained in the western-blot analyses at day 3 PWI/6 days post-tamoxifen) in the original manuscript (**Manuscript S11A**). Interestingly, at 7 days PWI/10 days post-tamoxifen, we showed a significant higher level of phosphoERK1/2 in Hras^{G12V/+}-mosaic than in wild-type-mosaic in both uninjured and injured conditions and the levels of phosphoERK1/2 did not increase significantly upon injury in either the wild-type or Hras^{G12V/+}-mosaic models (**Rebuttal Fig. 16B**). These changes in ERK1/2 activity at 7 days PWI were consistent with the proliferative rates we measured in Hras^{G12V/+}- and wild-type-mosaic models at the same time point. In the wild-type-mosaic model, the initial increase in proliferation rate at 3 days post-injury return to baseline by 7 days PWI and looked similar to uninjured mice. In Hras^{G12V/+}-mosaic mice, the initially elevated proliferation at 3 days PWI also decreased by 7 days PWI but it looked greater than wild-type-mosaic models, specifically for wild-type cells (**Manuscript Fig. 3C, D**).

REDACTED

To further examine a potential role of ERK1/2 activation in the competition between wild-type and Hras^{G12V/+} cells, we performed phosphoERK1/2 immunostaining in Hras^{G12V/+}-mosaic epidermis in both uninjured and injured ears at 6 days post-tamoxifen injection. In the uninjured condition, we observed a higher number of phosphoERK1/2 positive cells in Hras^{G12V/+} clones compared to wild-type ones supporting the competitive and proliferative advantage of Hras^{G12V/+} cells over wild-type cells, as the reviewer mentioned (**Manuscript Fig S11C, D**). Interestingly, during injury-repair, phosphoERK1/2 immunostaining showed that the number of Hras^{G12V/+} cells positive for phosphoERK1/2 was similar in both uninjured and injured scenarios showing that Hras^{G12V/+} cells have already maximized their ERK1/2 activation before injury consequently limiting their ability to increase proliferation during injury-repair. On the other hand, the number of wild-type cells positive for phosphoERK1/2 increased together with the proliferative events upon injury (**Manuscript Fig S11C, D**). Collectively, this new phosphoERK1/2 immunostaining data at the single cell level in Hras^{G12V/+}-mosaic model supported our western-blot data (**Manuscript Fig S11A**). Furthermore, we investigated the activation of another Ras-downstream pathway PI3K/AKT involved in cell proliferation, and it was not significantly affected by injury-repair (**Manuscript Fig. S11B**). Overall, these data provide new molecular insights on *the selectivity of the wound in increasing wild-type cell proliferation and phosphoERK1/2 upon injury induction* that is already maximized in Hras^{G12V/+} cells.

R3.10) The authors claim in the text that the wt cells are more proliferative under mosaic conditions than under wt conditions only (Figure 3G, right two upper (control mosaic) and lower panel (wt/mutant mosaic)). However, this difference is rather minimal and not directly compared within the same graph applying stringent statistics. Looking at all of the data, it seems that wounding induces a similar proliferative response in wt cells regardless whether Ras Mutant cells are present in mosaic conditions.

We thank the reviewer to point out the lack of clarity in the graphical representation. We modified the graph in the former Figure 3G, H (**Manuscript Fig. 3C, D**) to show appropriate statistical analysis and facilitate the comparison between Hras^{G12V/+} and wild-type cells in uninjured and injured conditions. In the uninjured skin epithelium, wild-type cells were significantly more proliferative in Hras^{G12V/+}-mosaic models than in wild-type mosaic models in the last two time points analyzed (day-17 and 1-month post-tamoxifen induction). In the injured skin epithelium, wild-type cells in Hras^{G12V/+}-mosaic models had more mitotic events than under wild-type condition between 10 days to 1-month post-tamoxifen injection. Thus, ***the balanced proliferation of wild-type and Hras^{G12V/+} cells sustained at later time points after injury would effectively continue to prevent the expansion of Hras^{G12V/+} cells in the Hras^{G12V/+}-mosaic model.***

R3.11) Some of the statistics is not clear from the legend. Sometimes the authors refer to N= X mice but sometimes just to N=X, in case of the latter what does that mean? Some of the data is also based on n=2 mice, which does not help to show biological variation (n=3 would be the absolute minimum).

We changed the figure legends to show that N is always referred to the number of mice analyzed. We agree with the reviewer that every experiment requires a minimal of 3 replicates to show biological variation. Each experiment in the manuscript was always performed on at least three mice from different litters and breeding cages to ensure a correct interpretation of the outcome. The only exception was in figure 5. We initially had 2 mice for wild-type-mosaic and Hras^{G12V/+}-mosaic models and now we have added an additional replicate. The comparison remains consistent considering the little biological difference within the replicates.

Referee #4 (Remarks to the Author):

Expertise skin stem cells, cancer (including HRAS-driven) models

The authors use sophisticated imaging and molecular genetics to study the role of wounds in tumorigenesis. They use an oncogenic allele of Hras (and Kras) to stimulate squamous tumorigenesis with and without wounding. It is known that chronic wounds are associated with tumorigenesis in the skin, but whether the effect of the wound is on wildtype or mutated cells is not clear. Interestingly, the authors use imaging to follow the progression of both wildtype and mutant clones of cells in the skin, and found that wildtype cells preferentially expand over mutant cells upon wounding. On the other hand, in the absence of a wound, Hras mutant cells expand. They go on to show that p21, a cell cycle inhibitory protein plays a role in this effect. They summarize their observations to conclude that wildtype cells actually suppress the expansion of Hras mutant cells in response to wounding as a means to suppress tumorigenesis. They suggest that this will lead to improved treatments for cancer via the promotion of wildtype cells in wound repair. The methods employed are elegant and the data collected appear robust. While there are certainly some interesting observations presented, the data raise many questions, and many of the assertions are not well defined. The relation of this study to cancer is also not clear, which raises the issue of impact of the study. If this is not a cancer study is it really more relevant to cell competition? wound healing?

We thank the reviewer for their evaluation of our work. We have responded to each point raised by the reviewer and changed the manuscript accordingly. We think that our study highlights the importance of wild-type epithelial cells as sentinels of the skin epithelium. Specifically, we identify how wild-type cells molecularly and behaviorally preserve homeostasis and suppress the emergence of aberrant oncogenic growth even in the pro-tumorigenic condition of the wound. Thus, this work is placed at the interface of regeneration, cell competition and oncogenic mutations.

R4.1), the authors refer to tumorigenesis throughout the manuscript, but never show the formation of a tumor.

do they mean hyperplasia? benign cyst or growth? papilloma? There are no macroscopic images to show what is really happening.

We thank the reviewer for highlighting the absence of macroscopic views of the injured area of Hras^{G12V/+} and Kras^{G12V/+}-max models. We added macroscopic images of the aberrant oncogenic growth with benign squamous papillomatous changes detectable through H&E analysis of injured Hras^{G12V/+} and Kras^{G12D/+}-max models (**Manuscript Fig. S1 and Fig. S6**).

R4.2), The authors should address the possibility of oncogene induced senescence (OIS). One way to interpret the data is that the Hras-mutant cells are out-competed due to senescence driven by constitutive activity of the Hras allele. Large tumors typically only form in this model when coupled to mutations in p53, which allows mutant cells to overcome OIS.

We thank the reviewer for the helpful suggestion, which we address below in detail. We agree with the reviewer that ERK hyperactivation in Hras^{G12V/+} cells could trigger cell cycle exit and cell senescence. We tackled this question with various experimental approaches. First, to answer this and the questions from other reviewers about molecular mechanisms, we have generated a new scRNA-sequencing dataset to determine the differentially expressed genes between Hras^{G12V/+} cells and wild-type cells in the injured context. This approach gave us an opportunity to determine the behaviors of Hras^{G12V/+} cells, including senescence, that can potentially participate in the competition scenario observed between wild-type and mutant cells. Senescence-associated transcripts were not among the differentially expressed genes in the comparison above. In contrast, we observed increased expression of differentiation genes in Hras^{G12V/+} cells after injury (**Manuscript Fig. S5E**). These data were confirmed by immunostaining for marker of proliferation (phosphoHistone-3) and differentiation (Keratin10) upon injury in Hras^{G12V/+}-mosaic models (**Manuscript Fig. 3B, H, S3H**). The observed increase in differentiation capacity could be the result of a persistent activation of the ERK signaling as shown in previous reports (Aikin TJ et al., 2020).

REDACTED

Second, we tested via immunofluorescence for two senescence biomarkers: beta-galactosidase and cell cycle inhibitor p21. We did not observe any specific positive staining in any of the skin samples analyzed for beta-galactosidase staining (**Rebuttal Fig. 17** – kidney as a positive control). Interestingly, we found a down-regulation of

p21 expression in Hras^{G12V/+}-mosaic models when compared to wild-type -mosaic models during injury-repair (**Rebuttal Fig. 18**).

Overall, our data showed that oncogene induced senescence (OIS) is not detected in the Hras^{G12V/+} cells in the injured Hras^{G12V/+}-mosaic model but Hras^{G12V/+} cells were able to respond to the injury by changing their differentiation capacity.

REDACTED

R4.3), According to data from Phil Jones and others, sun exposed human skin keratinocytes almost universally are mutated for Ras, Notch etc, so it is not clear if the data presented here are relevant to human data since SCCs only really form in sun exposed skin. In other words, how could one promote the wildtype cells to suppress mutant cells, where there are no wildtype cells?

Recent sequencing studies on healthy skin in human showed a large presence of cells without driver mutations identified as normal or wild-type cells (Martincorena et al., 2015; Fowler et al., 2020). Typically, only a handful of driver mutations are positively selected in the skin epithelium, including Ras oncogenes, and many of the other mutations are essentially neutral (Martincorena et al., 2015). Other tissue such as esophageal and endometrial epithelia have a similar mutational landscape and develop cancers as sun-exposed skin (Martincorena et al., 2018; Moore et al., 2020). Thus, we believe that the analysis of behaviors and molecular pathways of cells that undergo positive selection and have an elevated risk of squamous skin cancer such as Hras^{G12V/+} and Kras^{G12D/+} cells are relevant to identify preventive therapy applicable also to human skin and other tissues. The suppression of the positive selection of Ras mutant cells during injury-repair provides a cellular and molecular mechanism that could be therapeutically exploited to empower neutral/wild-type cells in the competition with the mutant neighbors.

R4.4), The authors probably should have shown that mutations in Hras or Kras using the strategy employed here coupled with injury actually leads to tumorigenesis, since the study is based on this assumption.

As mentioned above in 4.1 comment, we added macroscopic images to show the aberrant oncogenic growth in the injured areas of Hras^{G12V/+} and Kras^{G12D/+}-max models. We are thankful for the reviewer's note because the changes we made in the supplementary figures reinforce our conclusion about the role of fewer wild-type cells in preventing oncogenic growth during injury-repair (**Manuscript Fig. S1 and Fig. S6**).

R4.5), The authors restricted their analysis to ear punches as a relatively straightforward approach for both uniformity (and probably imaging?). However, in these GEMM models, tumors rarely ever form on the ear, so I am curious if choosing to study the ear has limited the interpretation of the data? can the authors replicate the findings on the backskin?

We thank the reviewer for their comment. We would like to highlight that we adopted the ear skin in our analysis to track the exact same region of the skin epithelium over time without invasive imaging approach that could create additional variables. This is not possible in the back skin due to the usage of pins necessary to stabilize the back skin from respiratory motion during imaging. While the wounds these pins create are small, they may influence the behaviors of cells within the tissue. Nevertheless, an analysis of a single time point is possible. To address this reviewer question, we followed the same experimental design adopted for the injury in the ear and performed a 4 mm punch biopsy in the back skin of both Hras^{G12V/+}-mosaic and -max models. At 14 days post-wound induction, we observed abnormal oncogenic growth only in Hras^{G12V/+}-max models, as we documented for the wound in the ear (**Manuscript Fig. S1F, G, H, I**). Thus, despite of the local differences in the back and in the ear skin, wound promotes oncogenic growth only when nearly all cells in the basal stem layer expressed Hras^{G12V/+}.

R4.6), In addition, there are strains of mice that heal ear punches very quickly, it might be worthwhile to study those to see if their proposed mechanism is supported.

We agree with the reviewer's comment that crossing Hras^{G12V/+}-mosaic mice with fast-healing mice might further support our conclusions. Indeed, it has been reported in Bedelbaeva et al., 2010 that p21^{null} is a fast-healing model due to an enhanced proliferative capacity of epithelial cells that we also documented in the manuscript (**Manuscript Fig. S9H, I**). Specifically, p21^{null} mice display morphological and histological regenerative responses like the super-healer MRL mice (Bedelbaeva et al., 2010). We collected data and replicates of injured p21^{null}/Hras^{G12V/+}-mosaic mice. The data showed the lack of the competitive advantage of Hras^{G12V/+}/p21^{null} cells during injury-repair supporting the mechanism proposed in the manuscript. Overall, our data showed that the enhance injury-repair process achieved by p21 loss does not change the competitive dynamic between wild-type and Hras^{G12V/+} cells observed in injured Hras^{G12V/+}-mosaic mice (**Rebuttal Fig. 19**).

REDACTED

R4.7), This is perhaps a minor point, but the authors refer to Kras as an "aggressive" allele, but I am not familiar with this distinction with regard to this particular model system, can the authors show macroscopically that the animals with Kras have a more severe phenotype than those with Hras?

We thank the reviewer for pointing out the lack of clarity in the terminology used. As we also replied to Reviewer 1 in comment 1.1. We define Kras as an aggressive allele because mice with homogeneous activation of Kras^{G12D/+} in the skin epidermis rapidly developed oncogenic growth in areas subjected to constant abrasions such as anus and mouth in contrast to Hras^{G12V/+}-max models. These data are consistent with other reports in literature (Lapouge et al., 2011; van der Weyden et al., 2011). We changed the text in the manuscript to clarify the terminology used (**Rebuttal Fig. 1A, B**).

Reviewer Reports on the First Revision:

Referee #1 (Remarks to the Author):

The authors have carefully addressed my major concerns and the manuscript has improved significantly from the extensive revision. I have one last point that the authors will need to clarify to enhance the message of the manuscript and solidify the conclusions.

The authors use the phrase proliferative advantage. What do they mean by this? Are mutant cells outcompeting neighbors via elevated replication rates or via an increased fitness? In the sebaceous gland modelling of quantitative fate mapping data revealed enhanced fitness and very little change to cell cycle parameters upon KRasG12D expression (Andersen et al., 2019, Nature Cell Biology). Can the author exclude that the observed effects of HRas selective advantage during “steady state” is due to increased proliferation/faster cycle or simply imbalanced cell fate choices? And that the injury setting places cells on an equal playing field as all cells do not enter a state of imbalanced cell fate choices when the tissue needs to expand to cover the wound.

Referee #2 (Remarks to the Author):

The authors have done an outstanding job addressing my prior concerns (and appear to have also done so for the other reviewers, although I'll defer to them as my expertise is not in skin). They have performed an impressive amount of new experimentation, including new mouse models that induce the KRAS mutation in a small fraction of cells (via in utero injection of LV-CreER, showing that even rare KRAS mutational events can be competitively suppressed by WT cells under injury), scRNAseq studies to explore mechanisms (including by showing increased EGFR ligand production by fibroblasts and epithelial cells upon injury), and pharmacological and genetic methods to identify a role for the EGFR signaling pathway in providing a competitive advantage to WT cells during injury. This last point is particularly important, as they now provide a mechanism for how injury specifically boosts the fitness of the WT cells, via EGFR, MAPK and through modulation of p21, with strong support using multiple methods. In all, this revised manuscript provides much more mechanistic insight and improved models. These studies describe an important new mechanism whereby injury can promote the elimination of oncogenically initiated cells, thus reducing the risks of oncogenesis that might otherwise accompany repair – a novel tumor suppressive mechanism. On a side note, their results could also be relevant to the clinical observation that treating patients with RAS mutant lung cancers with EGFR inhibitors actually leads to worse outcomes (due to selective impairment of the WT cells?).

Referee #3 (Remarks to the Author):

The authors have addressed most of my main concerns, by adding substantial new data that show in two different ways that inhibiting proliferation in wt cells allows H-Ras cells to grow out now under mosaic conditions. They then have done in depth single cell sequencing in which they identify EGFR

ligands to be unregulated upon wounding and showing that the wt cells are still able to respond to these ligands upon wounding, whereas the H-RAS cells are already maxed out in their response. Although I do understand that all of this data has not ended up in the main figures to not distract from the flow, I feel a considerable amount of important data is now not easily accessible by being hidden in the supplementary figures. The authors should consider bringing at least some of these data in, especially on the differential response in EGFR signaling in wt versus H-Ras mutant cells in non-wounded versus wounded.

Referee #4 (Remarks to the Author):

The authors have substantially updated the manuscript with new data, quantification and clarification. Particularly impressive is their efforts to define the mechanism for the difference between WT and Ras+. In their 35 page rebuttal the authors satisfactorily addressed most of the concerns raised by reviewers. The following issues still linger for this reviewer:

1, the context in which the key observation is made (WT advantage over Ras+) is quite specific. There has to be a wound, but it cannot be too deep. The number of Ras+ cells is specific (not too many!). Presumably this is only observed when there is just a single Ras allele mutated. As a result, the relevance of the study to broader biology such as cancer is not quite clear. Yes, the authors showed that tumors form in this version of the manuscript, but despite the increased mechanistic insight in this version, there is no study of cancer, or the implications of the WT vs Ras+ battle. Beyond changes in clone size, what do the findings here mean for cancer?

2, I am surprised that the overlap between tomato+ and Ras+ is only 30-50%. This is concerning since most of the quantification in the manuscript depends on the Tomato as a proxy for the Ras mutation. In addition, it appears as though there is only a 2x difference in the number of cells that get Ras+ through recombination between the mosaic and the max conditions, so it is hard to call this sparse versus complete labeling. I tend to agree with the authors interpretation of their data, but if less than half of the Tomato+ cells actually have the mutation, then other scenarios are possible. In addition, the authors bring up the issue of leaky Cre activity which further clouds interpretation. Perhaps the authors can describe all the possible scenarios considering these experimental limitations and then state why their data most prominently points to one of these possibilities?

3, the authors did attempt to rule out OIS playing a role in WT advantage over Ras+ through immunostaining and RNA profiling. I would recommend running the b-galactosidase activity assay instead of immunostaining for the enzyme, which is not standard in the field. This is a simple assay, done on fresh frozen tissue, and many kits are available. In addition, while p21 is upregulated in some senescence contexts, people typically use p16 as a more broadly applicable marker.

Referee #5 (Remarks to the Author):

I have mainly read the manuscript in light of the single-cell data that have been added to this manuscript. Overall, these data are only very briefly discussed in the main paper. This is a bit

surprising, as on at least two locations in the manuscript they seem to provide key insights in the mechanisms proposed. For instance, on page 8 from line 1-5, scRNA-seq provide proof that injury triggers an increase of differentiation markers and a decrease of stemness markers. Likewise, on page 10, line 10-13, scRNA-seq is used to identify EGFR ligands as the key mediator influencing epithelial cell proliferation. In the supplementary figures, scRNA-seq are shown in a bit more detail, but a lot of information about these analyses is still lacking. Particularly, the flow of the scRNA-seq analysis is difficult to follow, it is difficult to see how specific conclusions are drawn and overall, I feel that the potential value of the scRNA-seq experiments is not fully explored.

I did enjoy reading the manuscript, and I do agree that (most probably) the key conclusions of the manuscript will not change based on my remarks. However, I do feel that the scRNA-seq data should be presented in more detail, according to current standards in the field (especially when considering a journal with the prestige of Nature). It should also be more clear how these data contribute to the manuscript. I have therefore highlighted some questions that I feel really should be addressed:

1) Are experiments referred to on page 8 and 10 the same and just focusing on a different cell type compartment, or do they represent different experiments from different mice ? Can a UMAP showing all cell types and their marker genes be included ? How many cells were profiled altogether for each mouse ? Can some QC parameters of the scRNA-seq data be provided at the very least ? Can details about when samples were collected for scRNA-seq be included in the main text ?

2) It is sometimes difficult to follow the analysis and strategy of scRNA-seq analyses because data are only described in the legends of Supplementary figures.

- For instance, I fail to understand what Fig S4 is showing. Is this a trajectory analysis - I don't see the specific trajectory that is referred to ? This really needs more details (a lot more than referring to a paper that is only available on BioRx; ref. 23). How were these trajectories constructed ? This is quite confusing.

- Is Ki67 sufficient to define the G2M status of cells – additional markers could be shown.

- And is keratin 10 expression (one marker) sufficient to define 3 different epithelial cell states ? In scRNA-seq data you do have the possibility to look at many additional genes or gene signatures, in addition to those assessed by immunofluorescence. I feel this should be explored much better throughout. The same applies for stemness markers.

- Are we supposed to see differences between the trajectories ? If so, I fail to see such differences because the appropriate quantification/statistics is not provided.

3) Same applies to Fig S5. More details are needed.

- Is keratine6a sufficient to distinguish between healthy cells and cells that respond to injury (blue versus yellow) ? Can additional markers be included ? This would be more convincing.

- Are we supposed to see a difference between the models (injured wild-type mosaic versus injured HRAS mosaic) because no statistics is provided.

- What is Fig 5SC showing ? I guess it provides evidence for 'the earlier differentiation' in the injury model. But is this effect significant ? I don't see any confidence intervals, and also the binning strategy is unclear. Why do you consider 8 bins and why not do this in a continuous manner ?

- In Fig S5D – a pathway analysis is shown, but the corresponding differential gene expression analysis is not given. At the very least a volcano or list of differentially-expressed genes should be provided allowing the reader to assess which genes and how many of them are contributing to the

pathways identified.

- Finally, in the manuscript other methods are used to show that apoptosis or proliferation are (not) changed. These could also be further explored in the scRNA-seq data.

4) Additional details are also needed for Figure S7.

- For instance, in panel A it is very unclear how the green subcluster was identified. It is referred to as the 'source of EGFR ligands' subcluster, but this is vague. I would really recommend the authors to properly annotate the subclusters with additional marker genes (find marker gene function in Seurat).

- In Fig 7SC, several conditions are compared. But, since in the main manuscript data are only very briefly referred to, it is unclear if there are differences between groups. If this panel is indeed meant to indicate differences, proper statistical analyses should be performed (comparing abundancies of subclusters for the 3 mice tested between each of the subgroups). Also, the number of cells detected for some subclusters is quite low.

- Fig 7SE shows a volcano plot and highlights numerous genes that are differentially expressed. These include 2 genes from the EGFR family. It should be made clear in the main manuscript that besides these 2 genes, numerous other genes also differentially expressed and that these could therefore also act as potential mediators.

- Figure 7SF/G: it might be better to also show median values and individual dots on the violin plots. Regarding statistics, it would be better to additionally perform statistics per mouse tested rather than comparing expression between all individual single cells between conditions (irrespective of the number of mice tested).

5) Fig S8E and F: Several of the genes are barely expressed yet significant between healthy cells and cells that respond to injury. I wonder to what extent this bears biological significance. If a gene is barely expressed, it might be better to just indicate that expression is very low (and probably not biologically relevant).

Author Rebuttals to First Revision:

Referees' comments:

Referee #1 (Remarks to the Author):

The authors have carefully addressed my major concerns and the manuscript has improved significantly from the extensive revision. I have one last point that *the authors will need to clarify to enhance the message of the manuscript and solidify the conclusions.*

The authors use the phrase proliferative advantage. What do they mean by this? Are mutant cells outcompeting neighbors via elevated replication rates or via an increased fitness? In the sebaceous gland modelling of quantitative fate mapping data revealed enhanced fitness and very little change to cell cycle parameters upon KRasG12D expression (Andersen et al., 2019, Nature Cell Biology). Can the author exclude that the observed effects of HRas selective advantage during “steady state” is due to increased proliferation/faster cycle or simply imbalanced cell fate choices? And that the injury setting places cells on an equal playing field as all cells do not enter a state of imbalanced cell fate choices when the tissue needs to expand to cover the wound.

We thank the reviewer for their comment, and we integrate here the analysis of cellular behaviors we performed to determine their contribution in wild-type and oncogenic-Ras cell competition in uninjured (“steady state”) and injured settings.

As shown in our previous rebuttal, angle of cell division does not contribute to the competition outcome in either uninjured or injured conditions. Those results were consistent with ours and previous studies showing that in the adult epidermis, differentiation is not achieved by perpendicular cell division, but rather by expression of Keratin10 starting in the basal layer and eventual delamination into the suprabasal layer.

Therefore, to determine which behavior is responsible for the competitive scenarios of uninjured and injured settings, we compared the rate of proliferation and differentiation in Hras^{G12V/+} and wild-type-mosaic models.

In uninjured conditions, we observed that oncogenic-Ras cells have a higher proliferative capacity compared to wild-type neighbors, and this was not balanced by an equally increased differentiation rate between Hras^{G12V/+} cells and wild-type cells, inferred from the Keratin10 staining and quantifications (**Rebuttal - Table 1**). Therefore, oncogenic-Ras cells contribute more than wild-type cells to replenishing cells lost to differentiation in the basal stem cell layer of the skin epithelium in the uninjured condition.

In injured conditions, the increases in differentiation rates are similar in wild-type and Hras^{G12V/+} cells. In contrast, the proliferative rate of wild-type cells, but not Hras^{G12V/+} cells, increases after injury in the Hras^{G12V/+}-mosaic model. We infer that the lack of expansion of Hras^{G12V/+} cells during injury-repair is not due

	Injured (day 3 PWI)		Injured (day 7 PWI)		Uninjured	
	wild-type cells	Hras ^{G12V/+} cells	wild-type cells	Hras ^{G12V/+} cells	wild-type cells	Hras ^{G12V/+} cells
Proliferation (Gain of cells)	++	=	=	=	=	++
Differentiation (Loss of cells)	+	+	=	=	=	+
	Net gain (compared to WT cells in uninjured ear)	Net loss (compared to Hras ^{G12V/+} cells in uninjured ear)	Net 0 (compared to Hras ^{G12V/+} cells in day7 pwi ear)	Net 0 (compared to WT cells in day 7 pwi ear)	Net 0 (compared to WT cells in uninjured ear)	Net gain (compared to WT cells in uninjured ear)

Table 1. Comparison of proliferation and differentiation rates between wild-type and Hras^{G12V/+} cells in uninjured and injured Hras^{G12V/+}-mosaic models (day-3 and day-7 PWI). The fold change differences in cells positive for phospho-Histone3 (proliferation – day-3 PWI) or mitotic figures (proliferation – day-7 PWI) and Keratin10 (differentiation) staining in the **Manuscript - Figure 3b, d and Extended Data Figure 3i, j** are used to calculate the gain or loss of cells in Hras^{G12V/+}-mosaic skin epithelium.

to altered differentiation but instead arises from altered proliferation of wild-type cells (**Rebuttal - Table 1**). We would like to add that, while proliferation showed a transient and selective spike in wild-type cells on day-3 post-injury induction, tracking the rate of mitotic events at day-7 post-injury showed that the proliferation and differentiation rates of wild-type and Hras^{G12V/+} cells reached a similar level (**Manuscript - Figure 3d and Extended Data Figure 3j**). We propose that this eventually equal proliferation and differentiation rates continues to prevent the expansion of Hras^{G12V/+} cells and suppresses the Hras^{G12V/+} cell competitive advantage against wild-type cells during injury-repair.

Referee #2 (Remarks to the Author):

The authors have done an outstanding job addressing my prior concerns (and appear to have also done so for the other reviewers, although I'll defer to them as my expertise is not in skin). They have performed an impressive amount of new experimentation, including new mouse models that induce the KRAS mutation in a small fraction of cells (via in utero injection of LV-CreER, showing that even rare KRAS mutational events can be competitively suppressed by WT cells under injury), scRNAseq studies to explore mechanisms (including by showing increased EGFR ligand production by fibroblasts and epithelial cells upon injury), and pharmacological and genetic methods to identify a role for the EGFR signaling pathway in providing a competitive advantage to WT cells during injury. This last point is particularly important, as they now provide a mechanism for how injury specifically boosts the fitness of the WT cells, via EGFR, MAPK and through modulation of p21, with strong support using multiple methods. In all, this revised manuscript provides much more mechanistic insight and improved models. These studies describe an important new mechanism whereby injury can promote the elimination of oncogenically initiated cells, thus reducing the risks of oncogenesis that might otherwise accompany repair – a novel tumor suppressive mechanism. On a side note, their results could also be relevant to the clinical observation that treating patients with RAS mutant lung cancers with EGFR inhibitors actually leads to worse outcomes (due to selective impairment of the WT cells?).

We thank the reviewer for appreciating the improvements and how we addressed their concerns. We agree with the reviewer that our discoveries have identified a new mechanism to counteract oncogenic Ras-cells during injury-repair that could have profound implications for therapeutic interventions such as surgery or tumor relapse. As indicated by the reviewer, we have also highlighted the importance of identifying the best therapeutic strategy towards oncogenic mutations present in the primary tumor as to allow mosaic tissues to deploy natural defenses against tumor emergence such as the proliferation of wild-type cells.

Referee #3 (Remarks to the Author):

The authors have addressed most of my main concerns, by adding substantial new data that show in two different ways that inhibiting proliferation in wt cells allows H-Ras cells to grow out now under mosaic conditions. They then have done in depth single cell sequencing in which they identify EGFR ligands to be unregulated upon wounding and showing that the wt cells are still able to respond to these ligands upon wounding, whereas the H-RAS cells are already maxed out in their response. Although I do understand that all of this data has not ended up in the main figures to not distract from the flow, I feel a considerable amount of important data is now not easily accessible by being hidden in the supplementary figures. The authors should consider bringing at least some of these data in, especially on the differential response in EGFR signaling in wt versus H-Ras mutant cells in non-wounded versus wounded.

We thank the reviewer for appreciating our new data. We agree with the reviewer's suggestions to bring into the main figure key data related to the identification of molecular mechanism. Therefore, the data showing the differential response in EGFR signaling between wild-type and Hras^{G12V/+} mutant models in both uninjured and injured conditions have been moved from supplementary figure into the **Manuscript - Figure 5b**.

Referee #4 (Remarks to the Author):

The authors have substantially updated the manuscript with new data, quantification and clarification. Particularly impressive is their efforts to define the mechanism for the difference between WT and Ras+. In their 35 page rebuttal the authors satisfactorily addressed most of the concerns raised by reviewers. The following issues still linger for this reviewer:

1, the context in which the key observation is made (WT advantage over Ras+) is quite specific. There has to be a wound, but it cannot be too deep. The number of Ras+ cells is specific (not too many!). Presumably this is only observed when there is just a single Ras allele mutated. As a result, the relevance of the study to broader biology

such as cancer is not quite clear. Yes, the authors showed that tumors form in this version of the manuscript, but despite the increased mechanistic insight in this version, there is no study of cancer, or the implications of the WT vs Ras+ battle. Beyond changes in clone size, what do the findings here mean for cancer?

We thank the reviewer for giving us the opportunity to explain the implication of our study for cancer biology and revisit the discussion of the manuscript to reflect this observation (**Manuscript - page 16; lines 3-16**). In this study we discovered the importance of wild-type cell proliferation as a first line of defense against tumorigenic stimuli: injury and oncogenic-Ras cells. Traditional therapeutic approaches used for cancer treatment involve suppressing the proliferation of all cells, both mutant and wild-type. While these approaches restrain tumor expansion, they also impair the opportunity for the tissue to deploy endogenous mechanisms of tissue homeostasis, such as selective promotion of wild-type cell proliferation.

Specifically, our study highlights the importance of understanding not only how tumors with different mutations, such as in Ras oncogenes, might respond to targeted therapies but also therapy timing might affect tumor response, depending on the tumor's state: established malignancy versus a premalignant state. For example, EGFR inhibition is an effective treatment in advanced head and neck squamous cell carcinomas. Tumors with mutations in Ras are resistant to EGFR inhibition, thus suggesting that the manipulation of EGFR pathway with drugs acting epistatic to Ras in the EGFR signaling cascade, such as Gefitinib, would have little efficacy in treating RAS mutated tumors. Our data would argue that in early cancers or in precancerous states, EGFR promotion (rather than inhibition) might provide a competitive advantage to the wild-type cells that co-exist with oncogenic-Ras cells. In the model of head and neck squamous cell carcinoma, approximately 30-40% of precancerous squamous cell papillomas of the aerodigestive tract harbor Kras or Hras mutations (Sasaki E. et al., 2020). This is considered a precancerous state, and based on the data in our study, we suggest that localized EGFR activation in the oropharynx could resolve papillomas and decrease the risk of malignant transformation by specifically promoting wild-type cell proliferation.

Extrapolating more broadly, the implications of this phenomenon can be explored outside of RAS specific mutant clones to evaluate how wild-type cells may control more complex clones in the premalignant state. In oncology, many efforts are being made for advancing "field therapy", which is a term for treating broad areas of premalignant tissue to decrease the risk of malignant transformation. For example, treatment of the premalignant condition actinic keratoses is important for reducing the risk of transformation to cutaneous squamous cell carcinoma. Current field therapies use antimetabolites like 5-fluorouracil or induce an immune response against the precancerous cells using TLR-7/8 agonist Imiquimod. It would be very attractive to instead or in conjunction promote wild-type clonal advantage over these premalignant clones to clear field damage by oncogenic clones.

Another possible future clinical application of our discoveries could be on relapses upon the surgical removal of tumors with Ras mutation. The area around the preexisting tumor, likely also harboring Ras mutations, could be considered a pre-malignant tissue that could benefit from treatments to promote the competitive advantage of non-oncogenic wild-type cells at the expense of oncogenic-Ras cells, such as possibly EGFR ligands. These therapies could effectively prevent tumor recurrence and more invasive treatments.

Collectively, our research has far-reaching impacts across stem cell and cancer biology because it reveals a novel therapeutic target, EGFR and its ligands, to suppress tumorigenesis by manipulating cell competition.

2, I am surprised that the overlap between tomato+ and Ras+ is only 30-50%. This is concerning since most of the quantification in the manuscript depends on the Tomato as a proxy for the Ras mutation. In addition, it appears as though there is only a 2x difference in the number of cells that get Ras+ through recombination between the mosaic and the max conditions, so it is hard to call this sparse versus complete labeling. I tend to agree with the authors interpretation of their data, but if less than half of the Tomato+ cells actually have the mutation, then other scenarios are possible. In addition, the authors bring up the issue of leaky Cre activity which further clouds interpretation. Perhaps the authors can describe all the possible scenarios considering these experimental limitations and then state why their data most prominently points to one of these possibilities?

We thank the reviewer for prompting us to better clarify in the text how our conclusions are made in the context of the cited technical characteristics of our genetic models (**Manuscript - page 4 - lines 16-19; page 6 - line 7 and Methods**).

With respect to CreER leakiness: we have clarified in both the methods and main text how the live imaging approach we established allows us to monitor labelled cells that are the result of leaky inducible CreER systems as we track the same large tissue areas over time. Additionally, given that CreER leakiness occurs to the same extent in all models, we extracted our conclusions by always comparing Hras^{G12V/+}-mosaic to wild-type-mosaic models.

With respect to tdTomato being a surrogate for identifying mutant cells: the expansion of the tdTomato population in Hras^{G12V/+}-mosaic in the inducible K14CreER model was observed in two additional systems. First, we have added a model in our resubmission that recreated mosaicism of cells expressing inducible Cre (i.e., via in utero injection of lentiviruses expressing CreER (LV-CreER)). In this system, only a subset of epidermal cells is infected with LV-CreER, and Cre activity can be induced in all infected cells with tamoxifen using our 'maximal' dose (stated in Methods), thereby maximizing the ability of Cre to recombine LoxP sites from both the tdTomato allele and the mutant Ras one within the same cell. Second, an already published model from our lab (Pineda, JCB 2019) that recreated mosaicism of cells expressing an epidermal-specific, constitutive Cre (i.e., LV-Cre via in utero injection approach). This constitutively expressed Cre ensures the recombination of all LoxP sites present in the genome of infected cells because it is active from embryonic stages (~E9.5) and its expression and activity are persistent and do not depend on externally provided agent (i.e., tamoxifen injection).

Finally, given the evidence that Hras^{G12V/+} clones expand in the uninjured condition, the possibility of false negatives in the tdTomato population would, if anything, imply an underestimation of our described phenotype. Moreover, the floxed fragment in the Hras^{G12V/+} knock-in locus (2.4 kb) is significantly longer than the tdTomato locus (only a STOP codon). Thus, we infer that the tdTomato negative population should have exceedingly few or no recombination events that would therefore not influence the competitive dynamics observed between wild-type and Hras^{G12V/+} cells.

3, the authors did attempt to rule out OIS playing a role in WT advantage over Ras+ through immunostaining and RNA profiling. I would recommend running the *b-galactosidase activity assay* instead of immunostaining for the enzyme, which is not standard in the field. This is a simple assay, done on fresh frozen tissue, and many kits are available. In addition, while p21 is upregulated in some senescence contexts, people typically use *p16* as a more broadly applicable marker.

We thank the reviewer for suggesting the β -galactosidase activity assay to examine oncogenic-induced senescence in fresh frozen tissues. To address this point, we collected the same amount of fresh frozen tissue of skin epidermis preparation (3 mouse models for each condition analyzed) and positive controls (both mouse pancreas and kidney) and we performed the β -galactosidase activity assay following the manufacturer instructions (**Manuscript - Extended Data Fig. 3h; page 8 - line 3**). To further control the assay, we used different concentrations of the positive control provided by the kit. To quantify and monitor the formation of a green, fluorescent product (Fluorescein) from the reaction of β -galactosidase enzyme with the substrate, we used an ELISA plate reader (excitation: 475 / emission: 500-550) for 30 minutes, as indicated in the kit instruction. This assay confirmed our previous staining of the enzyme that the presence of Hras^{G12V/+} mutation does not induce senescence and β -galactosidase expression at day-3 post-wound induction.

Lastly, we utilized our scRNA-seq dataset to interrogate p16/Cdkn2a expression as the reviewer suggested. We found that in comparing basal uninjured and injured Hras^{G12V/+} cells only very few cells even express the p16/Cdkn2a RNA (**Rebuttal - Figure 1**). These data corroborate the conclusions obtained with the β -galactosidase assay and p21/Cdkn1a analyses that senescence is not triggered in our context.

Referee #5 (Remarks to the Author):

I have mainly read the manuscript in light of the single-cell data that have been added to this manuscript. Overall,

these data are only very briefly discussed in the main paper. This is a bit surprising, as on at least two locations in the manuscript they seem to provide key insights in the mechanisms proposed. For instance, on page 8 from line 1-5, scRNA-seq provide proof that injury triggers an increase of differentiation markers and a decrease of stemness markers. Likewise, on page 10, line 10-13, scRNA-seq is used to identify EGFR ligands as the key mediator influencing epithelial cell proliferation. In the supplementary figures, scRNA-seq are shown in a bit more detail, but a lot of information about these analyses is still lacking. Particularly, the flow of the scRNA-seq analysis is difficult to follow, it is difficult to see how specific conclusions are drawn and overall, I feel that the potential value of the scRNA-seq experiments is not fully explored.

I did enjoy reading the manuscript, and I do agree that (most probably) the key conclusions of the manuscript will not change based on my remarks. However, I do feel that the scRNA-seq data should be presented in more detail, according to current standards in the field (especially when considering a journal with the prestige of Nature). It should also be more clear how these data contribute to the manuscript. I have therefore highlighted some questions that I feel really should be addressed:

We thank the reviewer for their close engagement and insightful suggestions on the scRNA-seq data analysis and presentation. Guided by these constructive questions, we have comprehensively revised the scRNA-seq parts including new analysis (reflected in the updates of the main text, methods and figures), QC parameters, annotation of all cells (UMAP, cluster identity, dot plot of characteristic genes), identification and robustness of wound cells, statistical comparison of wild-type and *Hras*^{G12V/+} basal cell differentiation in uninjured and injured conditions, as well as clarification on EGFR ligand significance and cell sources.

Please find the point-by-point answers to all the questions below.

1. Are experiments referred to on page 8 and 10 the same and just focusing on a different cell type compartment, or do they represent different experiments from different mice?

The sequencing data on page 10 (focusing on fibroblasts and immune cells) includes the mosaic-induction scRNA-seq data presented in page 8 (focusing on keratinocytes) and additional sequencing data from full(max)-induction experiments to address questions that were raised by one reviewer in the previous revision. We have clarified the different datasets and their use in the methods.

- a. Can a UMAP showing all cell types and their marker genes be included?

We have now included the UMAP showing the broad clustering of all major cell types, a dot plot with their respective characteristic marker genes and additional UMAPs showing marker gene expression patterns (**Rebuttal - Figure 2**). In the manuscript figure (**Manuscript - Extended Data Figure 4b**) we included panel b and c shown in the **Rebuttal - Figure 2**;

biological replicate distributions are shown in higher detail for all the second level analyses regarding epidermal cells, fibroblasts and immune cells (**Manuscript - Extended Data Figures 4c, 6d and 7d**).

b. How many cells were profiled altogether for each mouse?

The following cell numbers passed QC: HMU 11437; HM12U 9864; HM5U 9650; HM5W 9590; HM12W 10281; HMW 8922; WTM11U 11856; WTM13U 9916; WTM3U 8416; WTMW11W 9181; WTMW13W 8747; WTMW3W 8320. This information is now included in the methods part.

c. Can some QC parameters of the scRNA-seq data be provided at the very least?

We have now included violin plots showing the total counts per cell, number of genes expressed per cell, and percentage of mitochondrial reads per cell (**Manuscript - Extended Data Figure 4a**, mosaic models; **Manuscript - Extended Data Figures 6a, 7a**, max models).

d. Can details about when samples were collected for scRNA-seq be included in the main text?

This information has been included in the main text (**Manuscript - page 8; lines 7-8**).

2. It is sometimes difficult to follow the analysis and strategy of scRNA-seq analyses because data are only described in the legends of Supplementary figures.

We have comprehensively revised the presentation of the scRNA-seq data. In the original Extended Figure 4 we presented separate UMAPs, proliferation status, differentiation markers etc. for each condition. We now show all datasets combined, added violin plots showing basal-, differentiation- and wound-related gene expression, etc. to increase clarity (**Manuscript - Figure 3 and Extended Data Figure 4**).

a. - For instance, I fail to understand what Fig S4 is showing. Is this a trajectory analysis - I don't see the specific trajectory that is referred to? This really needs more details (a lot more than referring to a paper that is only available on BioRx; ref. 23). How were these trajectories constructed? This is quite confusing.

We have combined the different sample types into an integrated representation, showing the different classifications and epidermis (IFE) populations. While the dataset does show a biological trajectory of cells from basal/cycling to differentiating populations, we do not show pseudotime analysis (see **Rebuttal - Figure 3**). To better (i.e., more intuitively) visualize the ongoing differentiation process, we now show clustering analysis and corresponding changes in gene expression from basal to suprabasal cell identity (**Manuscript - Figure 3g, h**). Concerning differentiation trajectories, please also see answer in d. below.

b. - Is Ki67 sufficient to define the G2M status of cells – additional markers could be shown.

The cell cycle analysis was performed with a list of 43 S and 54 G2M specific genes (Tirosh I et al., 2015) and using the 'score_genes_cell_cycle' function implemented in Scanpy. Ki67 (and the newly added Top2a) are exemplar cell-cycle related genes to better characterize the identified clusters and visualize actively cycling cell populations on the UMAP. The cell cycle scoring is described in the methods, and we included two cell cycle representative genes (Mki67, Top2a) as violin plots in the **Manuscript - Figure 3h** as well as the G1/S/G2M scoring of cells in UMAP (**Manuscript - Extended Data Figure 4g**).

- c. - And is keratin 10 expression (one marker) sufficient to define 3 different epithelial cell states? In scRNA-seq data you do have the possibility to look at many additional genes or gene signatures, in addition to those assessed by immunofluorescence. I feel this should be explored much better throughout. The same applies for stemness markers.

We would like to apologize for the lack of clarity and thank the reviewer for pointing it out. In our recent work (Cockburn, Annusver et al. NCB 2022; previous BioRxiv ref. 23) we defined the molecular trajectory of the epidermal differentiation process (Cockburn, Annusver et al. Figure 2; shown below), where we show that the onset of *Krt10*

expression marks epidermal stem cell commitment to differentiation. I.e., *Krt10* expression level is a good proxy to visualize the non-committed stem cells and differentiation-committed progenitors in the scRNA-seq data (original Fig. S4; Keratin 10 - UMAP). However, the classification of cells into the 3 main differentiation-related populations (original Fig. S4; Basal non-committed to Delaminated - UMAP) was done by nearest-neighbor-mapping of the dataset onto the Cockburn et al. 2022 differentiation trajectory (Cockburn, *Annusver et al. Figure 2e*) where we characterized and classified these different cell populations based on a much higher number of genes. To better clarify this point, we now show *Krt10* as well as additional differentiation-related markers (*Sbsn* and *Flg2*) to show the later stages of differentiation, and both *Krt14* and *Krt5* to confirm the basal (stem) cell status (**Manuscript - Figure 3h**).

- d. - Are we supposed to see differences between the trajectories? If so, I fail to see such differences because the appropriate quantification/statistics is not provided.

The reviewer is correct in that quantitative differences between trajectories cannot be assessed from this data presentation. Displaying expression of stem- and differentiation-associated genes along the differentiation trajectory, we indicate the differentiation status of cells within clusters, while emphasizing the injury-responsive cells. In the **Manuscript - Figure 3g-h** it can be better seen that wound cells also undergo similar gradual differentiation from more stem-cell like (*Krt5*, *Krt14* high; *Krt10*, *Sbsn* low) towards differentiated cells (*Krt5*, *Krt14* decreased; *Krt10*, *Sbsn* high).

3. Same applies to Fig S5. More details are needed.

- a. - Is keratine6a sufficient to distinguish between healthy cells and cells that respond to injury (blue versus yellow) ? Can additional markers be included ? This would be more convincing.

We sincerely thank the reviewer for this question, which resulted in a substantially improved presentation of robustness of wound-cell selection and its visualization.

We would like to first emphasize that the wound-cell selection, which we had characterized for each of our mouse genotype and conditions separately (HMU, HMW, WTMU, WTMW; see **Rebuttal - Figure 4**), was based on ‘wound-cluster membership’ and was not solely based on *Krt6a* expression. We utilized *Krt6a* expression to confirm the wound-responsive cell classification, because it has been previously shown that *Krt6a* is a robust marker for wound-responsive IFE keratinocytes (Joost et al. 2018; and K6 protein staining shown in **Manuscript - Extended Data Figure 4e** – previous Fig. S5).

To confirm the robustness of the wound cell identification with an independent computational approach, we now integrated all 12 samples at once and performed Milo (miloR) analysis (Dann E et al., 2021) to test for differential abundance of “injury-responsive” cells between the uninjured and the injured datasets.

Reassuringly, the results show that the injury-enriched neighborhoods match with the previously annotated injury-responsive cells (**Manuscript - Extended Data Figure 4d**). We also counted the contribution of all individual replicates (uninjured and injured) within the injury-responsive clusters (see **Manuscript - Figure 3g**, clusters 3,4,5) which show that around ~10% of injury-responsive cells come from uninjured samples (replicate names ending with “U”) (**Rebuttal - Figure 4**). Such a minor contribution is expected and can be explained by the existence of stressed or injury-responsive-like cells in the uninjured dataset, which show transcriptional similarities with actual injury-responsive

cells. Overall, these analyses support the assignment of clusters 3, 4 and 5 as injury-responsive (**Manuscript - Figure 3g-i**).

To further enhance clarity, we now included the markers *Krt6b*, *Krt16* and *Krt17* in addition to *Krt6a*, which enables for the reader a quicker identification of the three injury-related clusters (**Manuscript - Figure 3g-h**). Finally, we also performed a differential gene expression analysis comparing all the injury-responsive cells to the homeostatic cells, further highlighting *Krt6a*, *Krt6b*, *Krt16* and *Krt17* as among the best markers to identify this population (**Rebuttal - Figure 5**).

Rebuttal - Figure 5. Volcano plot showing the differential gene expression analysis between all injury-responsive cells and homeostatic cells. Shown are log₂-foldchanges on x-axis, absolute z-score values underlying the p-value computation on the y-axis and -log₁₀(adjusted p-value) encoded as the size of the points. Highlighted cells in blue have adjusted p value < 0.05 and absolute log₂foldchange > 1.

Note that the more conventional way of showing -log₁₀(adjusted p value) on the y-axis was not optimal for showing the marker genes in this case, because all of the annotated genes (and other genes with -log₁₀(adjusted p-value) of 300 formed a “line” of inseparable points on the top of the plot due to their p-value being 0.0 (lower than the computable limit in Scanpy) – noted as assigned p value of 1e-300.

- b. - Are we supposed to see a difference between the models (injured wild-type mosaic versus injured HRAS mosaic) because no statistics is provided.

We thank the reviewer for this excellent question of whether there is a statistically significant difference in cell differentiation between the distinct conditions. I.e.: the original Figure S5C was based on the approach used in *Cockburn, Annusver et al 2022- Figure 2g* to visualize gene expression changes along a differentiation-pseudotime

ordered from basal to suprabasal cells. It is still debated, and an unresolved challenge in the field, how to best align pseudotimes of cells from distinct conditions (e.g.: uninjured, injury-repair, expression of $Hras^{G12V/+}$). Thus, the original Figure S5C was indeed a qualitative data representation. While there are packages available that can compare and align pseudotimes from different analyses of similar cell populations (e.g., Capital and cellAlign) (Sugihara R et al., 2022, Alpert A et al., 2018), they did not properly work for combining our data which was not entirely unexpected. In brief, the alignments were confounded by 2 major factors – i.e.: large populations of cycling cells (clusters 0 and 1) as well as injury-responsive cells (clusters 3, 4, 5) which also include their own sub-populations of cycling cells.

To assess if there is a statistical difference between the distinct conditions, we again used nearest-neighbor-mapping to assign every cell their most likely cell state (*basal stem cell*, *basal committed cell*, *suprabasal cell*) based on the cells previously defined in Cockburn, Annusver et al. 2022. We then focused our analysis on basal cells, determining and comparing the respective ratio of basal stem cells vs committed progenitors for all conditions, which is shown in the **Manuscript - Figure 3j**. In the **Manuscript - Figure 3k**, we additionally provide associated violin plots for *Krt6a* (a representative injury-induced gene) and *Krt10* (a representative marker for committed basal cells). In sum, our analysis confirms that injury repair increases the proportion of differentiating cells.

- c. - What is Fig 5SC showing? I guess it provides evidence for ‘the earlier differentiation’ in the injury model. But is this effect significant? I don’t see any confidence intervals, and also the binning strategy is unclear. Why do you consider 8 bins and why not do this in a continuous manner?

We would like to refer the reviewer to our answer in b, just above.

- d. - In Fig S5D – a pathway analysis is shown, but the corresponding differential gene expression analysis is not given. At the very least a volcano or list of differentially-expressed genes should be provided allowing the reader to assess which genes and how many of them are contributing to the pathways identified.

We have now included the respective differential gene expression volcano plots, highlighting genes that were used to generate the GO terms (**Manuscript - Extended Figure 4i, j**).

- e. - Finally, in the manuscript other methods are used to show that apoptosis or proliferation are (not) changed. These could also be further explored in the scRNA-seq data.

Apoptosis is unfortunately difficult to determine with conventional scRNA-seq studies because we actively eliminate cells that are undergoing apoptosis during different filtering steps (e.g., near-empty droplets, low reads per cell, high fraction of mitochondrial reads). This process is carried out to get good quality data for downstream gene expression analysis. Thus, the final dataset including all the existing annotation is not optimal for highlighting apoptotic cells. For cell proliferation we assessed the proliferation signature that emerged in GO-terms for biological process in uninjured and injured settings when comparing $Hras^{G12V/+}$ -mosaic to wild-type-mosaic models. This analysis show that $Hras^{G12V/+}$ cells display an enrichment in proliferation-associated GO-terms in comparison to wild-type cells in an uninjured setting, a GO-term enrichment that did not emerge in the same comparison of cells in an injured setting, further corroborating our live imaging observations. We now have also included the volcano plots underlying the GO terms (**Manuscript - Extended Fig. 4i, j**).

4. Additional details are also needed for Figure S7.

- a. - For instance, in panel A it is very unclear how the green subcluster was identified. It is referred to as the ‘source of EGFR ligands’ subcluster, but this is vague. I would really recommend the authors to properly annotate the subclusters with additional marker genes (find marker gene function in Seurat).

We have now updated the nomenclature assigning more neutral cluster names. The new cluster names are based on the cell’s homeostatic vs injury-associated cluster assignment, informed by the sample contribution of

uninjured and injured conditions, as well as differential gene expression analysis (see newly added dot plot in **Manuscript - Extended Data Figure 6f**), and by previous literature (Korosec A et al. 2019).

- b. - In Fig 7SC, several conditions are compared. But, since in the main manuscript data are only very briefly referred to, it is unclear if there are differences between groups. If this panel is indeed meant to indicate differences, proper statistical analyses should be performed (comparing abundancies of subclusters for the 3 mice tested between each of the subgroups). Also, the number of cells detected for some subclusters is quite low.

These panels are qualitatively showing the distribution of cells from different datasets and that every cluster has contributions from several different datasets (**Manuscript - Extended Data Figure 6d**). Moreover, it highlights that the homeostatic cells come from both uninjured and injured samples, while the injury-responsive clusters – especially the upper-right cluster enriched for EGFR ligands – consists almost exclusively of injury-sample cells.

- c. - Fig 7SE shows a volcano plot and highlights numerous genes that are differentially expressed. These include 2 genes from the EGFR family. It should be made clear in the main manuscript that besides these 2 genes, numerous other genes also differentially expressed and that these could therefore also act as potential mediators.

We agree with the reviewer's suggestion and have updated the text accordingly (**Manuscript - page 10; lines 16 - 19**).

- d. - Figure 7SF/G: it might be better to also show median values and individual dots on the violin plots. Regarding statistics, it would be better to additionally perform statistics per mouse tested rather than comparing expression between all individual single cells between conditions (irrespective of the number of mice tested).

We have now updated all the violin plots to include a boxplot, showing the 25, 50, 75 quartiles of the data. Including the individual data points appeared less clear as the dots are covering the underlying violins (**Rebuttal - Figure 6**). We also updated the statistical testing according to the reviewer's suggestion (t-tests comparing the mean expression levels per mouse tested for the appropriate genotype and condition, rather than comparing expression between all individual single cells between conditions), which is now also presented in the figures as p values (**Manuscript – Extended Data Figures 6h, i and 7f, g**).

5. Fig S8E and F: Several of the genes are barely expressed yet significant between healthy cells and cells that respond to injury. I wonder to what extent this bears biological significance. If a gene is barely expressed, it might be better to just indicate that expression is very low (and probably not biologically relevant).

We have updated these violin plots according to the previous points by including a boxplot and performing statistical testing by comparing means of biological replicates. Due to the genes being expressed in only a small number of cells, this analysis indeed shows no significant differences between uninjured and injured cells in the immune dataset (**Manuscript - Extended Data Figure 7f**). The only remaining significantly differentially expressed gene Hbegf still exhibits a heterogeneous expression pattern, with no clear pattern in specific wound cell types showing increase expression (**Rebuttal - Figure 7**)

Reviewer Reports on the Second Revision:

Referee #1 (Remarks to the Author):

My concern related to the arguments brought forward by the authors around increased proliferation remains. The authors assess the fraction of pHistoneH3-S10P positive cells, cells that are undergoing mitosis (essentially the same as pHistoneH3-S10P) and the fraction of cells that express keratin10. Yet, they never measure the length of the cell cycle nor the probability of cells to undergo differentiation. I agree with the authors that this cannot be measure by the angle of cell division, however, the authors still need to quantify and qualify what they mean by a “selective increase in wild-type cell proliferation”. Do they mean that the cells are proliferating faster or that cells are undergoing imbalanced cell fate decisions? If it is the former and there is no change in the probabilities to differentiate WT clones will never be able to outcompete their neighbors. If however it is the latter and the majority of cells upon cell division will maintain their basal features, WT cells will eventually be able to outcompete their HRas mutant neighbors not necessarily by an increased replication rate, but via imbalanced cell fate decisions. There will essentially be more cells that can go on to divide. Similarly, these arguments are also relevant when the authors discuss the effect of Ras mutants during “steady state” is it increased replication rates or imbalanced cell fate decisions?

The authors therefore need to somehow address this as it is the basis for their entire paper. The title states “Injury induces wild-type cell proliferation to suppress oncogenic Ras cells”. Yet, they never qualify what they mean by proliferation beyond assessing the fraction of cells undergoing mitosis. Is this proliferation? It might be from a population perspective, but this can be achieved via a number of different routes, and this is essentially the essence of the story the authors are trying to tell here. Again, I would like to refer the authors to (Andersen et al., 2019, Nature Cell Biology) where using quantitative fate mapping it is demonstrated that KRasG12D activation in the sebaceous gland has no effect on replication rates, but rather an effect on cell fate decisions.

Referee #4 (Remarks to the Author):

While the authors did address most of the points raised by the reviewers this time around, the rebuttal to the point about cancer is not very satisfying. If there is not an experimental way to show that the proliferation bias between wt and ras-mutated cells has a role in the initiation of cancer, then this idea should be moved strictly to the discussion. The story presented is nice and well founded, it is just not clear from the data presented that it has anything to do with cancer.

Referee #5 (Remarks to the Author):

I have now been able to review the scRNAseq analyses from the paper of Gallini et al. in detail. Compared to the original manuscript some rough quality metrics, the number of mice and cells retrieved per experiment have been mentioned. There are each time also the primary UMAP, the marker genes used, etc. while violin and volcano plots have now also been incorporated. Where

appropriate statistics have been added. Overall, I think the scRNAseq data are now much more clear/self-explaining (especially in extended data figure 4, 6 and 7) and as such they are also much better integrated in the manuscript. The only (very minor) comment I could come up with is that on Extended data figure 6D perhaps the percentage of positive cells could be included on the figure (to make it more clear whether there is a difference or not). With respect to the rest of the manuscript, I have also no further comments. It was clearly written and the findings were exciting.

Author Rebuttals to Second Revision:

Referee #1 (Remarks to the Author):

My concern related to the arguments brought forward by the authors around increased proliferation remains. The authors assess the fraction of pHistoneH3-S10P positive cells, cells that are undergoing mitosis (essentially the same as pHistoneH3-S10P) and the fraction of cells that express keratin10. Yet, they never measure the length of the cell cycle nor the probability of cells to undergo differentiation. I agree with the authors that this cannot be measure by the angle of cell division, however, the authors still need to quantify and qualify what they mean by a “selective increase in wild-type cell proliferation”. Do they mean that the cells are proliferating faster or that cells are undergoing imbalanced cell fate decisions? If it is the former and there is no change in the probabilities to differentiate WT clones will never be able to outcompete their neighbors. If however it is the latter and the majority of cells upon cell division will maintain their basal features, WT cells will eventually be able to outcompete their HRas mutant neighbors not necessarily by an increased replication rate, but via imbalanced cell fate decisions. There will essentially be more cells that can go on to divide. Similarly, these arguments are also relevant when the authors discuss the effect of Ras mutants during “steady state” is it increased replication rates or imbalanced cell fate decisions?

The authors therefore need to somehow address this as it is the basis for their entire paper. The title states “Injury induces wild-type cell proliferation to suppress oncogenic Ras cells”. Yet, they never qualify what they mean by proliferation beyond assessing the fraction of cells undergoing mitosis. Is this proliferation? It might be from a population perspective, but this can be achieved via a number of different routes, and this is essentially the essence of the story the authors are trying to tell here. Again, I would like to refer the authors to (Andersen et al., 2019, Nature Cell Biology) where using quantitative fate mapping it is demonstrated that KRasG12D activation in the sebaceous gland has no effect on replication rates, but rather an effect on cell fate decisions.

We appreciate the reviewer’s close engagement with our work and understand their point and interest about the changes in mitotic activities that we have captured. While investigating in-depth cell fate determination is an interesting topic, we would like to point out that these additional experiments will not change the main discovery and message of the paper about the Hras^{G12V/+} cell expansion being suppressed upon injury in mosaic skin and the role of the EGFR pathway as a key molecular mechanism.

In addition, reviewer’s comment that “if there is no change in cell fate decisions simply increasing proliferation will not provide a selective advantage” is correct when the renewal/differentiation rate of a homeostatic tissue is 0.5 (i.e. exactly half of all daughter cells maintain the progenitor potential). In our prior analyses of wild-type epidermal homeostasis, we observed an ever-slight bias towards progenitor renewal (Rompolas, Mesa et al., Science 2016; Fig. 2a), that may be there to prevent/account for progenitor loss through drift. In addition, Beronja’s group used two independent assays in their analyses of renewal rates of IFE progenitors (Ying et al., NCB 2018) and arrived at a similar rate of 0.52-0.55. Such a renewal rate is sufficient to facilitate expansion of a mitotically more active, proliferative cell population, and the phenomenon we report in our study.

In our revised manuscript we have now added a short discussion on the inter-dependence of proliferation and progenitor renewal/differentiation in population dynamics among IFE progenitors (*page 16 lines 1-7*) and modified title and abstract (*changes highlighted in red in the Manuscript*). Moreover, as a reflection of how seriously we considered the reviewer’s request to expand our study into the progenitor renewal field here we discuss a number of general issues we would encounter, followed by a detailed response to the suggested experiment and its challenges, as well as providing an answer to the core question of the reviewer to *what we mean by a “selective increase in wild-type cell proliferation”*.

First, while the additional experiments are expected to provide further insights, they will require substantial extra resources including time: we would need roughly eight to nine months from generating mouse models that do not contain the K14H2B-GFP fluorescent line (in order to free a microscope channel to accommodate staining), establish assays for BrdU labelling and detection in whole mounts, generating all replicates in the various conditions and perform quantifications. Additionally, as we examined the mitotic activity we saw that it is location dependent (i.e.: border between $Hras^{G12V/+}$ and WT cells or center in the two populations; data not in the manuscript) creating more variables to be scored/taken into account therefore extending the time needed to perform these quantifications.

Second, our recently gained knowledge that the initiation of differentiation (marked by K10+ expression) is both temporally and functionally uncoupled from cell cycle exit (Cockburn, Annusver et al., NCB 2022) adds a new level of complexity for the proposed cell fate analysis (expanded in response to experimental suggestion below).

Overall, such a revision would provide data for an entire new manuscript shifting the focus from the *new mechanism of how WT cells can keep oncogenic cell expansion in check toward single cell dynamics in mosaic tissue*.

Response to experimental suggestion:

The optogenetic solution is at present not established yet in our field. However, we have carefully considered the editor and reviewer 1 experiment to interrogate fate determination by scoring for K10 distribution between daughter cells in a single time point after a single BrdU pulse and a short chase (4 hours). There are several caveats around this experiment based on our recent work where we tracked the exact fate of cells that turn on K10 by revisiting the same area of the skin over time (Cockburn, Annusver et al., NCB 2022):

1. K10 expression, one of the earliest commitment markers, is not exclusively expressed after cell division but cells could be born K10 negative and then turn it on within hours or days after birth. After a cell turns on K10 expression, it can still divide before leaving the basal layer, with the decision and time required for either process varying depending on the necessity of the entire skin to preserve its barrier function.

2. In an injury setting (as shown in Cockburn, Annusver et al., NCB 2022 via the use of tape stripping), the dynamics of K10 expression and K10 cell occupancy are significantly changed, which makes direct comparison of uninjured and injured settings challenging and highly time intensive in order to control for all variables.

3. While Aragona et al., 2020 focused on homeostatic conditions, the higher mitotic activity in the wound context decreases confidence in the scoring of daughter cells versus proliferation happening at the same time in neighboring cells (**Rebuttal Figure 1**). Therefore, such analysis is not interpretable. Furthermore, the cytoplasmic tdTomato reporter is quenched by the BrdU staining protocol, making it even more challenging to track the $Hras^{G12V/+}$ cells.

In sum, to perform a correct characterization of the differentiation process a long-term tracking would be the most insightful approach considering that proliferation occurs before and after commitment to differentiation (i.e. measured by K10 expression). To track this process in the injured condition, a series of troubleshooting are required to provide what we discovered in the uninjured skin in Cockburn, Annusver et al., NCB 2022 in our new context (mosaic and wound).

For all the reasons above and the complexity of the differentiation process, we adopted a fate analysis at population level that will give an overall idea of how many cells are committed to go up into the suprabasal layer by scoring the K10+ cells in WT and Hras^{G12V/+} areas at Day-3 and Day-7 PWI, where we observed differences in the respective proliferative rates of WT and Hras^{G12V/+} cells (Figure 2b, d and Extended Data Figure 3i, j). The analyses of angle of cell division we provided in the previous rebuttal were helpful towards this point. Should the angle have been found to be *perpendicular* it would have indeed, as the reviewer pointed out, nullified the daughter cell contribution to the occupancy of the basal stem cell layer upon division. However, we have identified that *all divisions* are parallel to the basement membrane in both uninjured and injured contexts and thus cell division rate directly affects the occupancy of the basal cell layer. To further validate this last point, we scored the density of the basal layer in WT and Hras^{G12V/+} areas and identified that in the uninjured condition Hras^{G12V/+} cells are denser than WT neighbors and in injured condition we observed the opposite. These results are consistent with the proliferative rates we quantified at Day 3 PWI (**Rebuttal Figure 2**).

Regarding proliferation rate, we further would like to highlight that the overall proliferative capacity within the epidermis (i.e.: the sum of WT and/or Hras^{G12V/+} divisions together) do not change between wild-type-mosaic and Hras^{G12V/+}-mosaic models. The distribution of proliferative events transiently shifts between WT and Hras^{G12V/+} cells in the Hras^{G12V/+}-mosaic injury context. We explained this transient shift molecularly. WT cells compensate for: 1) the unresponsive behavior of Hras cells to EGFR ligands secreted during wound-repair and 2) the generally higher differentiation capacity of Hras^{G12V/+} cells. WT cells are not expanding over Hras^{G12V/+} cells but both populations together maintain the same occupancy of the basal layer. Indeed, after the spike in WT cell proliferation at Day 3 PWI, the proliferative rates shifted toward a similar rate between WT and Hras^{G12V/+} cells at Day 7 PWI (which is maintained at least up until 30 days indicating that Hras^{G12V/+} are not taking over during this time window), and also the differentiation rates became similar between WT and Hras^{G12V/+} cells at Day 7 PWI.

REDACTED

In sum, our study unequivocally uncovers a previously unrecognized mechanism of how WT cells can keep oncogenic cell expansion in check via the unresponsive behavior of Hras^{G12V/+} cells to respond to the wound environment and the capacity of WT cells to compensate this deficiency and preserve a healthy wound repair process.

Referee #4 (Remarks to the Author):

While the authors did address most of the points raised by the reviewers this time around, the rebuttal to the point about cancer is not very satisfying. If there is not an experimental way to show that the proliferation bias between wt and ras-mutated cells has a role in the initiation of cancer, then this idea should be moved strictly to the discussion. The story presented is nice and well founded, it is just not clear from the data presented that it has anything to do with cancer.

We thank the reviewer for their comment and positive evaluation of our discoveries. To integrate the reviewer suggestion in the manuscript, we moved the possible application of our discoveries on tumor prevention only in the discussion section.

Page 10 line 8-10 and Page 14 line 13-15 (changes highlighted in red in the Manuscript)

Reviewer Reports on the Third Revision:

Referee #1 (Remarks to the Author):

Specific comments:

The authors never qualify what they mean by proliferation, and this still represents a problem in the way certain parts of the manuscript has been written. Do the authors mean that more cells are proliferating (higher proportion of progenitors), or do they mean that the proliferation rate (cell cycle time) is decreased. They only show that the fraction of pHist H3 and M-phase cells is increased, which could be explained by both options.

Textual suggestions to reflect the included data. These are just examples:

Line 26-27: increase in the fraction of proliferating wild-type cells

Line 30-31: diminished the proportion of dividing wild-type cells

Line 31-33: Is not accurately reflecting the data – what do authors mean by proliferation?

Line 36: “proliferative” could be exchanged with selective

Line 123-125: “To examine the proliferation rate, we scored mitotic cells in uninjured or injured skin by immunostaining for the mitotic marker phospho-Histone3.” This is not a measure for proliferation rate, but rather the number of cells in M-phase. The following sentence should also reflect this. States right now “an increase in epithelial cell proliferation”, when it is an increase in the number of dividing cells.

Line 126-128: “In sharp contrast, although we observed an increase in the mitotic events of wild-type cells in the HrasG12V/+ mosaic model, HrasG12V/+ cell proliferation was unaltered during repair (Fig. 3b). These findings were corroborated by measuring mitotic figures (Extended Data Fig. 3a, b, c).” Looking at phospho-Histone3 and mitotic figures is looking at exactly the same phase of the cell cycle. Can they truly state that this further substantiate the observation that there are more mitotic events?

Line 132-142: “In the wild-type-mosaic model, the initial increase in proliferation observed in both tdTomato+ and tdTomato– wild-type cells returned to baseline by 7 days PWI”. For clarity this should be “the number of dividing cells”. In the remainder of this paragraph the authors repeatedly used the word proliferation without qualifying what they mean.

Line 202-203 “We found that injury-repair in mosaic skin triggers a specific increase in the proliferation of wild-type cells but not KrasG12D/+ and HrasG12V/+ cells.” Exchange proliferation with “fraction of dividing cells”.

Line 249-250: “We hypothesized that HrasG12V/+ cells downregulate p21 expression to promote a high proliferation rate.” What do the authors mean by proliferation rate? They never assess the speed of cell cycle, and it could simply reflect that more cells retain the ability to divide. The authors again need to be much more precise in their description of the actual hypothesis and data.

Line 258-259: “mimicking the selective increase in wild-type cell proliferation during injury-repair of HrasG12V/+ mosaic mice”. The authors have deleted selective in all other parts of the manuscript.

Line 262-264: “At the same time, differentiation of wild-type cells was also accelerated by loss of p21, which explains why wild-type cells did not out-compete HrasG12V/+ cells in the HrasG12V/+ p21null mosaic model (Extended Data Fig. 8j).” The authors never measure dynamics of differentiation and cannot conclude on speed. They measure the fraction of cells of a given phenotype as a snapshot.

Line 281-284: “Overall, these data suggest that injury-repair or loss of p21 specifically increase the activity of a downstream effector of Ras, ERK1/2, in wild-type cells to increase their proliferation, enabling them to effectively suppress the competitive advantage of oncogenic Ras-mutant cells in mosaic mice.” The authors need to qualify what they mean by proliferation or at least mention the options 1) increases the fraction of cells that retain the ability to divide or 2) increases the cell cycle speed. Currently they cannot exclude either or these, and they never reflect upon this until the very last part of the discussion.

Line 327-329: “Curiously, as the epidermis maintains a positive progenitor renewal rate (rate > 0.5)^{58,60}, possibly as a way to counter progenitor loss through neutral drift, changes in progenitor proliferation rate alone can substantially impact epidermal expansion over time.” This part of the conclusion does not make any sense. How can the renewal rate of a tissue be more than 50% in homeostasis. This will imply that the tissue is expanding. Ying et al., 2018, states in the paper that they observe a renewal rate of 0.5 in the adult epidermis during homeostasis.

Author Rebuttals to Third Revision:

Referee #1 (Remarks to the Author):

Specific comments:

The authors never qualify what they mean by proliferation, and this still represents a problem in the way certain parts of the manuscript has been written. Do the authors mean that more cells are proliferating (higher proportion of progenitors), or do they mean that the proliferation rate (cell cycle time) is decreased. They only show that the fraction of pHist H3 and M-phase cells is increased, which could be explained by both options.

We thank the reviewer for the helpful comments and detailed suggestions on the usage 'proliferation' in our manuscript. We have made the suggested textual changes, and overall, more stringently used the suggested phrasings.

Textual suggestions to reflect the included data. These are just examples:

All the textual changes in the manuscript are highlighted in red for ease of identification

Line 26-27: increase in the fraction of proliferating wild-type cells

We have made the requested textual change in the abstract.

Line 30-31: diminished the proportion of dividing wild-type cells

We have made the requested textual change in the abstract.

Line 31-33: Is not accurately reflecting the data – what do authors mean by proliferation?

In previous studies using p21null mice the term proliferation has been frequently used and also has been established in studies focusing on the skin (and its stem cells). Therefore, we feel that this term here – in light that we specify in the results what we exactly measure (number of mitotic events) – can be used.

Line 36: “proliferative” could be exchanged with selective.

We have made the requested textual change.

Line 123-125: “To examine the proliferation rate, we scored mitotic cells in uninjured or injured skin by immunostaining for the mitotic marker phospho-Histone3.” This is not a measure for proliferation rate, but rather the number of cells in M-phase. The following sentence should also reflect this. States right now “an increase in epithelial cell proliferation”, when it is an increase in the number of dividing cells.

We have made the requested textual change.

Line 126-128: “In sharp contrast, although we observed an increase in the mitotic events of wild-type cells in the HrasG12V/+ mosaic model, HrasG12V/+ cell proliferation was unaltered during repair (Fig. 3b). These findings were corroborated by measuring mitotic figures (Extended Data Fig. 3a, b, c).” Looking at phospho-Histone3 and mitotic figures is looking at exactly the same phase of the cell cycle. Can they truly state that this further substantiate the observation that there are more mitotic events?

We have used two approaches to validate the number of divisions: leveraging two photon microscopy in live mice tracking nuclei by histone H2B GFP and immunostaining of phospho-Histone3. Thus, the immunostaining results corroborates the results captured by live imaging.

Line 132-142: “In the wild-type-mosaic model, the initial increase in proliferation observed in both tdTomato+ and tdTomato- wild-type cells returned to baseline by 7 days PWI”. For clarity this should be “the number of dividing cells”. In the remainder of this paragraph the authors repeatedly used the word proliferation without qualifying what they mean.

We have made the requested textual change in the remainder of the paragraph.

Line 202-203 “We found that injury-repair in mosaic skin triggers a specific increase in the proliferation of wild-type cells but not KrasG12D/+ and HrasG12V/+ cells.” Exchange proliferation with “fraction of dividing cells”.

We have made the requested textual change.

Line 249-250: “We hypothesized that HrasG12V/+ cells downregulate p21 expression to promote a high proliferation rate.” What do the authors mean by proliferation rate? They never assess the speed of cell cycle, and it could simply reflect that more cells retain the ability to divide. The authors again need to be much more precise in their description of the actual hypothesis and data.

We have updated this text part.

Line 258-259: “mimicking the selective increase in wild-type cell proliferation during injury-repair of HrasG12V/+ mosaic mice”. The authors have deleted selective in all other parts of the manuscript.

We have made the requested textual change.

Line 262-264: “At the same time, differentiation of wild-type cells was also accelerated by loss of p21, which explains why wild-type cells did not out-compete HrasG12V/+ cells in the HrasG12V/+-p21null-mosaic model (Extended Data Fig. 8j).” The authors never measure dynamics of differentiation and cannot conclude on speed. They measure the fraction of cells of a given phenotype as a snapshot.

We have removed the reference to the speed of differentiation and highlighted the measurement of the fraction of differentiating cells.

Line 281-284: “Overall, these data suggest that injury-repair or loss of p21 specifically increase the activity of a downstream effector of Ras, ERK1/2, in wild-type cells to increase their proliferation, enabling them to effectively suppress the competitive advantage of oncogenic Ras-mutant cells in mosaic mice.” The authors need to qualify what they mean by proliferation or at least mention the options 1) increases the fraction of cells that retain the ability to divide or 2) increases the cell cycle speed. Currently they cannot exclude either or these, and they never reflect upon this until the very last part of the discussion.

We have clarified that it's the fraction of dividing cells.

Line 327-329: “Curiously, as the epidermis maintains a positive progenitor renewal rate (rate > 0.5)^{58,60}, possibly as a way to counter progenitor loss through neutral drift, changes in progenitor proliferation rate alone can substantially impact epidermal expansion over time.” This part of the conclusion does not make any sense. How can the renewal rate of a tissue be more than 50% in homeostasis. This will imply that the tissue is expanding. Ying et al., 2018, states in the paper that they observe a renewal rate of 0.5 in the adult epidermis during homeostasis.

The paragraph was removed from the discussion part as suggested by the reviewer.